# Several supplementary concepts for applied category-theoretical states over an extended Petri net using an example relating to genetic coding: Toward an abstract algebraic formulation of molecular/genetic biology

**Jitsuki Sawamura** [1]*, **Shigeru Morishita**[2], **Jun Ishigooka**[3]

**1** Nishigahara Hospital, Tokyo, Japan, **2** Depression Prevention Medical Center, Inariyama Takeda Hospital, Kyoto, Japan, **3** Institute of CNS Pharmacology, Tokyo, Japan

* jsawamura246@gmail.com

**Data Availability Statement:** The data used in this manuscript were built up and prepared for the

## Abstract

Abstract algebraic concepts such as category are considered cornerstones on which logical consistency relies in any sophisticated study of natural phenomena. However, to the best of our knowledge, in molecular/genetic biology, their application is still severely limited because they capture neither the dynamics nor provide a visual form. The Petri net (PN) has often been used to illustrate visually parallel, asynchronous dynamic events in small data systems. A prototypal hybrid model combining both category theory and extended PNs may instead be indispensable for that purpose. This hybrid model incorporates 1) token-like elements of a group, 2) object-like places of a category, 3) square poles (rather than pentagon poles) that enable unique identifications of single-strand DNA sequences from the shape of its polygonal line, 4) creation/annihilation morphisms that generate/erase tokens, 5) Cartesian products '$Z_5 \times Z_2 \times \ldots$' that enable conversions between DNA and RNA sequences, 6) somatic recombinations (VDJ recombinations) for antibodies displayed concretely in category-theoretic form, 7) 'identity protein Δ' translated from a triplet of identity bases 'EEE' as an advanced concept from our previous display of the canonical central dogma, 8) illustrations of an incidence-matrix-like matrix A that includes operators as coordinates, and 9) basic topics concerning the canonical central dogma being displayed concretely using concepts of conventional category theory such as 'adjoint', 'adjoint functor', 'natural transformation', 'Yoneda's lemma' and 'Kan extension'. These ideas provide more advanced tools that expand our previous model concerning nucleic-acid-base sequences. Despite the nascent nature of our methodology, our hybrid model has potential in a variety of applications, illustrated using molecular/genetic sequences, in particular providing a simple dynamic/visual representation. With further improvements, this approach may prove effective in reducing the need for large data-storing systems.

purpose of elucidating the methodology, and were not collected from actual experiments or real-world sources. They were strictly created within the text to facilitate readers' comprehension. The data, represented in the form of equations, are exemplified within fictional scenarios and do not have a corresponding minimal data set underlying the results outside of the paper. While there are sections in which figures and methodologies are cited from other papers, this is exclusively confined to the aforementioned context (with proper citation).

**Funding:** The authors received no specific funding for this work.

**Competing interests:** The authors have declared that no competing interests exist.

# §1. Introduction

Because dynamic phenomena in molecular/genetic biology are ordinarily too complex to describe, descriptive models often do not provide sufficient visual imagery. To our knowledge, abstract algebraic concepts including monoid, group, ring, field, variety, and category are crucially important in classifying, integrating, and studying dynamic phenomena or systems in a rational and sophisticated manner. In particular, category theory is considered a cornerstone on which logical consistency arises in studying almost all biological phenomena in a sophisticated manner [1–3]. From that perspective, we suggested previously a basic prototypal model using group theory and related concepts [4–6]. However, the applicability of the category concept remains doggedly limited because the respective states of the category-theoretical model lack a visual and/or dynamic formulation with which users can draw on at any stage. Furthermore, we believe that an axiomatic and tautological form of category theory is not always sufficient to describe complex phenomena exhibited by living creatures—humans, eukaryotes, bacteria, and other genetic-based species. To describe systems involving parallel, asynchronous dynamic events visually, the Petri net (PN) is a mathematical language capable of rendering dynamic models of such phenomena and has often been relied upon for that purpose over the last five decades [7–10]. We envisage that a hybrid model of abstract algebraic and dynamic elements formed from a combination of categories and extended (modified/advanced) PNs (EPNs) may have the potential to broaden the applicability of existing systems especially in a more visual manner [11–13]. The PN is a specialized form/part of category theory, which we discuss later. However, we believe that hybridizing the object 'category' in a heterogeneous way may impose specific limitations on it but may be more advantageous when these limitations align with the characteristics of the specific systems, for example, those in biology. Because of its limited formalism, the hybrid model may be composable from a less-weighted procedure. Recently, in genetic biology, a PN model based on applied category theory was reported by Wu [14, 15] and called the 'open PN'[16]. We envisage that our model adds in part several meaningful supplemental devices to that concept. Although our methodology is rudimentary, a simple example is presented to help the reader understand its application to molecular/genetic biology, chemistry, physics, and mathematics. In particular, based on our previous work [4–6], we develop models that involve the canonical interpretation of the central dogma for which advanced devices are added in modeling deoxyribonucleic acid (DNA)/ribonucleic acid (RNA) base sequences. Despite the need for further improvements in the future, this rationally exhaustive formalism provides the basis on which the dynamics of the hybrid model can be understood visually via its dynamic changes in state.

# §2. Methods

## §2–1. Data availability statements

Regarding the availability statement in this paper, the data used in this manuscript were built up and prepared for the purpose of elucidating the methodology, and were not collected from actual experiments or real-world sources. They were strictly created within the text to facilitate readers' comprehension. The data, represented in the form of equations, are exemplified within fictional scenarios and do not have a corresponding minimal data set underlying the results outside of the paper. While there are sections in which figures and methodologies are cited from other papers, this is exclusively confined to the aforementioned context (see S1 File for a proper citation).

## §2–2. Preliminary explanation of categories and PNs

### §2-2-1. Postulates of category theory.
In general, category theory is a language that is used to express abstract concepts [8, 9]. The category is defined as the combination of source

object X, target object Y, and a directed arrow '→'. The directed arrow represents 'a morphism' that relates X and Y. Additionally, if f, g, and h are specified morphisms, f: X→Y; dom(f) = X, cod(f) = Y, g: Y→Z; dom(g) = Y, cod(g) = Z, and h: Z→W; dom(h) = Z, cod(h) = W, then the target of f is the source of g. Hence, the successive application of the two morphisms f and g is a new morphism, the composition being denoted f;g (= g˚f). Similarly, the target of g is the source of h yielding morphism g;h (= h˚g). Moreover, the successive application of all three yields composition f;g;h (= h˚g˚f): X→Y→Z→W, which obeys the law of associativity,

$$(f; g); h = f; (g; h) \text{ (alternatively, } h \circ (g \circ f) = (h \circ g) \circ f). \tag{1}$$

The above are the necessary conditions to fulfill the postulates of category theory.

In addition, for each morphism f, there exists morphisms '$1_X$' and '$1_Y$', a morphism $1_X$: X→X and a morphism $1_Y$: Y→Y, that satisfy the relationships necessary for the composition of a category, specifically

$$1_X \circ f = f = f \circ 1_Y. \tag{2}$$

If supplemented, and there exists (f) that satisfies 'f;(f) (= (f)˚f) = $1_X$' and '(f);f (= f˚(f)) = $1_Y$', (f) is called the inverse of morphism f (denoted by $f^{-1}$). However, the existence of an inverse for all morphisms is not a prerequisite of category theory.

**§2-2-2. Postulates of a classical PN.**   The PN by contrast is a directed bipartite graph. Invented by Carl Adam Petri [10], it is used as a language or a tool in designing or visualizing a parallel, discrete, asynchronous, distributed and dynamical system. In practice, the classical PN, as well as various types of extended PNs, have been applied to various fields such as the dynamics of molecular biological systems [7–33], genetic activity [34, 35], biological and flexible manufacturing systems [36, 37], multi-valued genetic regulatory networks [38], the algebraic application of Boolean differential calculus [39], programming languages [40], artificial intelligence [41], fuzzy PNs [42, 43], and quantum computation using quantum PNs [44, 45].

## §2–3. Illustrations of hybrid models based on category theory and EPN

At this stage, to help the readers' understanding, we illustrate the extended PN (EPN) using category theory [**Fig 1(A)**]. From the viewpoint of the standard EPN, the hybrid model that we present in this article may be regarded as one for which the standard EPN is composed using category theory as a material with some options. The main characteristics of our model may be summarized in three points: 1) places correspond to objects, 2) arcs correspond to morphisms, and 3) tokens are defined as those that can act as operators. For simplicity, throughout the article, we mainly consider a 'state machine' where any transition has only one input place and one output place [7–45]. However, several parts of our model are not 'state machines' because a description of more complex systems is needed. Although, as mentioned above, the PN is a specific form/part of a category, by hybridizing specific parts of the PN and that of the category in a heterogeneous way, we are able to give some rules to those that are lacking a corresponding standard PN: specifically, the compositionality (operationality) between 1) two morphisms, and 2) a token in a place and a morphism in the same place (to be described in detail later).

**§2-3-1. Places of an EPN having characteristics of a monoid/group.**   In this section, we define essential rules that are referred to in the Methods section.

***P*:** A finite set of places (symbol: ○). Place $p_k$ is defined as a substitution of an object of the category corresponding to set $A_k$. They are the objects of focus in the PN of the model. (Here, $k$ = 1,2,...,$m$ indexes the places; $m$ is their total number.) Each place $p_k$ includes elements of $A_k$; however, the selection of tokens may be arbitrary. The set $A_k$ may form a group or a related

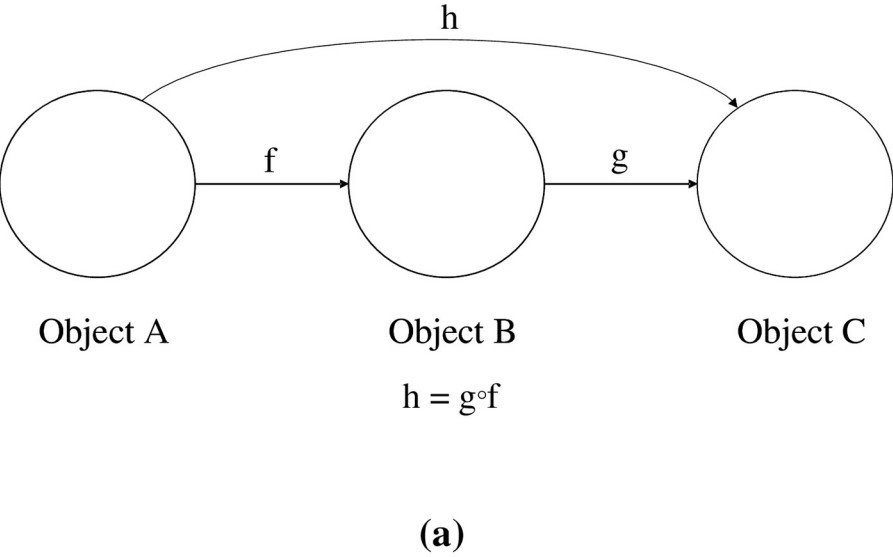

**(a)**

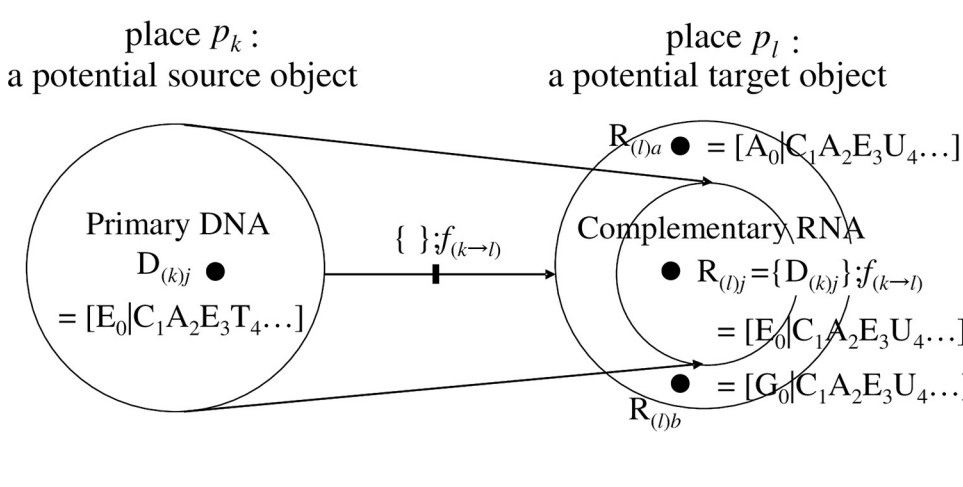

**(b)**

**Fig 1. Illustrations of extended Petri nets (EPNs).** (a). In the realm of formal system modeling, integrating category theory concepts within the framework of PNs presents an intriguing avenue for mathematical abstraction. The fundamental elements of category theory are objects A, B, and C and mappings f: A → B, g: B → C, and h: A → C, denoting morphisms between the objects. This abstract structure has a correspondence in PNs: **Places:** A (Place representing Object A). B (Place representing Object B). C (Place representing Object C). **Transitions:** f (Transition representing Arrow f). g (Transition representing Arrow g). h (Transition representing Arrow h). **Arcs:** Input arc from A to f (Signifying the starting point of Arrow f being Object A). Input arc from B to f (Denoting the endpoint of Arrow f being Object B). Input arc from B to g (Indicating the starting point of Arrow g being Object B). Input arc from C to g (Representing the endpoint of Arrow g being Object C). Input arc from A to h (Symbolizing the starting point of Arrow h being Object A). Input arc from C to h (Marking the endpoint of Arrow h being Object C). This PN representation elegantly captures the essence of category theory constructs, whereby transitions correspond to arrows and places represent distinct objects. The interconnected arcs delineate the relationships between these objects and morphisms, forming a cohesive and visually intuitive model. (b). A necessary relationship for the definition of category is '{potential source object $p_k$ (place $p_k$)};$f_{(k \to l)} \subseteq$ potential target object $p_l$ (place $p_l$)'. Hence, provided that '$D_{(k)j} = [E_0|$

$C_1A_2E_3T_4||E_5E_6...$]' is a token that is a potential source object in place $p_k$, then $R_{(l)j}$ is the image of it under morphism $f_{(k\to l)}$ (that is, $R_{(l)j} = \{D_{(k)j}\};f_{(k\to l)}$), '$R_{(l)j} = [E_0|C_1A_2E_3U_4||E_5E_6...] \in$ place $p_l$. The tokens belong to place $p_l$; they do not belong to '$\{$place $p_k\};f_{(k\to l)}$' as $R_{(l)a} = [A_0|C_1A_2E_3U_4||E_5E_6...] \notin \{p_k\};f_{(k\to l)}$', and '$R_{(l)b} = [G_0|C_1A_2E_3U_4||E_5E_6...] \notin\{p_k\};f_{(k\to l)}$'. That is, the initial coordinate of any DNA/RNA sequence can only be $E_0$ because, if $E_0$ were only a symbol, then no partial function for the morphism can occur, and the (potential) objects may satisfy the postulates of category theory, specifically, Eq (4).

object such as monoid, ring, field with a binary operation or other monoidal products (defined on a case by case basis for the model). To keep the discussion simple, we assume the existence of an identity element and optionally, that of inverse elements.

The pre-place $p_k$ is connected to a transition $t_{(k\to l)}$ by an arc, and the post-place $p_l$ is connected from a transition $t_{(k\to l)}$ also by an arc. In our model, and in most instances, an arc from a pre-place has no operational meaning, but an arc to a post-place $p_l$ has a specific operational meaning in that it denotes 'a morphism (transition) $f_{(k\to l)}$'. An input place $p_k$ includes a token such as $a_{(k)i}$, and an output place $p_l$ includes a token such as $a_{(l)i}$ in place $p_l$ after the firing of morphism $f_{(k\to l)}$' to a token $a_{(k)i}$ (here $l$ is an integer, $l = 1,2,...,n$; two situations arise: $k = l$ and $k\neq l$).

**Token:** A token $a_{(k)i}$ with symbol ● denotes a token in place $p_k$ that is an element of an operational set (most often a group) $A_k$ ($i$; integer, $i = 1,2,...$) or an accessorized element such as $c\cdot a_{(k)i}$ ($c$ is a constant factor that does not have an operational function).

Place $p_k$ includes '$a_{(k)i}$'s or $c\cdot a_{(k)i}$, and all elements of set $A_k$ (or that of accessorized elements like $c\cdot a_{(k)i}$) are not always included simultaneously. Additionally, duplicated inclusions of the same elements '$a_{(k)i}$'s (or '$c\cdot a_{(k)i}$'s) in place $p_k$ are freely permissible. All tokens '$a_{(k)i}$'s (or '$c\cdot a_{(k)i}$'s) are included independently. An empty set (no token) is denoted by $\phi$.

***M***: A $w$-th marking of place $p_k$ is denoted $M_w(p_k)$ ($w$-th session, $w$ is the session number for marking, $w = 1,2,...$).

***T***: A finite set of morphisms $f_{(k\to l)}$ is equivalent to transitions $t_{(k\to l)}$ (symbol: '$-|\to$'). A $t_{(k\to l)}$ is defined between places $p_k$ and $p_l$. For '$k = l$', $t_{(k\to l)}$ is written $t_k$. A set of arcs (symbol: $\to$) is equivalent to a morphism/transition $f_{(k\to l)}$ ($k = l$ or $k\neq l$). Therefore, we substitute arcs (conventionally denoted by ***F***) by a morphism (transition), and transition $f_{(k\to l)}$ or $t_{(k\to l)}$ is integrated as a complex case of a morphism (transition). Arcs that are fireable/non-fireable are considered.

**§2-3-2. Firing rules: For a morphism (transition), there are two types of firing rules labeled I and II.** *Firing rule I: Firing defined between different places (via a trans-place morphism)*. A morphism $f_{(k\to l)}$ (transition $t_{(k\to l)}$) with '$k\neq l$' is defined between input and output places where an arc from the pre-place $p_k$ to the post-place $p_l$ is integrated with transition '|' as '$-|\to$'. Then, the morphism $f_{(k\to l)}$ ordering a token '$a_{(k)i}$' (or $c\cdot a_{(k)i}$; in the following description; this case may be omitted for simplicity without loss of generality) ($a_{(k)i}\in A_k$), denoted by '$\{a_{(k)i}\};f_{(k\to l)}$:' or '$f_{(k\to l)}°a_{(k)i}$:' is fireable when a token $a_{(k)i}$ is included in place $p_k$. We call $f_{(k\to l)}$ a 'trans-place morphism'. Through a firing of a morphism (transition) $f_{(k\to l)}$ ordering '$a_{(k)i}$', token $a_{(k)I}$ is removed from the input place $p_k$, and $a_{i(l)i}$ (= $\{a_{(k)i}\};f_{(k\to l)}$, the images of $a_{(k)i}$ by $f_{(k\to l)}$) ($f_{(k\to l)}\in A_1$) produced in the output place $p_l$, determined by the 'morphism $f_{(k\to l)}$: $a_{(k)i}\to a_{(l)i}$. Examples are illustrated in (**Fig 2A and 2B**). The firing rule for this is stipulated as follows; subject to '$k\neq l$',

$M_{w+1}(p_l) = M_w(p_l) + \{a_{(k)i}\};f_{(k\to l)}$: $p_l\in$(output place of the transition) ^ $p_l\notin$(input place of the transition). $P_k\in$(input place of the transition) ^ $p_k\notin$(output place of the transition),

$M_{w+1}(p_k) = M_w(p_k)-a_{(k)i}$: $p_k\in$(input place of the transition) ^ $p_k\notin$(output place of the transition), and

$$M_{w+1}(p_k) = M_w(p_k), M_{w+1}(p_l) = M_w(p_l) : \text{otherwise}, \tag{3}$$

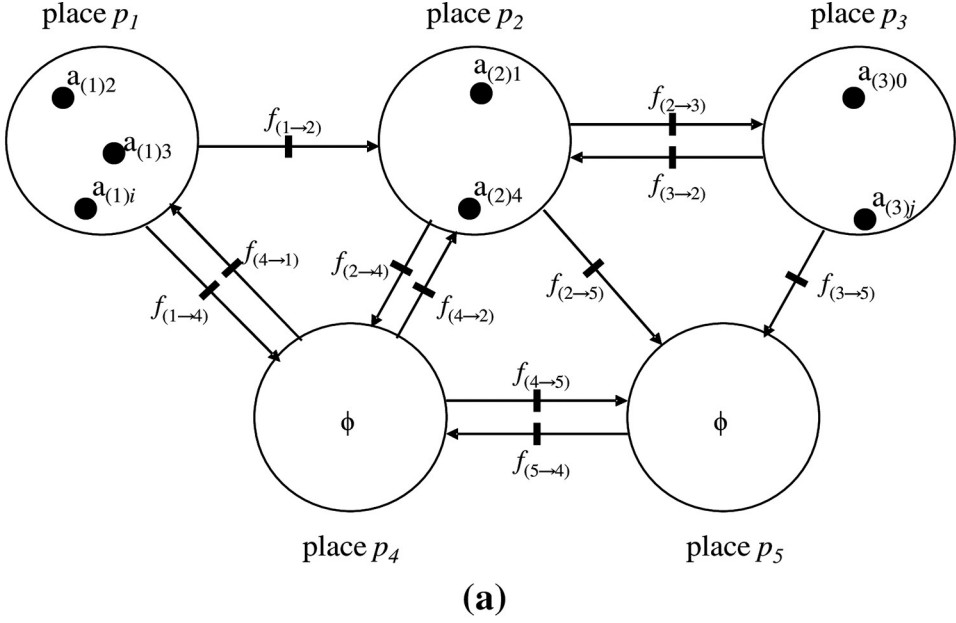

**(a)**

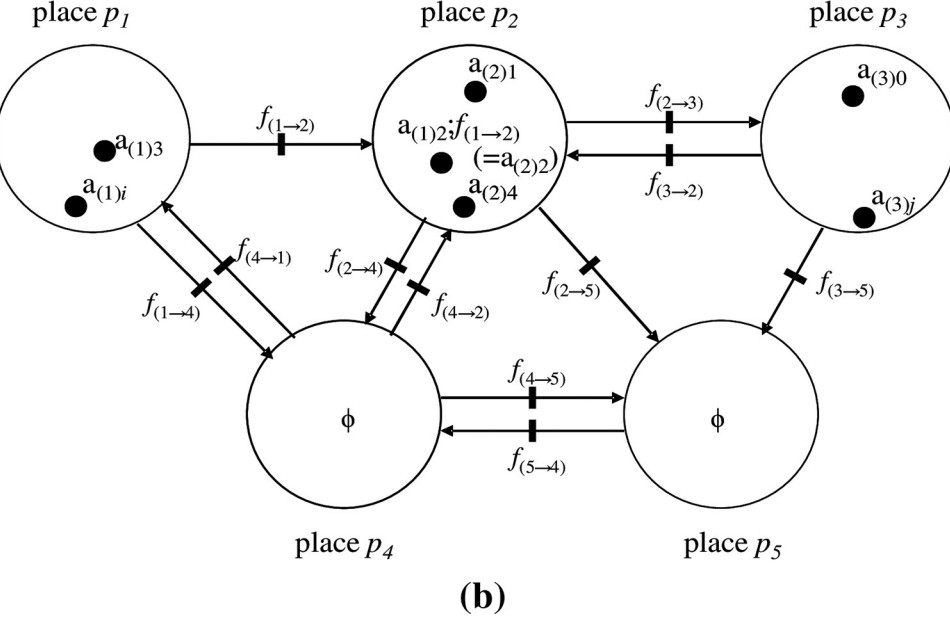

**(b)**

**Fig 2. An example of a state in the standard Petri net (PN).** (a). A state that exemplifies markings is drawn. All tokens are elements (e.g., $a_{(k)i}$) of monoid or group $A_k$ (where $a_{(k)i} \in A_k$) that are defined at place $p_k$ ($k = 1,2,\ldots m$). Different from ordinal object of category, each place does not always contain all elements of $A_k$. Moreover, duplication of the same elements such as '$a_{(k)i}, a_{(k)i}, a_{(k)i}$' in the same place $p_k$ is permissible. Arcs represent morphisms $f_{(k \to l)}$ that fire when there is at least one token '$a_{(k)i}$' on which morphism '$f_{(k \to l)}$' orders '$f_{(k \to l)}$:{$a_{(k)i}$}:' and is included in input place $p_k$. After firing, an image of $a_{(k)i}$ is outputted in an input place as a token $a_{(l)i}$ ($= a_{(k)i};f_{(k \to l)}$). We stipulate that any arc (morphism) has only one morphism (there are no multiple morphisms) that acts on one token (this is considered a 'state machine' in conventional PN). The index number $i$ of $a_{(k)i}$ ($i = 0,1,2,\ldots$) is the only optional mark that discriminates tokens. Hence, differently indexed tokens are equivalent; that is, '$a_{(k)i} = a_{(k)j} = a_{(k)3}$'. (b). After a firing $f_{(1 \to 2)}$ with order $a_{(1)2}$ in place $p_1$, token $a_{(1)2};f_{(1 \to 2)}$ ($= a_{(2)2}$) is the output in place $p_2$.

where '−' (minus) means the removal of tokens, and '+' (plus) the inclusion of tokens. If the token is 'c·$a_{(k)i}$', only an exchange between $a_{(k)i}$ and c·$a_{(k)i}$ enables the formula to be true because of the previous definition for tokens.

Next, a graph is produced when the compositionality between two morphisms and the existence of an identity morphism (meaning doing nothing) are omitted; therefore, if we regard the sequence of two firings '$f_{(k \to l)}$ and $f_{(l \to m)}$' under the condition for which the same token $a_{(k)i}$• as in the product of '$a_{(k)i};f_{(k \to l)};f_{(l \to m)} = a_{(m)i}$' is ordered, we may regard '$f_{(k \to m)}$' as the result of '$f_{(k \to m)} = f_{(k \to l)};f_{(l \to m)}$' with two firings of morphisms (transitions) $f_{(k \to l)}$ and $f_{(l \to m)}$ (here ';' denotes the diagrammatic order of operations; that is, the operation occurs in the order $f_{(k \to l)}$, then $f_{(l \to m)}$, and then $f_{(m \to n)}$). Furthermore, by ordering the same token $a_{(k)i}$, the associative law is confirmed as $(f_{(k \to l)};f_{(l \to m)});f_{m \to n)} = f_{(k \to l)};(f_{(l \to m)};f_{m \to n)})$ because '$\{a_{(k)i}\}(f_{(k \to l)};f_{(l \to m)}); f_{m \to n)} = \{a_{(k)i}\}f_{(k \to l)};(f_{(l \to m)};f_{m \to n)}) = a_{(n)i}$' (For later discussions, we label this comment **#remark 1**). The index '$i$' of $a_{(k)i}$, $a_{(m)i}$ or $a_{(n)i}$. . . is an optional mark and only means these tokens comprise a sequence of the same/unique flow as in '$a_{(k)i} \to a_{(m)i} \to a_{(n)i} \to$. . .', and '$i$' does not express a specifically ordered element within the group (or monoid) $A_k$ (in place $p_k$), $A_l$ (in place $p_l$) or $A_m$ (in place $p_m$). Moreover, there exists an identity morphism '$f_{(k)0}$' (that does nothing even to a token in place $p_k$) and '$f_{(l)0}$' (that does nothing even to a token in place $p_l$) that satisfies '$\{\}; f_{(k \to l)};f_{(l)0} (= f_{(l)0} {}^\circ f_{(k \to l)}) = f_{(k \to l)} = \{\}; f_{(k)0};f_{(k \to l)} (= f_{(k \to l)} {}^\circ f_{(k)0})$' for any morphism $f_{(k \to l)}$ (between places $p_k$ and $p_l$). In category theory, provided that $A_k$ is a domain, $A_l$ a codomain, and $f_{(k \to l)}$ a morphism such as $f_{(k \to l)}$: $A_k \to A_l$, the image of $A_k$ denoted $\{A_k\};f_{(k \to l)}$ satisfies

$$\{A_k\};f_{(k \to l)} \subseteq A_{l..} \tag{4}$$

We stipulate that $A_k$ is filled with one of the elements $a_{(k)i}$ (i = 0,1,2,. . .) ($\in A_k$). However, being different from $A_k$, place $p_k$ is a potential source object that may be filled with any element $a_{(k)i}$ ($\in A_k$) a multiple of times, $a_{(k)i}, a_{(k)i}$. . . or have no $a_{(k)j}$ ($\in A_k$). Likewise, $A_l$ is filled with one of the elements $a_{(l)i}$ (i = 0,1,2,. . .) ($\in A_l$), and place $p_l$ is a potential target object that may be filled with any element $a_{(l)i}$ ($A_l$) many times or not at all.

*Firing rule II*: *Firing defined within the same place (via an inner-place morphism).* Suppose that an element $f_{(k)i}$ is one of the morphisms (transitions) defined at place $p_k$ ($k$ is the place index, $k = 1,2,. . ., n,. . .$, and $n$ the number of places of interest in the PN), and $f_{(k)i}$ or c·$f_{(k)i}$ belongs to group $A_k$ ($f_{(k)i}$ or c·$f_{(k)i} \in$ group $A_k$). Additionally, a token $a_{(k)i}$• is one of the elements such as $f_{(k)i}$ of group $A_k$ or an accessorized element such as 'c·$f_{(k)i}$' (c a constant factor without any operational action). That is, '$a_{(k)i = f_{(k)i}}$' or '$a_{(k)i = }$c·$f_{(k)i}$' holds ($f_{(k)i}$ or c·$f_{(k)i} \in$ group $A_k$). There, we stipulate that a morphism $f_{(k)i}$ acts on a token $a_{(k)i}$• as

$$a_{(k)i} = c \cdot f_{(k)i} \text{ or in the simplest case } a_{(k)i} = f_{(k)i} \text{ with c} = 1. \tag{5}$$

Similar to the ordinal binary composition within a group $A_k$, the binary operation-like composition is '$\{a_{(k)i}\};f_{(k)j} (= f_{(k)j} {}^\circ a_{(k)i}) = f_{(k)i};f_{(k)j} (= f_{(k)j} {}^\circ f_{(k)i})$' ($\in$ group $A_k$) (where the content as an operation of token •'$a_{(k)i}$' is $f_{(k)i}$, which is denoted without •) or '$\{$c·$a_{(k)i}\};f_{(k)j} (= f_{(k)j} {}^\circ ($c·$a_{(k)i})) = \{$c·$f_{(k)i}\};f_{(k)j} (= f_{(k)j} {}^\circ ($c·$f_{(k)i})) =$ c·$\{f_{(k)i};f_{(k)j}\} (=$ c·$(f_{(k)j} {}^\circ f_{(k)i}))$' ($\in$ group $A_k$) (for which the token • is 'c·$a_{(k)}$'), as in Eq (5). Through firing a morphism (transition) $f_{(k)a}$ ordering a token $a_{(k)i}$• (= $f_{(k)i}$ or c·$f_{(k)i}$), a token $a_{(k)i}$ is removed from place $p_k$; after that, $a_{(k)h}$ (= $a_{(k)i};f_{(k)a}$, that is, the images of $a_{(k)i}$ by $f_{(k)a}$ ($\in A_k$)) are produced in the same place $p_k$, in accordance with 'morphism $f_{(k)a}$: $a_{(k)i} \to a_{(k)h}$ (= $f_{(k)h}$ or c·$f_{(k)h}$).

[Fig 3(A)]

We call $f_{(k)a}$, $f_{(k)l}$, and $f_{(k)j}$ 'inner-place morphisms'. Although $f_{(k)a}$ and $a_{(k)i}$ (= $f_{(k)i}$ or c·$f_{(k)i}$) are elements belonging to the respective unique sets $A_k = \{f_{(k)1}, f_{(k)2},. . .,f_{(k)i},. . .\}$ and = $\{$c·$f_{(k)1}$, c·$f_{(k)2},. . .,$ c·$f_{(k)i},. . .\}$, $a_{(k)i}$ is accompanied by a token •, and $f_{(k)a}$ is not. Additionally, between

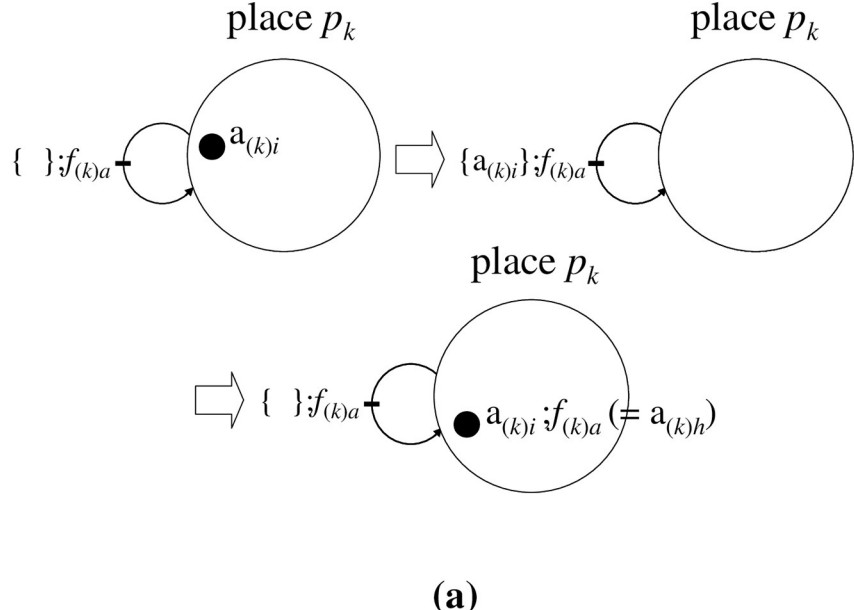

**(a)**

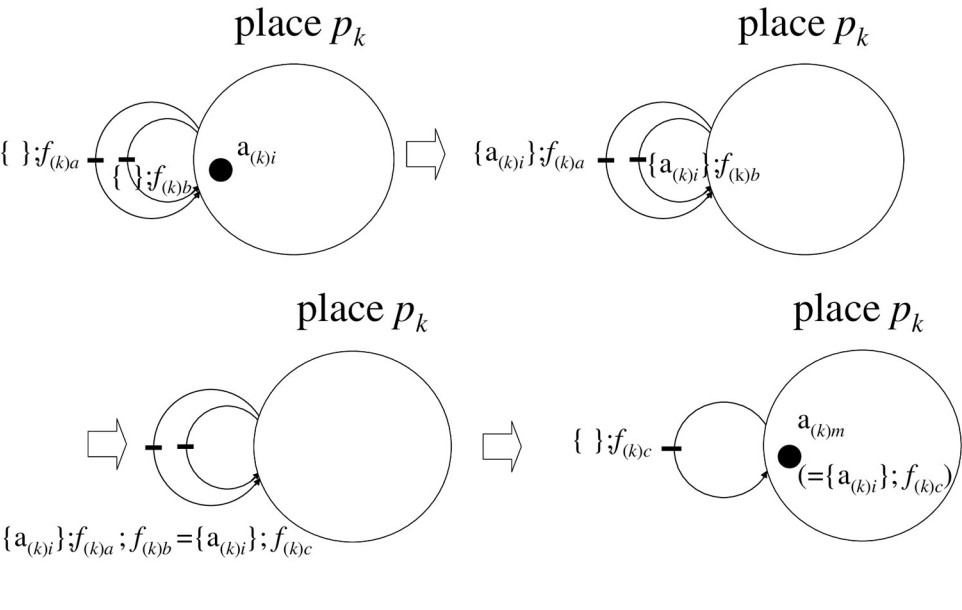

**(b)**

**Fig 3. Illustrations of operations on extended Petri nets (EPNs).** (a). In the present model, we mainly consider cases where a token $a_{(k)i}$ is also an operator such as $f_{(k)j}$ or $c \cdot f_{(k)j}$ ($f_{(k)j}$ being accessorized with constant c). Hence, through a firing of a morphism (transition) $f_{(k)a}$ orders a token $a_{(k)i}\bullet$ (= $f_{(k)j}$ or $c \cdot f_{(k)j}$), denoted by $f_{(k)a}$:{$a_{(k)i}$}, $f_{(k)a}$:c{$a_{(k)i}$} may become a composition like $f_{(k)a}$;$f_{(k)j}$ or $f_{(k)a}$;$c \cdot f_{(k)j}$ that also has one of each operator ($\in A_k$). A token $a_{(k)i}$ is at once removed from place $p_k$; subsequently, $a_{(k)h}$ (= $a_{(k)i}$;$f_{(k)a}$; that is, the image of $a_{(k)i}$ by $f_{(k)a}$ ($\in A_k$)) is produced in the same place $p_k$, in accordance with 'morphism $f_{(k)a}$. $a_{(k)i} \rightarrow a_{(k)h}$ (= $f_{(k)h}$ or $c' \cdot f_{(k)h}$). Note ';' means in diagrammatic order. A simple example is given in the relationship between DNA ($D_{(k)j}$) and its operator $B_{(k)(0\rightarrow j)}$ where '$D_{(k)j} = D_0$;$B_{(k)(0\rightarrow j)}$'; see main text for details. Although $f_{(k)a}$ and $a_{(k)i}$ (= $f_{(k)i}$ or $c \cdot f_{(k)i}$) are elements belonging to the respective unique set $A_k$ = {$f_{(k)1}, f_{(k)2},\ldots f_{(k)i},\ldots$} or = {$c \cdot f_{(k)1}, c \cdot f_{(k)2}\ldots, c \cdot f_{(k)i},\ldots$}, $a_{(k)i}$ is accompanied by token $\bullet$, and $f_{(k)a}$ is not. This

procedure expresses the individual operation on the token. (b). Provided a unique token $a_{(k)i}$ in place $p_k$ is ordered for each operation, a binary composition (operation) between morphisms $f_{(k)a}$ and $f_{(k)b}$ (both$\in A_k$) is performable, $\{a_{(k)i}\}$: $f_{(k)a};f_{(k)b} = \{a_{(k)i}\};f_{(k)c}$ when $f_{(k)a};f_{(k)b} = f_{(k)c}$ holds. In the former intermediate-state procedure, a transition (morphism) of $f_{(k)a}$ and $f_{(k)b}$ does not fire. In the latter intermediate-state procedure, a token $a_{(k)m}$ ($= a_{(k)i};f_{(k)c}$) is produced in the same place $p_k$ when $a_{(k)m}$ is the product of $a_{(k)i};f_{(k)c}$. We regard the entire procedure illustrated in this figure as a one-time firing of morphism $f_{(k)c}$ ($= f_{(k)a};f_{(k)b}$) that occurs by ordering $a_{(k)i}$. Moreover, a sequence of these procedures (the former and latter intermediate-state procedures) defines the binary composition between an arbitrary pair of morphisms that belongs to the same group (or related concepts) $A_k$ and place $p_k$.

morphisms $f_{(k)a}$ and $f_{(k)b}$ (both$\in A_k$), a binary composition (operation) is performable, i.e., '$f_{(k)a};f_{(k)b} = f_{(k)c}$.' An example is illustrated in a later section. [Fig 3(B)].

In general, the operation ';' is not commutative and may involve different operations depending on place $p_k$ on which it acts. Briefly, the firing rules for $f_{(k)a}$ on $a_{(k)i}$ in place $p_k$ are presented as follows:

$M_{w+1}(p_k) = M_w(p_k) + \{a_{(k)i}\};f_{(k)a} − a_{(k)i}$. Token $a_{(k)i}$ is transformed into $\{a_{(k)i}\};f_{(k)a}$ ($= f_{(k)a}°a_{(k)i}$) in the same place $p_k$ (substitution of object) at the same instant; otherwise,

$$M_{w+1}(p_k) = M_w(p_k). \tag{6}$$

Furthermore, by ordering the same token $a_{(k)i}$, the associative law holds, '$(f_{(k)a};f_{(k)b});f_{(k)d} = f_{(k)a};$ $(f_{(k)b};f_{(k)d})$' in place $p_k$ if the following relationship '$a_{(k)i}$};(f_{(k)a};f_{(k)b});f_{(k)d} = \{a_{(k)i}\};f_{(k)a};(f_{(k)b};f_{(k)d})$ $= a_{(k)h}$ ($\in$ group $A_k$)' is postulated. Together with this relation, there exists an identity morphism '$f_{(k)0}$' that satisfies '$f_{(k)a};f_{(k)0}$ ($= f_{(k)0}°f_{(k)a}$) $= f_{(k)a} = f_{(k)0};f_{(k)a}$ ($= f_{(k)a}°f_{(k)0}$)' for any morphism $f_{(k)a}$ within place $p_k$.

Moreover, we regard

$$a_{(k)i};f_{(k)a} \text{ or } f_{(k)a}°a_{(k)i} (= a_{(k)h}) \tag{7}$$

as the 'image of $a_{(k)i}$ by $f_{(k)a}$'; otherwise, '$a_{(k)i};f_{(k)a}$:' or '$f_{(k)a}°\{a_{(k)i}\}$:' means the 'map of morphism $f_{(k)a}$ on $a_{(k)i}$'. The former expresses a token, whereas the later displays the map of a morphism (operation).

Likewise, whereas '$\{a_{(k)i}\};f_{(k\to l)}$' ($= f_{(k\to l)}°a_{(k)i}$) expresses a token $a_{(l)i}$, '$\{a_{(k)i}\};f_{(k\to l)}$:' ($= f_{(k\to l)}°a_{(k)i}$:) displays a genuine map of a morphism $f_{(k\to l)}$ to $a_{(k)i}$. The symbol '$i$' of $a_{(k)i}$ and $a_{(l)i}$ means a relationship in which one is the image of the other in certain cases. However, this rule is optional and has no specific purpose. For example, with '$\{a_{(k)i}\};f_{(k)a} = a_{(k)h}$' of Eq (7), the relationship between '$i$' of '$a_{(k)i}$' and '$h$' of '$a_{(k)h}$' is unspecified. Then, associative law-like relationships hold when combining Firing Rules I and II under the condition that a unique '$a_{(k)i}$' is ordered as an initial token for the mapping of the operations; i.e., among $f_{(k)a}, f_{(k)b}, f_{(k\to l)}$, $f_{(l\to m)}, \cdots$,

$$\{a_{(k)i}\}; (f_{(k)a};f_{(k\to l)});f_{(l\to m)} = \{a_{(k)i}\};f_{(k)}a; (f_{(k\to l)};f_{(l\to m)}) = a_{(m)r} (\in \text{group } A_m). \tag{8}$$

In fact, the meaning of the composited map $f_{(k)a};f_{(k\to l)}$ seems unclear because the morphisms $f_{(k)a}$ and $f_{(k\to l)}$ are defined at different locations (the former is defined in the same place, whereas the latter is defined for different places), and an integrated morphism $f_{(k)a};f_{(k\to l)}$ seems to accomplish nothing other than a sequence of morphisms of $f_{(k)a}$ and $f_{(k\to l)}$. Hence, Eq (8) may better be regarded only for combinations of Firing Rules I and II because they do not break the category postulates, albeit there may be no qualitative meaning.

Similarly, the following relation may arise,

$$\{a_{(k)}i\}; (f_{(k)a};f_{(k)b});f_{(k\to m)} = \{a_{(k)}i\};f_{(k)a}; (f_{(k)b};f_{(k\to l)}) = a_{(l)s} (\in \text{group } A_l), \tag{9}$$

($r$, $s$; appropriate non-positive integers that provide adequate elements). There, the composite $f_{(k)a};f_{(k)b}$ may be integrated as a unique morphism $f_{(k)d}$ satisfying $f_{(k)d} = f_{(k)a};f_{(k)b}$ for which the monoidal operation at place $p_k$ is presumed. However, the combination of Firing Rules I and II $f_{(k)b};f_{(k\to m)}$ implies only a sequence of morphisms and this may only mean that this combination does not break the category postulates. Nonetheless, the commutativity-like law might be assumed for combinations of the two firing rules in the diagrammatically ordered display of morphisms such as for Eqs (8) and (9). Naturally, the inverse morphism may be included in Eq (8) or (9) if the appropriate definition of the inverse morphism is possible.

In this EPN model, we consider instances for which the compositionality among all target objects and source objects that are necessary for the construction of the category completely satisfies Eq (4). Therefore, under this assumption, the combination of Firing Rules I and II seems to be permissible without breaking the category postulates. At this moment, we must confirm that there are two type of morphisms in our model: 1) morphisms for which the **Firing Rules I and II** are always assumed to act on any single token; and 2) morphisms that act on two tokens (e.g., $f_{Int}\{,\}$, $f_{spl}\{,\}$, a topic that we shall return to later) for which the group or monoid postulates are not always satisfied.

In general, suppose that $C_k$ ($k = 1,2,\dots,m$) is an object of Category C, and object $C_k$ is unique; then C is termed a 'monoid'. In particular, for Firing Rule II, a pair of fireable morphisms $f_{(k)a}$ and $f_{(k)b}$ (a morphism is fireable if at least one token, say $a_{(k)i}$, for which index number '$k$' is the same as that included in place $p_k$) is viewed as a composition of two morphisms that are monoidal operations. If any morphism has an inverse, the associated object becomes a group. In our model, the individual 'place $p_k$' appears as a substitution of the object in the category to describe an individually specific state (marking) of an EPN. For simplicity, we assume instances of places, in which inverse morphisms $f_{(k\to l)}^{-1}$ exist with no regard to whether '$k \neq l$' holds.

**§2-3-3. Marking of a PN as an optional display.**   As an optional display, the marking of a PN at a certain moment (at session $w$, $w = 0,1,2\dots$) may be written as a direct sum of certain places,

$$S_w = \{a_{(1)2},\ a_{(1)3},\dots\}p_1 \oplus \{\dots, a_{(1)3},\dots\}p_2 \oplus \{\phi\}p_3 \oplus \dots$$
$$\oplus \{\dots, a_{(k)i},\ a_{(k)j},\dots\}p_k \oplus \dots \oplus \{a_{(n)2},\ a_{(n)5},\dots\}p_m. \tag{10}$$

The drawback that "PNs are monoids" has been reported by Meseguer et al. [46], although in their report, the tokens are ordinals and there is no difference between the respective tokens.

Additionally, in our model, the duplication and absence of tokens (elements) are permissible as $\{a_{(k)2}, a_{(k)2}, a_{(k)5},\dots\}$ in place $p_k$. There may also be instances in which no element is included in place $p_k$; they are displayed as $\{\phi\}$ in any place $p_k$. To be precise, $\phi$ is not a token; however, there may be instances for which $\phi$ is written as if it was a token,

$$\{\phi\}; f_{(\phi \to k)} = a_{(k)0} \text{ (in place } p_k), \tag{11}$$

which we describe below.

Next, we turn our attention to places that are themselves $m$-tuples ($m$ a fixed positive integer) and are sufficient to describe and illustrate an EPN model at a certain stage. The number of places ($m$) may be finite; nonetheless, occasions may arise needing $\infty$-tuple places for compositions of the model. With a broadened formalism, the integration of both may be possible, and the following relaxed rules may be effective for a marking $S_w$. Suppose that $S_w$ is accompanied by at least a very large number of places after the $m$-th place $p_m$, where no token is included (meaning that place $p_k$ includes $\phi$ for '$k \geq m+1$'); for this scenario, there may exist a multiplicity of settings for the position of symbol '$||$' because the determination does not

require a unique number '$m$' that indicates place $p_m$ (empty places like $\{\phi\}$ may be allowed before place $p_{m+1}$). That is,

$$S_w = \{a_{(1)2}, \ a_{(1)3}, \ldots\}p_1 \oplus \{\ldots, a_{(1)3}, \ldots\}p_2 \oplus \{\phi\}p_3 \oplus \ldots \oplus \{\ldots, \ a_{(k)i}, \ a_{(k)j}, \ldots\}p_k \oplus \ldots$$
$$\oplus \{a_{(n)2}, \ a_{(n)5}, \ldots\}p_m || \oplus \{\phi\}p_{m+1} \oplus \{\phi\}p_{m+2} \oplus \ldots . \tag{12}$$

Note that symbol '$||$' is an optional marking signifying that there are no tokens in places to the right of '$||$'. For, if an inclusion of $a_{(m+2)i}$ occurs in place $p_{m+2}$, the position of '$||$' must be moved to that between places $(m+2)$-th and $(m+3)$-th, meaning there are no tokens in places to the right of '$||$'. Up to this point, we assumed that a marking $S_w$ means the latter type in Eq (12) without loss of generality. Its omission may also be possible because if there is no interaction between the focused/recognized places $\{\phi\}p$, then there may be no contradiction within the model. We envisage that with this definition, the EPN formalism may gain from a more general description of a marking so that a certain infinite sequence of markings $S_w$ may be included in a unique set S as an element; if not, that means that a set S may include any sort of markings $S_{w'}$, $S_{w''}$, $S_{w'''}$ (having different definitions of groups $A_k$ at the associated place) all over the world. The set S may be expressed as

$$S = \{S_w | S_w \in \bigcup_{k=1}^{\infty} \oplus \{ \ \}p_k\}. \tag{13}$$

Although an infinite set of markings $S_w$ may belong to set S, there is a limit for the present EPN model; the compositionality for objects, for which individual tokens can transform smoothly from one place $p_k$ to another place $p_l$ ($k \neq l$ or $k = l$), needs some rules for the definition of the respective group (or related product with inverse) $A_k$ at place $p_k$. For example, we consider the following scenario. Let $S_1$ and $S_2$ have finite places,

$S_1 = \{a_{(1)2}, a_{(1)3}\}p_1 \oplus \{a_{(2)3}\}p_2 \oplus \{\phi\}p_3$,

$S_2 = \{a_{(1)6}\}p_1 \oplus \{a_{(2)4}\}p_2 \oplus \{a_{(3)5}\}p_3 \oplus \ldots \oplus \{a_{(10)7}\}p_{10} \oplus \{a_{(11)2}\}p_{11}$. The marking $S_1$ and $S_2$ are not regarded as belonging to the same set S that has an element with an $m$-tuple of markings. However, if we consider the above adjustment as in the following,

$$S_1 = \{a_{(1)2}, a_{(1)3}\}p_1 \oplus \{a_{(2)3}\}p_2 \oplus \{\phi\}p_3 \oplus \ldots \oplus \{\phi\}p_{10} \oplus \{\phi\}p_{11}|| \oplus \{\phi\}p_{12} \oplus \{\phi\}p_{13} \oplus \ldots,$$
$$S_2 = \{a_{(1)6}\}p_1 \oplus \{a_{(2)4}\}p_2 \oplus \{a_{(3)5}\}p_3 \oplus \ldots \oplus \{a_{(10)7}\}p_{10} \oplus \{a_{(11)2}\}p_{11}|| \oplus \{\phi\}p_{12} \oplus \{\phi\}p_{13} \oplus \ldots, \tag{14}$$

the latter expressions of $S_1$ and $S_2$ may belong to the set S given in Eq (13) with '$m = 11$ (or $m \geq 11$)' given in Eq (12).

For an identity marking, any marking $S_0$ can be considered such as

$$S_0 = \{\phi\}p_1 \oplus \{\phi\}p_2 \oplus \ldots \oplus \{\phi\}p_k \oplus \ldots \oplus \{\phi\}p_m || \oplus \{\phi\}p_{m+1} \oplus \{\phi\}p_{m+2} \oplus \ldots = f_0 \tag{15}$$

(an identity marking is equivalent to an identity morphism that performs nothing on any token).

We assume that an identity morphism $f_0$ can be used in any operation, marking where there is no change. Conversely, it is possible that an infinite number of firings of a morphism $f_0$ may occur any time between any two places. In this regard, we focus on instances where the number of firings of morphism (transition) $f_0$ is recognized.

The following relation can be confirmed straightforwardly,

$$\{S_! = \{a_{(1)0}\}p_1 \oplus \{a_{(2)0}\}p_2 \oplus \ldots \oplus \{a_{(k)0}\}p_k \oplus \ldots \oplus \{a_{(n)0}\}p_m || \oplus \{\phi\}p_{m+1} \oplus \{\phi\}p_{m+2} \oplus, \tag{16}$$

$$\neq S_0 (\text{and } f_0). \tag{17}$$

In other words, $S_0$ in Eq (15) is the combination of an initial object ($\ldots \{\phi\}p_k \oplus \ldots$) of a category

$A_k$, defined at each place $p_k$ by viewing the group (or monoid) $A_k$ as category $A_k$. In contrast, $S_!$ in Eq (16) is the combination of the terminal object $(\ldots\{a_{(k)0}\}p_k\oplus\ldots)$ of a category $A_k$, defined at each place $p_k$ by viewing the group (or monoid) $A_k$ as a category $A_k$. $S_! \neq S_0$ $(= f_0)$ as given by Eq (17) means the zero object where the initial object and the terminal object are equivalent (that is, $S_! = S_0$) does not exist in our model except for specific cases.

Hence, together, if two elements '$a_{(k)r}$ and $a_{(k)s}$ $(\in A_k)$' are further included in place $p_k$ in the above marking $S_w$, we stipulate that the result (the inclusion of tokens at place $p_k$) by '$S_1 = a_{(k)r}$ (+) $a_{(k)s}$' to $S_w$ is expressible as $S_w \oplus S_1 = \{a_{(1)2}, a_{(1)3}, \ldots\}p_1 \oplus \{\ldots, a_{(1)3}, \ldots\}p_2 \oplus \{\phi\}p_3 \oplus \ldots \oplus \{\mathbf{a_{(k)r}}, \mathbf{a_{(k)s}}, \ldots, a_{(k)i}, a_{(k)j}, \ldots\}p_k \oplus \ldots \oplus \{a_{(n)2}, a_{(n)5}, \ldots\}p_n \oplus \ldots$, where $S_w$ or $S_1$ is a marking that disregards the order of the places. However, this operation of inclusion may have an operational meaning as a monoidal addition; an arbitrary marking $S_w$ can be interpreted as the result of

$$S_0 \oplus S_{(0 \to w)} = S_w. \tag{18}$$

$S_0$ is then an identity operation of the monoid and therefore an identity morphism $f_0$. Meseguer et al. [46] reported that the PNs are monoids for which a monoidal operation is a simplistic inclusion of tokens. In addition, a zero transition (an identity transition) was also introduced.

Herein, we discriminate between the meanings of the monoidal additions. In the previous description, there are two types of monoidal addition for the present EPN model: type 1) the composition of morphisms such as '$f_{(k \to l)}; f_{(l \to m)} = f_{(k \to m)}$' where the monoidal addition is denoted by ';', and type 2) the inclusion/integration of tokens in the same place $p_k$ $(k = 1,2,\ldots)$ such as '$\ldots \oplus \{\ldots, a_{(k)i}, a_{(k)j}, \ldots\}p_k \oplus \ldots$'. We believe types 1) and 2) are also effective in the present EPN model. In this regard, from the viewpoint of abstract algebra, type 2) is trivial whereas type 1) is more interesting and has more potential for development in future studies because considerable topics of category theory are considered applicable in the form required of type 1). However, details are unclear at this stage.

As for the potential automatic treatments or records of changes in markings for an EPN, we express a sequence of firings as an ordered product (a diagrammatic order of morphisms) such as '$[M_1|\{a_{(k)i}\}; f_{(k)a} > f_{(k \to l)} > f_{(l \to m)} \cdot \ldots > = M_w$'; here, the initial marking is $M_1$, and the firing of $f_{(k)a} \ldots$ ordering $a_{(k)i}$ (in place $p_k$) occurs and produces $a_{(k)j}$ (in place $p_k$). After that, a firing of $f_{(k \to l)}$ ordering $a_{(k)j}$ occurs and produces $a_{(l)r}$ (in place $p_l$). Then, similarly, a firing of $f_{(l \to m)}$ ordering $a_{(l)r}$ occurs and gives $a_{(m)s}$ (in place $p_m$). In summary, the token $a_{(k)i}$ in place $p_k$ of $M_1$ is sent to $a_{(m)s}$ in place $p_m$ of $M_w$.

## §2–4. Annihilation/Creation morphism

To broaden the applicability of the model, we introduce an optional rule that has unique characteristics—annihilation and creation of elements; the symbol $\phi$ is adopted for this concept.

**§2-4-1. Annihilation morphism (transition).** We define the 'annihilation morphism (transition) $f_{(k \to \phi)}$' (also denoted by '$\{a_{(k)0}\}; f_{(k \to \phi)}$:' or '$f_{(k \to \phi)} \,^\circ \{a_{(k)0}\}$:' to mean the operation $f_{(k \to \phi)}$ ordering '$a_{(k)0}$') to be fireable when at least an explicit identity element $a_{(k)0}$ of set $A_k$ is included in place $p_k$. With regard to being explicit or implicit, an explicit identity is an ordinal identity $a_{(k)0}$ of a set (e.g., monoid, group) $A_k$. In contrast, we regard an implicit identity as doing nothing or being empty $\phi$; in category theory, $a_{(k)0}$ is an element of a terminal object, and $\phi$ (the empty category) is a first object (however, the empty set $\phi$ itself has no morphism). If we draw an arrow $f_{(0 \to \phi)}$ from the initial object to the terminal object or an arrow $f_{(\phi \to 0)}$ $(= f_{(0 \to \phi)}^{-1}$; the inverse arrow of $f_{(0 \to \phi)})$, there is a uniqueness between the initial object to the terminal object. In addition, both objects have uniqueness with respect to any arbitrary object.

Thus, we establish a unique relationship between $a_{(k)0}$ and $\phi$ as '$a_{(k)0} \leftrightarrow \phi$' via an annihilation/creation morphism.

An identity token $a_{(k)0}$ is needed in the appropriate place $p_k$ (there may be multiple identity tokens in the same place $p_k$) and similarly a terminal object in place $p_k$. However, we shall not use this concept for the moment. Instead, the initial place $p_\phi$, which includes $\phi$ (uncountable), is assumed to exist somewhere uniquely in all EPNs because the unique $\phi$ has no index such as '$k$' of $a_{(k)0}$. Moreover, $\phi$ may be found in any arbitrary place $p_k$ an infinite number of times. Therefore, we may perform the following conversion '$a_{(k)0} \leftrightarrow \phi$' freely in place $p_k$ via the following annihilation/creation morphisms.

After the firing $\{a_{(k)0}\}; f_{(k \to \phi)}$ ($= f_{(k \to \phi)} \circ a_{(k)0}$), meaning the image of $\{\}; f_{(k \to \phi)}$ ($= f_{(k \to \phi)} \circ \{\}$) ordering $a_{(k)0}$, an explicit token $a_{(k)0}$ is removed from the input place $p_k$, and $\phi$ ($= \{a_{(k)0}\}; f_{(k \to \phi)} = f_{(k \to \phi)} \circ a_{(k)0}$) is produced in the output place $p_\phi$.

At this moment, we stipulate that an annihilation morphism $f_{(k \to \phi)}$ is removed from any arbitrary place $p_k$ to a unique place $p_\phi$ with a one-tuple arc, for which the following annihilation of an explicit identity token $a_{(k)0}$ is deleted, specifically, morphism $f_{(k \to \phi)}$: $a_{(k)0} \to \phi$. Alternatively, $\{a_{(k)0}\}; f_{(k \to \phi)}$ ($= f_{(k \to \phi)} \circ \{a_{(k)0}\}$) $= \phi$ (in place $p_\phi$). In fact, no explicit token is produced in place $p_\phi$ because the emergence of $\phi$ means no token, and this procedure simply means the change '$a_{(k)0} \to \phi$' or the mere deletion of $a_{(k)0}$ in place $p_k$. That is, in place $p_\phi$, it is always necessary to express the above annihilation procedure because it suffices to consider that the input and output places of the annihilation morphism $f_{(k \to \phi)}$ are defined at the same place $p_k$.

In considering an operational meaning for the morphism $f_{(k \to \phi)}$ to marking $S_1$ expressed as a direct sum, the following change is possible; for place $p_k$ of marking $S_1$, '$f_{(k \to \phi)}$:$\{-a_{(k)0}\}$ $p_k + \{+\phi\} p_\phi$' acts on marking '$S_1 = \{a_{(k)0}\} p_k + \{\phi\} p_\phi$' giving

$$S_1; f_{(k \to \phi)}(= f_{(k \to \phi)} \circ S_1) = \{a_{(k)0} - a_{(k)0}\} p_k + \{\phi + \phi\} p_\phi = \{\phi\} p_k + \{\phi\} p_\phi. \tag{19}$$

Note that the meaning of '$\circ$' between '$\{\}; f$ (to a token such as $a_{(k)i}$) and $\{\}; f$ (to a marking such as $S_1$) is slightly different depending on the context (as, for example, in using the composite of the ordinal morphism $f_{(k \to l)}$, morphism $f_{(k)a}$ or a combination of multiple morphisms on marking $S_1$ as in Eq (19)). In Eq (19), '$\circ$' expresses the multiple effects of $f_{(k \to \phi)}$, which means a change between two pre-/post-firings of markings of an annihilation morphism.

Together with this, in Eq (19), we stipulate that whereas $a_{(k)0}$ is an explicit identity of set $A_k$, $\phi$ is an implicit identity signifying that place $p_k$ is empty. Naturally, $\phi$ may be regarded as belonging to any group $A_k$ ($k = 1,2,\ldots$) implicitly.

**§2-4-2. Creation morphism (transition).** Similarly, we introduce the 'creation morphism (transition) $f_{(\phi \to k)}$' (also denoted by '$\{\phi\}; f_{(\phi \to k)}$: or $f_{(\phi \to k)} \circ \{\phi\}$:' meaning the operation $f_{(\phi \to k)}$ ordering '$\phi$') that is fireable when $\phi$ is included in place $p_k$ (in this regard, this means any $f_{(\phi \to k)}$ is freely fireable at any time).

After the firing of a morphism '$\{\phi\}; f_{(\phi \to k)}$ (i.e., the image of $\{\}; f_{(\phi \to k)}$ ($= f_{(\phi \to k)} \circ \{\}$) ordering $\phi$, is denoted by $\{\phi\}; f_{(\phi \to k)}$ or the image of $f_{(\phi \to k)} \circ \{\phi\}$), an implicit token $\phi$ (empty) is removed from place $p_\phi$ and an explicit $a_{(k)0}$ is produced in place $p_k$.

At this moment, we stipulate that a creation morphism $f_{(\phi \to k)}$ is affiliated at all places with a one-tuple by an arc from a unique place $p_\phi$ to any arbitrary place $p_k$; that is, morphism $f_{(\phi \to k)}$: $\phi \to a_{(k)0}$ or alternatively, $\{\phi\}; f_{(\phi \to k)}$ ($= f_{(\phi \to k)} \circ \{\phi\}$) $= a_{(k)0}$.

[Fig 4(A)]

In fact, no explicit token would be removed in place $p_\phi$ because a deletion of $\phi$ means no change for a token, and this procedure simply means the change '$\phi \to a_{(k)0}$' or the insertion of $a_{(k)0}$ in place $p_k$. In addition, in place $p_\phi$, it is unnecessary to apply the above creation procedure

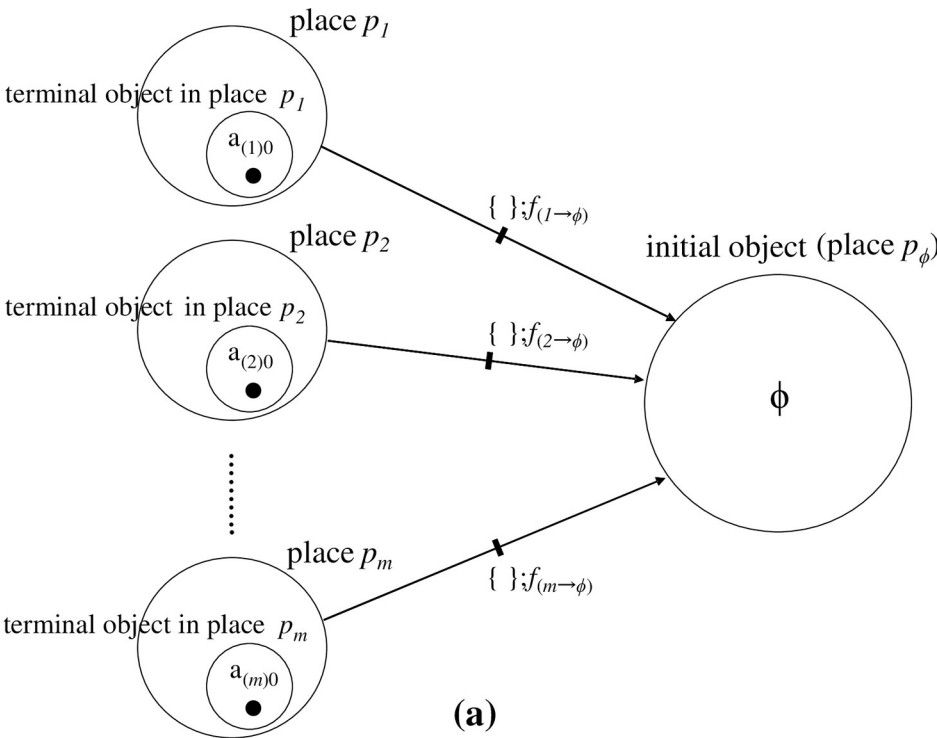

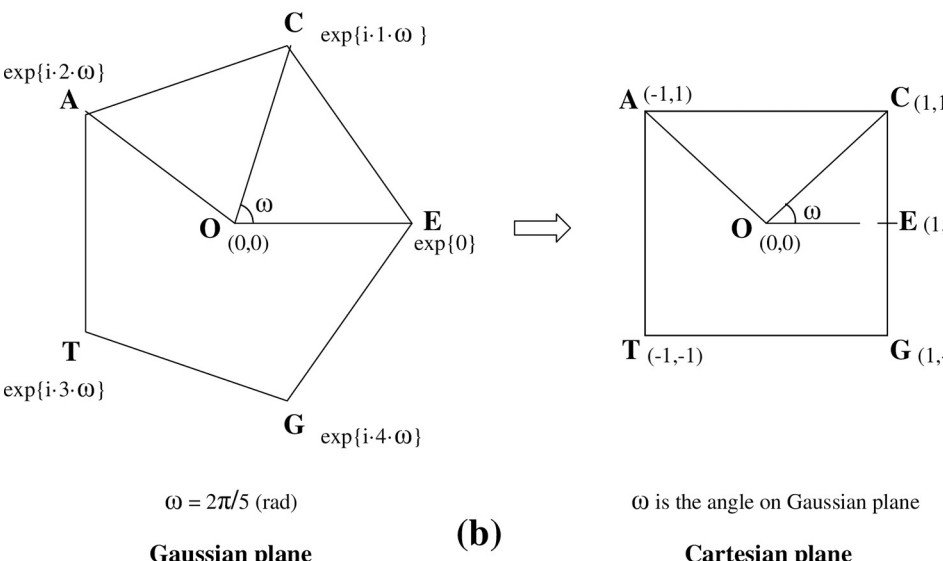

**Fig 4. Initial and terminal objects, and two types of phasor diagrams.** (a). For places $p_1$, $p_2$,..,$p_k$,...,$p_m$, there may exist 'terminal objects' in each place where the individual terminal object includes identity elements such as 'a$_{(1)0}$ for place $p_1$' and 'a$_{(2)0}$ for place $p_2$'. Moreover, there exists an initial object (initial place) $p_\phi$ that includes an empty token, which is denoted by $\phi$. If an identity token a$_{(k)0}$ is operated on by morphism $f_{(k\to\phi)}$, a$_{(k)0}$ is deleted from place $p_k$ and an empty token is generated in place $p_\phi$ (Note, a$_{(k)0}$ is annihilated from place $p_k$). Conversely, morphism $f_{(\phi\to k)}$ acts on $\phi$ creating a$_{(k)0}$ in place $p_k$ in accordance with $\{\phi\}$;$f_{(\phi\to k)} = $ a$_{(k)0}$. There is a uniqueness between '$f_{(\phi\to k)}$ and $f_{(k\to\phi)}$' whereby only a unique morphism is definable between a terminal object and an initial object. In this way, the annihilation/ creation morphism acts as an eraser/generator for a token. (b). In the bottom left panel (reproduced from **Fig 5** of

[Ref.6]), the five bases 'E, C, A, T/U, G' are represented on a phasor diagram (unit circle in the Gaussian plane) as five equi-spaced unit-modular phasors with inclusive angle '$2\pi/5$'. The general association of each base with a phasor is '$X_m \leftrightarrow \mathrm{Exp}(m\cdot\omega\cdot i)$' ('i' denotes the 'imaginary unit', $\omega = 2\pi/5$ (rad), and $m = \{0, 1, 2, 3, 4, 5\}$). To enhance the usability of this representation, we map the phasor diagram onto a square in the Cartesian plane (right panel) under the bijection E $(1,0) \leftrightarrow$ E $(= \mathrm{Exp}(0))$, C $(1,1) \leftrightarrow$ C $(= \mathrm{Exp}(\omega\cdot i))$, A $(-1,1) \leftrightarrow$ A $(= \mathrm{Exp}(2\cdot\omega\cdot i))$, T/U $(-1,-1) \leftrightarrow$ T/U $(= \mathrm{Exp}(3\cdot\omega\cdot i))$, G$(1, -1) \leftrightarrow$ G $(= \mathrm{Exp}(4\cdot\omega\cdot i))$ between two diagrams. The position of '$E_0$' should always be placed at the midpoint of the right side of the square.

because it suffices that the input and output places of a creation morphism $f_{(\phi\rightarrow k)}$ are defined at the same place $p_k$.

By considering an operational meaning of a morphism $f_{(\phi\rightarrow k)}$ to marking $S_2$, the following change is describable for place $p_k$ of marking $S_2$, '$f_{(k\rightarrow\phi)}:\{+a_{(k)0}\}p_k+\{-\phi\}p_\phi$' that act on marking '$S_2 = \{\phi\}p_k+\{\phi\}p_\phi$' as

$$S_2;f_{(\phi\rightarrow k)}(= f_{(\phi\rightarrow k)} \circ S_2) = \{\phi + a_{(k)0}\}p_k + \{\phi - \phi\}p_\phi = \{a_{(k)0}\}p_k + \{\phi\}p_\phi. \quad (20)$$

Here, $f_{(\phi\rightarrow k)}$ means the change between the pre-/post-firing of two markings of a creation morphism.

With the use of the annihilation/creation morphism, the interchange

$$a_{(k)0} \leftrightarrow \phi, \quad (21)$$

which implies reducing/increasing the number of tokens is performable almost freely.

A small novelty brought about by our idea is that an annihilation/creation morphism could be used as an eraser/generator of tokens; we demonstrate the idea below.

*Firing rule*: *Firing defined between place $p_k$ and place $p_\phi$ (via an annihilation morphism $f_{(k\rightarrow\phi)}$).* Suppose that the number of an identity token $a_{(k)0}$ in place $p_k$ ($k = 1,2,..$) or $p_\phi$ is given as $N_w(\{a_{(k)0}\}p_k)$ ($w$-th session, $w = 1,2,...$) or $N_w(\{a_{(k)0}\}p_\phi)$, respectively.

Transition (morphism) $f_{(k\rightarrow\phi)}$ is fireable $\Leftrightarrow N_w(\{a_{(k)0}\}p_k) \geq 1$;

$N_{w+1}(\{a_{(k)0}\}p_k) = N_w(\{a_{(k)0}\}p_k) - 1$: $p_k$ (input place of the annihilation morphism $f_{(k\rightarrow\phi)}$) ^ $p_\phi$ (output place of the annihilation morphism $f_{(k\rightarrow\phi)}$).

$N_{w+1}(\{a_{(k)0}\}p_\phi) = N_w(\{a_{(k)0}\}p_\phi)$: $p_\phi$ (output place of the annihilation morphism $f_{(k\rightarrow\phi)}$) ^ $p_k$ (input place of the annihilation morphism $f_{(k\rightarrow\phi)}$), i.e., $N(\{a_{(k)0}\}p_\phi) = N'(\{a_{(k)0}\}p_\phi) = 0$. In addition,

$N_{w+1}(\{a_{(k)0}\}p_k) = N_w(\{a_{(k)0}\}p_k)$, $N_{w+1}(\{a_{(k)0}\}p_\phi) = N_w(\{a_{(k)0}\}p_\phi)$: otherwise.

*Firing rule*: *Firing defined between place $p_k$ and place $p_\phi$ (via a creation morphism $f_{(\phi\rightarrow k)}$).* Transition (morphism) $f_{(\phi\rightarrow k)}$ is always fireable with

$N_{w+1}(\{a_{(k)0}\}p_k) = N_w(\{a_{(k)0}\}p_k) + 1$: $p_k$(output place of the creation morphism $f_{(\phi\rightarrow k)}$) ^ $p_\phi$ (input place of the creation morphism $f_{(\phi\rightarrow k)}$) and

$N_{w+1}(\{a_{(k)0}\}p_\phi) = N_w(\{a_{(k)0}\}p_\phi)$: $p_\phi$(input place of the creation morphism $f_{(\phi\rightarrow k)}$) ^ $p_k$ (output place of the creation morphism $f_{(\phi\rightarrow k)}$); i.e., $N(\{a_{(k)0}\}p_\phi) = N'(\{a_{(k)0}\}p_\phi) = 0$. In addition

$N_{w+1}(\{a_{(k)0}\}p_k) = N_w(\{a_{(k)0}\}p_k)$, $N_{w+1}(\{a_{(k)0}\}p_\phi) = N_w(\{a_{(k)0}\}p_\phi)$: otherwise.

The following relationships are straightforward to verify:

$N'(\{a_{(k)0}\}p_{all}) = N(\{a_{(k)0}\}p_{all}) - n_a + n_c$, ($N(\{a_{(k)0}\}p_{all})$: the number of the identity tokens in all places $p_1, p_2, p_3,..., p_m$; $n_a$: the number of the firing of the annihilation morphism, $n_c$: the number of the firing of the creation morphism) and

$M'(p_{all}) = M(p_{all}) - n_a + n_c$, ($M(p_{all})$: the number of tokens in all places $p_1, p_2, p_3,..., p_m$).

As seen above, the state of markings of an EPN may be describable in various forms as given for example in Eqs (3), (6), (11)–(14), and (18). In the present model, we take the common exhaustive approach that without loss of generality there exists a morphism $f_{(k\rightarrow l)}$ between any arbitrary pair of places that is potentially fireable, and for a morphism $f_{(k\rightarrow l)}$ that

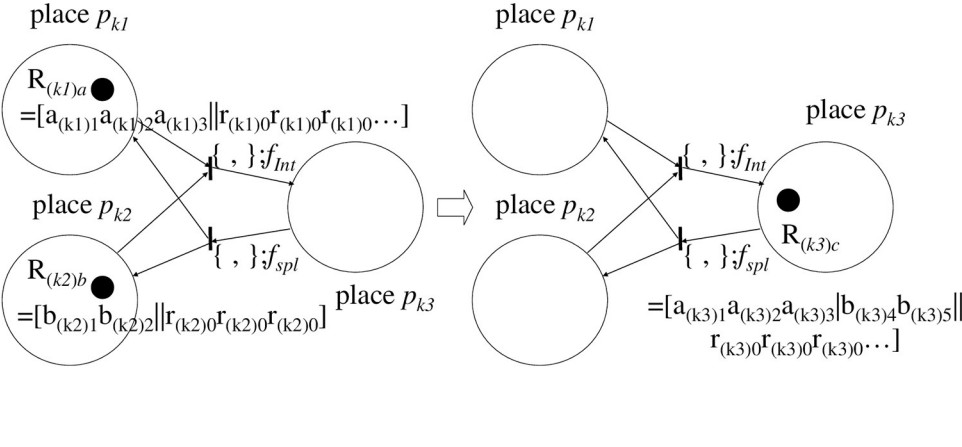

**(a)**

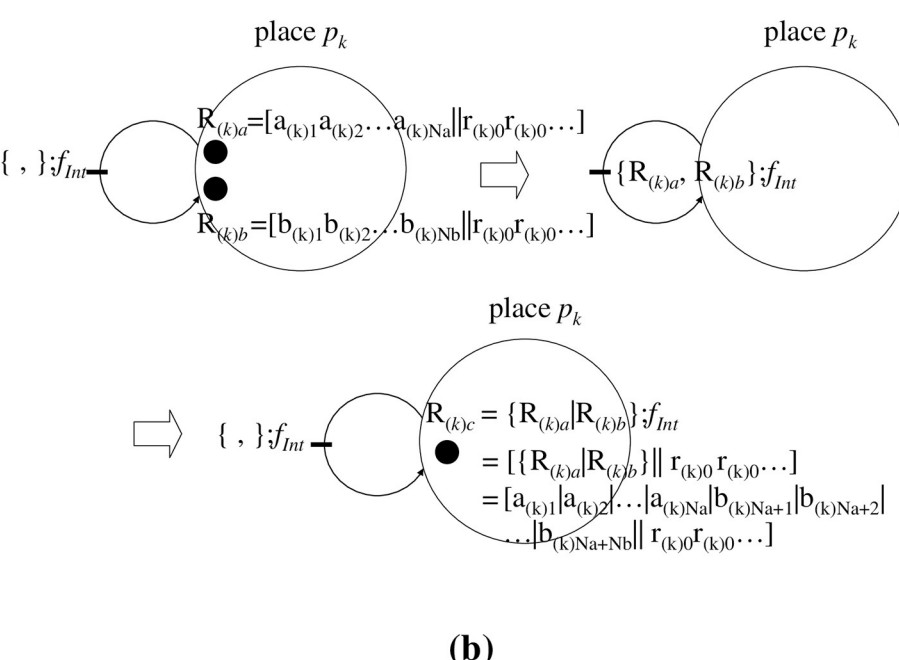

**(b)**

**Fig 5. Illustrations of integration using two tokens.** (a). Provided that token '● $R_{(k1)a} = [a_{(k1)1}|a_{(k1)2}|a_{(k1)3}||r_{(k1)0}r_{(k1)0}$ ...] = $[\{R_{(k1)a}\}||r_{(k1)0}r_{(k1)0}$ ...]' is included in place $p_{k1}$, and a token '●$R_{(k2)b} = [b_{(k2)1}|b_{(k2)2}||r_{(k2)0}\ r_{(k2)0}$ ...] = $[\{R_{(k2)b}\}||$ $r_{(k2)0}r_{(k2)0}$ ...]' is also in place $p_{k2}$, by a firing of morphism $f_{Int}\{,\}$ (= {|}; $f_{Int}$), an integrated state '● $R_{(k3)c}$' is produced in place $p_{k3}$; i.e., $R_{(k3)c} = f_{Int}\{R_{(k1)a}, R_{(k2)b}\} = \{R_{(k1)a}, R_{(k2)b}\} = \{R_{(k1)a}\}\cup\{R_{(k2)b}\} = \{R_{(k1)a}\}++\{R_{(k2)b}\} = \{R_{(k1)a}|R_{(k2)b}\} = [\{R_{(k3)a}|R_{(k3)b}\}||r_{(k3)0}\ r_{(k3)0}\ |\ .\ .\ .\ .\ .\ ]$ (in place $p_{k3}$). The inverse procedure is also illustrated. A firing of $f_{spl}\{,\}$ (= {//}) acting on token ●$R_{(k3)c}$ (in place $p_{k3}$), a splitting occurs of $R_{(k3)c}$ into two tokens $R_{(k1)a}$ (in place $p_{k1}$) and $R_{(k2)b}$ (in place $p_{k2}$), yielding '$f_{spl}\{R_{(k1)a}, R_{(k2)b}\}$ = '$\{[R_{(k1)a}]//[R_{(k2)b}]\} = \{R_{(k1)a}\}$ ((+)) $\{R_{(k2)b}\}$ ($R_{(k1)a}$ in place $p_{k1}$, $R_{(k2)b}$ in place $p_{k2}$)'. (b). A simple case of assuming 'place $p_k$ = place $p_{k''}$' is illustrated. After the firing of morphism $f_{Int}\{A, B\}$ (= {A|B}), which performs an integration of A● and B●, the integrated state '●$R_{(k3)c}$' is produced in place $p_{k3}$ as follows: $R_{(k)c}$ (in place $p_k$) = $f_{Int}\{R_{(k)a}, R_{(k)b}\} = \{R_{(k)a}, R_{(k)b}\} = \{R_{(k)a}\}\cup\{R_{(k)b}\} = \{R_{(k)a}\}++\{R_{(k)b}\} = \{R_{(k)a}|R_{(k)b}\}$ (both, in place $p_k$). = $[\{R_{(k)a}|R_{(k)b}\}||r_{(k3)0}\ r_{(k3)0}\ |\ .\ .\ .\ .\ .\ ] = [a_{(k)1}|a_{(k)2}|a_{(k)3}|b_{(k)1}|b_{(k)2}||r_{(k)0}\ r_{(k)0}$ ...] (in place $p_k$).

is inadequate (non-composable), it suffices that we consider the morphism (transition) $f_{(k \to l)}$ as being permanently non-fireable. In **§3**, to aid the reader's understanding, the specific use of the annihilation and creation morphisms is explained more explicitly.

## §3. An example of composition in the DNA model

### §3–1. A simplistic model to integrate/split tokens of DNA sequences

First, for integrating/splitting tokens, the following manipulation is considered provided that the two tokens

$R_{(k1)a} \bullet = [a_{(k1)1}|a_{(k1)2}|a_{(k1)3}|\ldots|a_{(k1)i}|\ldots|a_{(k1)(Na-1)}|a_{(k1)Na}|| r_{(k1)0\ (Na+1)}|r_{(k1)0\ (Na+2)}|\ldots]$ (in place $p_{k1}$) = $[\{R_{(k1)a}\}||r_{(k1)0\ (Na+1)}|r_{(k1)0\ (Na+2)}|\ldots]$ (in place $p_{k1}$) and

$R_{(k2)b} \bullet = [b_{(k2)1}|b_{(k2)2}|b_{(k2)3}|\ldots|b_{(k2)\mu}|\ldots|b_{(k2)(Nb-1)}|b_{(k2)Nb}|| r_{(k2)0\ Nb+1}|r_{(k2)0\ Nb+2}|\ldots]$ (in place $p_{k2}$) = $[\{R_{(k2)b}\}||r_{(k2)0\ Nb+1}|r_{(k2)0\ Nb+2}|\ldots\ldots]$ (in place $p_{k2}$)

are included independently; here, $N_a$ and $N_b$ are the least numbers from where there are no coordinates other than the identity coordinates such as $r_{(k)0}$; the display of operations determined for each coordinate is omitted. In general, the integrated (combined) state of tokens $A\bullet$ and $B\bullet$ is denoted

$$f_{Int}\{A, B\} = \{A|B\} = \{A\} \cup \{B\} = \{A\} + +\{B\} = [\{A|B\}||r_{(k)0}r_{(k)0}\ldots]$$
$$= [A|B||r_{(k)0}r_{(k)0}\ldots] = [C||r_{(k)0}r_{(k)0}\ldots](\text{in place } p_{k3}), \tag{22}$$

and displayed as another token $C\bullet$ ('++' means the gluing/combining of two neighboring tokens, A and B, with the non-commutative order of these tokens noted in the map $f_{Int}\{,\}$. Additionally, the use of '$||r_{(k)0}r_{(k)0}\ldots]$' needs delicate treatment). Note that, in the above description, the display of coordinates of tokens is essentially based on our previous article [6].

Hence, the integrated state of $R_{(k1)a}\bullet$ and $R_{(k2)b}\bullet$, denoted by $R_{(k3)c}\bullet$ is displayed as

$R_{(k3)c}\bullet = f_{Int}\{R_{(k1)a}, R_{(k2)b}\} = \{R_{(k1)a}\}\cup\{R_{(k2)b}\}$

$= \{R_{(k1)a}\}++\{R_{(k2)b}\} = \{R_{(k3)a}|R_{(k3)b}\}\bullet$ (in place $p_{k3}$)'.

$[\{R_{(k3)a}|R_{(k3)b}\}||r_{(k3)0\ Na+Nb+1}|r_{(k3)0\ Na+Nb+2}|\ldots\ldots]\bullet$ (in place $p_{k3}$)

$$= [a_{(k3)1}|a_{(k3)2}|a_{(k3)3}|\ldots|a_{(k3)i}|\ldots|a_{(k3)(Na-1)}|a_{(k3)Na}||b_{(k3)Na+1}|b_{(k3)Na+2}|b_{(k3)Na+3}|\ldots|b_{(k3)Na+\mu}|\ldots$$
$$|b_{(k3)(Na+Nb-1)}|b_{(k3)(Na+Nb)}||r_{(k3)0\ Na+Nb+1}|r_{(k3)0\ Na+Nb+2}|\ldots]\bullet(\text{in place } p_{k3}) \tag{23}$$

[see **Fig 5(A)**]. In this example, token $R_{(k3)c}\bullet$ is included in a different place $p_{k3}$.

Conversely, the splitting the state is denoted by

'$f_{spl}\{A, B\} = \{A//B\} = \{A\}\bullet$ (= $[A||r_{(k)0}r_{(k)0}\ldots]\bullet$ (in place $p_{k1}$)) ((+)) $\{B\}\bullet$ (= $[B||r_{(k)0}r_{(k)0}\ldots]$

$\bullet$ (in place $p_{k2}$))', where '(())' means that respective tokens are included in different places $p_{k1}$ and $p_{k2}$). (24)

In practice, for a token $R_{(k3)c}\bullet$ (= $\{R_{(k1)a}|R_{(k2)b}\}\bullet$), the following is true;

$f_{spl}\{R_{(k1)a}, R_{(k2)b}\}\bullet = \{R_{(k1)a}//R_{(k2)b}\} = \{R_{(k1)a}\}\bullet$ ((+)) $\{R_{(k2)b}\}\bullet$ (where $R_{(k1)a}$ is in place $p_{k1}$, $R_{(k2)b}$ is in place $p_{k2}$).

In detail,

'$R_{(k1)a}\bullet = [\{R_{(k1)a}\}||r_{(k1)0\ Na+1}|r_{(k1)0\ Na+2}|\ldots\ldots] = [a_{(k1)1}|a_{(k1)2}|a_{(k1)3}|\ldots|a_{(k1)i}|\ldots|a_{(k1)Na-1}|a_{(k1)Na}$

$|| r_{(k1)0\ Na+1}|r_{(k1)0\ Na+2}|\ldots]$ (in place $p_{k1}$)', and

$R_{(k2)b}\bullet = [\{R_{(k2)b}\}||r_{(k2)0\ Nb+1}|r_{(k2)0\ Nb+2}|\ldots\ldots] = [b_{(k2)1}|b_{(k2)2}|b_{(k2)3}|\ldots|b_{(k2)\mu}|\ldots|b_{(k2)Nb-1}|b_{(k2)Nb}$

$||r_{(k2)0\ Nb+1}|r_{(k2)0\ Nb+2}|\ldots]$ (in place $p_{k2}$) \hfill (25)

[see **Fig 5(A)**].

[**Fig 5(A)**] displays general cases, but here we assume conditioned cases for which places $p_{k1}$ and $p_{k2}$ are the same as place $p_k$ ($= p_{k1} = p_{k2} = p_{k3}$) to simplify the examples.

For tokens $R_{(k)a}\bullet$ and $R_{(k)b}\bullet$ (both, in place $\mathsf{p_k}$), the following relation can be verified;

$$R_{(k)c}\bullet(\text{in place } \mathsf{p_k}) = f_{\mathsf{Int}}\{R_{(k)a}, R_{(k)b}\} = \{R_{(k)a}\} \cup \{R_{(k)b}\}$$
$$= \{R_{(k)a}\} + +\{R_{(k)b}\} = \{R_{(k)a}|R_{(k)b}\}\bullet(\text{in place } \mathsf{p_k}), \tag{26}$$

$$[\{R_{(k)a}|R_{(k)b}\}||r_{(k)0 \ Na+Nb+1}|r_{(k)0 \ Na+Nb+2}|\ldots]\bullet(\text{in place } \mathsf{p_k}) \tag{27}$$

$$= [a_{(k)1}|a_{(k)2}|a_{(k)3}|\ldots|a_{(k)i}|\ldots|a_{(k)Na-1}|a_{(k)Na}|b_{(k)Na+1}|b_{(k)Na+2}|b_{(k)Na+3}|\ldots|b_{(k)Na+i}|\ldots|b_{(k)Na+Nb-1}$$
$$|b_{(k)Na+Nb}||r_{(k)0Na+Nb+1}|r_{(k)0Na+Nb+2}|\ldots]\bullet(\text{in place } \mathsf{p_k}), \tag{28}$$

[**Fig 5(B)**]

$$= \{A\}\bullet((+))\{B\}\bullet(\text{both, in place } \mathsf{p_k}) = \{A\}\bullet(+)\{B\}\bullet(\text{both, in place } \mathsf{p_k}). \tag{29}$$

For this example, the following is also confirmed; for token $R_{(k)c}\bullet$ ($= \{R_{(k)a}|R_{(k)b}\}\bullet$),

$$f_{spl}\{R_{(k)a}, R_{(k)b}\}\bullet = \{R_{(k)a}//R_{(k)b}\} = \{R_{(k)a}\}\bullet((+))\{R_{(k)b}\}\bullet(\text{both, in place} \mathsf{p_k}) = \{R_{(k)a}\}\bullet$$
$$(+)\{R_{(k)b}\}\bullet(\text{both, in place } \mathsf{p_k}). \tag{30}$$

In detail,

$$R_{(k)a}\bullet = [\{R_{(k)a}\}||r_{(k)0 \ Na+1}|r_{(k)0 \ Na+2}|\ldots\ldots] = [a_{(k)1}|a_{(k)2}|a_{(k)3}|\ldots|a_{(k)i}|\ldots|a_{(k)Na-1}|a_{(k)Na}$$
$$||r_{(k)0 \ Na+1}|r_{(k)0 \ Na+2}|\ldots](\text{in place } p_k), \tag{31}$$

and

$$R_{(k)b}\bullet = [\{R_{(k)b}\}||r_{(k)0 \ Nb+1}|r_{(k)0 \ Nb+2}|\ldots\ldots] = [b_{(k)1}|b_{(k)2}|b_{(k)3}|\ldots|b_{(k)\mu}|\ldots|b_{(k)Nb-1}|b_{(k)Nb}$$
$$||r_{(k)0Nb+1}|r_{(k)0Nb+2}|\ldots] (\text{in place } p_k). \tag{32}$$

We next consider rules for which the tokens are single-stranded sequences of DNA denoted by $D_{(k)j}\bullet$ and of RNA denoted by $R_{(k')j}\bullet$ (here, the indices mean the $j$-th session in place $p_{k'}$). They are associated with biological products with a mature messenger RNA produced from DNA. Each place is based on the sets composed of $D_{(k)j}$, $R_{(k')j}$, and related products. The morphism $f_{(k \to l)}$ has conditions pertaining to the enzyme and circumstances that act on $D_{(k)j}$ and consequently change $D_{(k)j}$ into another similar token $D_{(k)l}$ at the same or a different place. The essential stipulations are almost equivalent to the rules of **§3.**

In a previous article, we suggested the DNA wallpaper pattern for which the base units are placed in a cruciform pattern (**Fig 1** of [Ref. 6]; see S1 File). The pattern was introduced to develop a category-like model for genetics (in fact, molecular biology). However, in that article, the sequence of the single-strand DNA and related products were expressed marginally in a category-like formalism using a polygonal line in a plane. Because the shape of the polygonal line changed considerably, using it proved inconvenient. Therefore, we developed a more stable display.

In essence, in the canonical Watson–Crick DNA base pairing scheme, adenine (A) forms a base pair with thymine (T) and guanine (G) forms a base pair with cytosine (C) [47, 48]. Similarly, RNA is a molecule that has a much shorter chain of nucleotides but has various biological roles. The DNA sequence consisting of bases 'A, C, T, and G' is transcribed into RNA composed of bases 'A, C, U, and G'; the sets only differ in that 'U (uracil)' replaces 'T (thymine)'.

We introduced to these bases, a base 'E' to indicate an 'empty' base, which is treated in the same way for display purposes as an identity base. In the previous article, with five bases, we considered the set $C_5$ = {C, A, T, G, E}. In this example, the token $D_j\bullet$ is rewritten as $D_{(k)j}\bullet$.

## §3–2. A bijective expression of DNA sequences using a diagram-pole of five equi-spaced phasors and a square diagram-pole

First, we introduced a phasor diagram in which each base 'E, C, A, T/U, G' was assigned to one of five equi-spaced phasors located on the unit circle in the Gaussian plane, each with a '$2\pi/5$' angular phase separation to the next [**Fig 4(B),** left panel]; reproduced from **Fig 5** of [6]).

We then introduced phase angle '$\omega$' defined as the counterclockwise rotational angle '$\omega$ = $2\pi/5$ (rad)' and a binary composition of '$\omega$' denoted by '$\bullet$' written '$\omega_2\bullet\omega_4 = \omega_1$', and an action on bases also denoted by $\circ$. A rotational group W is defined; W = {$\omega_0$, $\omega_1$, $\omega_2$, $\omega_3$, $\omega_4$}, with the rule of '$\omega_m = \omega^m$' and '$m$ = {0, 1, 2, 3, 4, 5}' ($\omega_0$ is the identity meaning no rotation). The bijection rule; $X_0 \leftrightarrow$ E, $X_1 \leftrightarrow$ C, $X_2 \leftrightarrow$ A, $X_3 \leftrightarrow$ T, $X_4 \leftrightarrow$ G, then allows the general form of an arbitrary base to be expressed as '$X_m \leftrightarrow$ Exp($m\cdot\omega\cdot$i)' (here, 'i' is the 'imaginary unit', $m$ = {0, 1, 2, 3, 4}). E.g., '$X_1\cdot\omega_3 = X_{1+3} = X_4$' means 'C$\cdot\omega_3$ = G' (base C is transformed to G by operator $\omega_3$). In practice, E$\circ\omega_0$ = E, E$\circ\omega_1$ = C, E$\circ\omega_2$ = A, E$\circ\omega_3$ = T, E$\circ\omega_4$ = G. All compositional scenarios are based on and referenced as diagrams (**Figs 2 and 3** [Ref. 6]; see S1 File).

In this paper, we introduce an additional rule; the 0-th base '$E_0$' is always assumed explicitly in any single-strand DNA sequence or fragments such as $D_{(k)j}$ = [$E_0|C_1|A_2|E_3|T_4|E_5|G_6||\mathbf{E_7 E_8 E_9}$...].

[Figs 6(A) and 7(A)]

The 0-th coordinate of any DNA/RNA sequence can only be $E_0$ if $E_0$ is only a symbol; hence, a partial function for a morphism (e.g., [$A_0|C_1|A_2|E_3|T_4|E_5|G_6||\mathbf{E_7 E_8 E_9}$...], [$G_0|C_1|A_2|$ $E_3|T_4|E_5|G_6||\mathbf{E_7 E_8 E_9}$...] that breaks (4)) cannot occur. Therefore, the objects satisfy the postulates of category theory (the compositionality of a source object and a target object) as mentioned in (4).

[Fig 1(B)]

At this juncture, we refer the reader to our previous report in which the DNA sequences are described using an operator that is defined on a five-fold phasor diagram [**Fig 4(B),** left panel; see **Fig 5** of [6]]. In that report, we considered a little devised Cartesian product of the group W, '$B_{(k)(j\rightarrow l)}$ = [$\mathbf{\omega_0}_0|\omega_1{}_1|\omega_0{}_2|\omega_2{}_3|\omega_3{}_4|\omega_4{}_5||\mathbf{\omega_0}_6|\mathbf{\omega_0}_7|$....] (an operator/a morphism defined on DNA, RNA or related products that are expressed by the bases; each $B_{(k)(\rightarrow)}$ is defined in place $p_k$, $k$ = 1,2,...)'. By making use of an identity DNA sequence '$D_{(k)0}$ (in place $p_k$) = [$\mathbf{E_0}|E_1|$ $E_2|E_3|$...|$E_{N-1}|E_N||\mathbf{E_{N+1}|E_{N+2}}$...]', any $D_{(k)j}$ (in place $p_k$) ($j$; the session number, $j$ = 1,2,3,...) is expressed as

$$
\begin{aligned}
D_{(k)j} &= D_0; B_{(k)(0\rightarrow j)} \\
&= [\mathbf{E_0}|E_1|E_2|E_3|E_4|E_5|\mathbf{E_6}|\mathbf{E_7}|\ldots]\circ[\mathbf{\omega_0}_0|\omega_1{}_1|\omega_0{}_2|\omega_2{}_3|\omega_3{}_4|\omega_4{}_5||\mathbf{\omega_{06}}|\mathbf{\omega_0}_7|\ldots] \\
&= [\mathbf{E}\circ\mathbf{\omega_0}_0|E\circ\omega_1{}_1|E\circ\omega_0{}_2|E\circ\omega_2{}_3|E\circ\omega_3{}_4|E\circ\omega_4{}_5||\mathbf{E}\circ\mathbf{\omega_0}_6|\mathbf{E}\circ\mathbf{\omega_0}_7|\ldots] \\
&= [\mathbf{E_0}|C_1|E_2|A_3|T_4|G_5||\mathbf{E_6}|\mathbf{E_7}|\ldots],
\end{aligned}
\tag{33}
$$

For the above description, we replace the symbol '$\circ$' with ';' used in [6] to indicate diagrammatic ordering of the application of the operations. We stipulate that there is an infinite number of 'E's and the trailing sequence of 'E's are implicitly implied as in '$D_{(k)j}$ = [$\mathbf{E_0}|C_1|A_2|T_3|$ $G_4||\mathbf{E_5 E_6 E_7}$...]'. In addition, the general form is, for example, '$D_{(k)j}$ = [$\mathbf{E_0}|C_1|A_2|T_3|G_4|$...........| $T_{N-1}|A_N||\mathbf{E_{N+1} E_{N+2}}$...]'. Here, the coordinates from the 1st to N-th entry are actual base sequences that include the case in which the 'E's are incorporated as for example in '$D_{(k)j}$ =

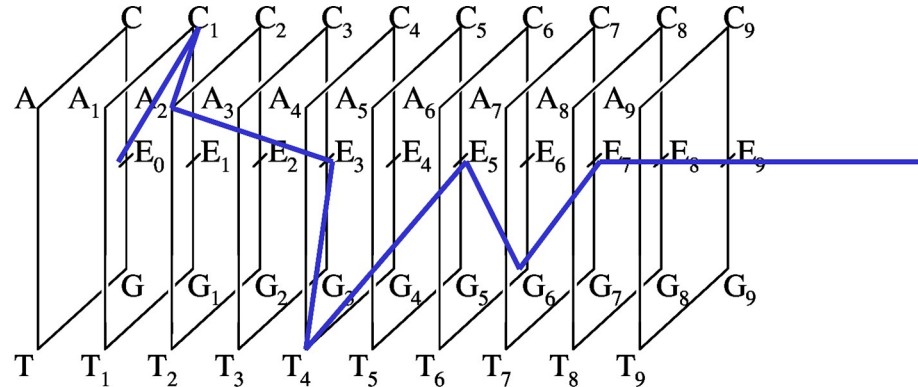

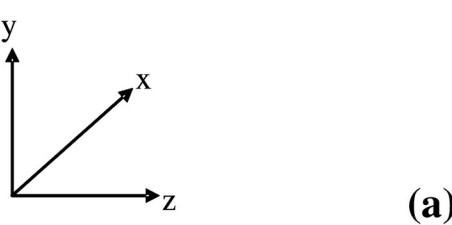

**(a)**

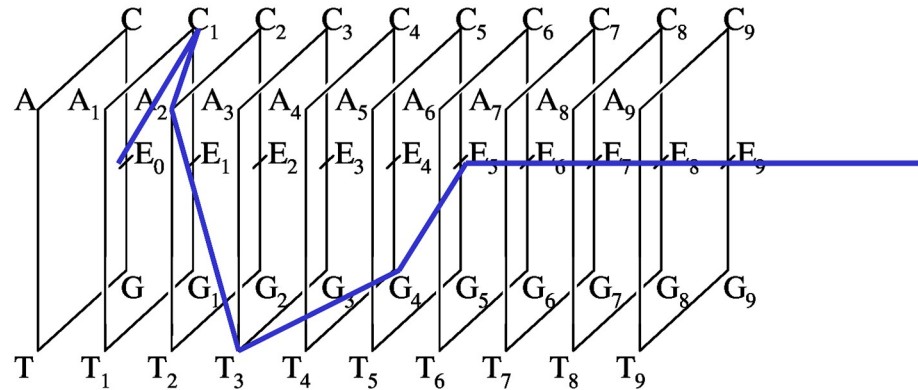

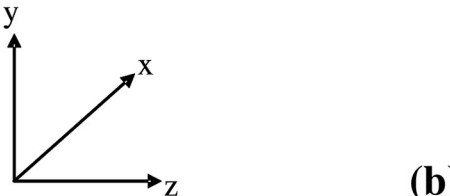

**(b)**

**Fig 6. Illustrations of DNA sequences with/without $<>$ on a square pole.** (a). Simple diagram of a single-strand DNA sequence '$D_{(k)j} = [E_0|C_1|A_2|E_3|T_4|E_5|G_6||\mathbf{E_7E_8E_9}\ldots]$', and its 3-dimensional polygonal line on a square pole. We stipulate that for any $D_{(k)j}$, the 0-th coordinate $E_0$ and the infinite tail '$\mathbf{EEE}\ldots$' are glued automatically with the actual sequence '$C_1A_2E_3T_4E_5G_6$'. (b). To highlight the actual sequence from 1st to 6-th '$C_1A_2E_3T_4E_5G_6$' (in this case, N = 6; the appropriate range is determined by the user), and by making use of the symbol $<>$, planes that include 'E's as

actual bases (i.e., $C_1A_2E_3T_4E_5G_6$) of $<D_{(k)j}>$ are deleted completely, except those planes of the 0-th coordinate $E_0$ and the infinite tail '$\mathbf{EEE}$. . .'; hence, $<D_{(k)j}> = <[E_0|C_1|A_2|E_3|T_4|E_5|G_6||\mathbf{E_7E_8E_9}. . .]> = [E_0|C_1|A_2|T_3|G_4||\mathbf{E_5E_6E_7}. . .]$.

$[\mathbf{E_0}|C_1|A_2|E_3|T_4|E_5|G_6||\mathbf{E_7E_8E_9}. . .]$' with an infinite tail sequence of 'E's devised so that any $D_{(k)j}$ is regarded as an element in a single set; see [Ref. 6].

## §3–3. Regarding the compatibility and treatment between the explicit and implicit identity elements

In applying the creation/annihilation morphism on an explicit 'E' and an implicit (), the conversion '() →E' (this morphism being unique) is incorporated or eliminated from any sequence as in

'$D_{(k)a} = [E_0|G_1|T_2|C_3|A_4||\mathbf{E_5E_6E_7}. . .]$

$= [E_0|G_1()T_2|C_3()()A_4||\mathbf{E_5E_6E_7}. . .] \rightarrow [E_0|G_1|\mathbf{E_2}|T_3|C_4|\mathbf{E_5}|\mathbf{E_6}|A_7||\mathbf{E_8E_9E_{10}}. . .] = D_{(k)b}$'

so long as they are recognized/tracked during the entire process. We stipulate that there are two ways of using E, specifically, the explicit base E (a sort of terminal object) and () the implicit base E (a sort of initial object). The deletion of E is performed via the inverse procedure of the above manipulation using the annihilation 'E→()' (this morphism being also unique). Note that the annihilation/creation of bases (coordinates) of '$D_{(k)a}$', the insertion and deletion of an explicit E, and the conversion between 'E ↔ ()' are actual morphisms, and there exists an ordinal operator that substitutes for this change. For example, the above annihilation/creation can also be performed by an ordinal operator of the Cartesian product such that '$B_{(k)(a\rightarrow b)} = [\omega_{0\ 0}|\omega_{0\ 1}|\omega_{2\ 2}|\omega_{2\ 3}|\omega_{4\ 4}|\omega_{4\ 5}|\omega_{0\ 6}|\omega_{2\ 7}||\boldsymbol{\omega_{0\ 8}}|\boldsymbol{\omega_{2\ 9}}|\boldsymbol{\omega_{0\ 10}}|. . . .]$' belongs to group W, for which the relation '$D_{(k)a}°B_{(k)(a\rightarrow b)} = D_{(k)b}$' can be verified. Henceforth, for this example of a hybrid model for genetic/molecular biology, all coordinates can be treated using this sort of ordinal operation as a Cartesian product such with '$B_{(k)(a\rightarrow b)}$' composed of multiple elements of group W.

Furthermore, $X_m$ and $X_{5-m}$ are inverse operations '$\omega_m^{-1} = \omega_{5-m}$'. With the notation (5→3), a sequence '$D_{(k)j} (5\rightarrow3) = [E_0|C_1|A_2|E_3|T_4|E_5|G_6||\mathbf{E_7E_8E_9}. . .]$' is transformed into its complementary base '$D_{(k)j}^\dagger (3\rightarrow5) = [E_0|G_1|T_2|E_3|A_4|E_5|C_6||\mathbf{E_7E_8E_9}. . .]$' simply from '$D_{(k)j}^\dagger = D_0;B_{(k)(0\rightarrow j)}^{-1}$' in accordance with examples given in our previous report [6]. Here, because any pair of bases, such as C and G, is related by a symmetry around the arbitrary E (also for A↔T), the sequence $D_{(k)j}^\dagger$ can be obtained simply by performing a counterclockwise rotation by π (rad) around any arbitrary E of the DNA wallpaper (to be clarified later). The terms '(5→3)' and '(3→5)' express the prime endings <5'(five prime)→3'(three prime)> or <3'→5'> accompanying a sequence designation. The notation '(5→3)' or '(3→5)' is simply an additional label representing the two possible types of endings of a single-stranded DNA. Then, this pair of homologous double-stranded DNA would be described using '/' meaning a pair of bases as in the following,

$[D_j(5\rightarrow3)/D_j^\dagger(3\rightarrow5)]$

$= [E_0/E_0|C_1/G_1|A_2/T_2|E_3/E_3|T_4/A_4|E_5/E_5|G_6/C_6||\mathbf{E_7/E_7}|\mathbf{E_8/E_8}|\mathbf{E_9/E_9}|. . .]$.

Here, the bi-operator '$B_j(5\rightarrow3)/B_j^\dagger(3\rightarrow5)$' yields

$$[D_j(5 \rightarrow 3)/D_j^\dagger(3 \rightarrow 5)] = [D_0(5 \rightarrow 3)/D_0(3 \rightarrow 5)]^\blacktriangle[B_{(0\rightarrow j)}(5 \rightarrow 3)/B_{(j\rightarrow 0)}(3 \rightarrow 5)]$$
$$= [D_0; B_j/D_0; B_j^\dagger], \tag{34}$$

where ▲ denotes the bi-operation of ';/;', and its operation on the pair of identity bases sequence $D_0/D_0$ by an operator such as $B_j$ is denoted '$D_0;B_j$'. In addition, the composition between $B_j$ and $B_k$ is rewritten as '$B_j;B_k$' but was written as '$B_j●B_k$' in our previous report [6].

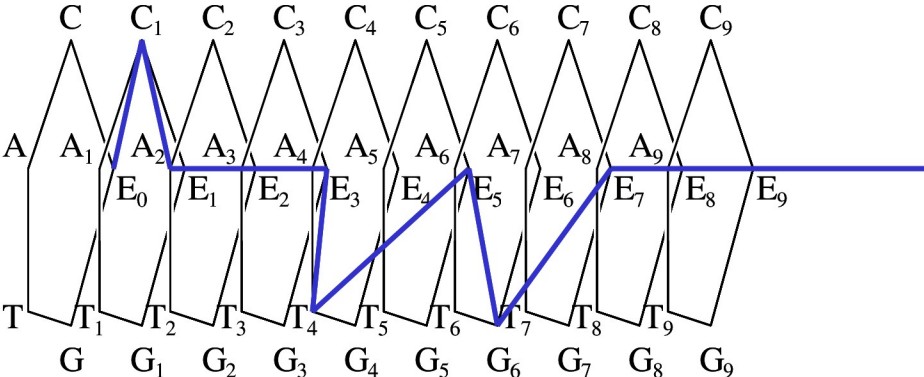

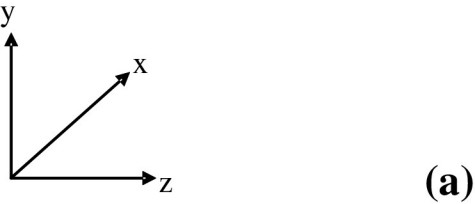

**(a)**

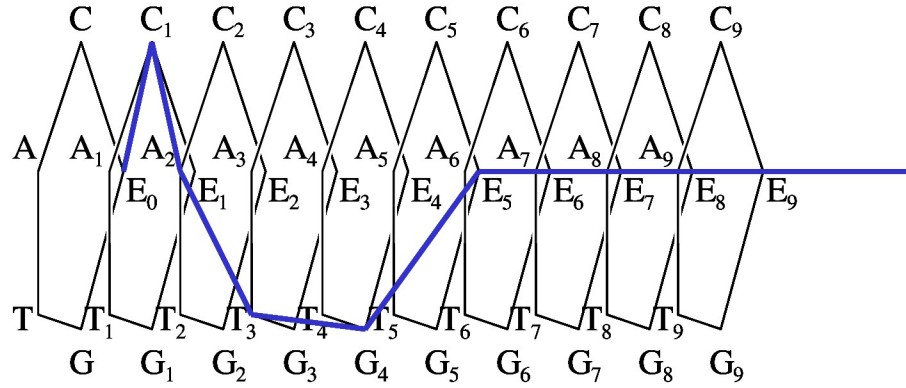

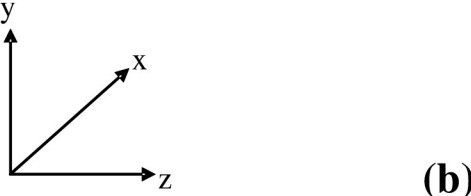

**(b)**

**Fig 7. Illustrations of DNA sequences with/without $<>$ on a pentagon pole.** (a). A simple diagram for sequence '$D_{(k)j} = [E_0|C_1|A_2|E_3|T_4|E_5|G_6||\mathbf{E_7E_8E_9}...]$' and its 3-dimensional polygonal line on a pentagon pole. In addition, we stipulate that for any $D_{(k)j}$, the 0-th coordinate $E_0$ and the infinite tail '$\mathbf{EEE}...$' are glued automatically with the actual sequence '$C_1A_2E_3T_4E_5G_6$'. (b). To highlight the actual sequence from 1st to 6-th bases '$C_1A_2E_3T_4E_5G_6$', planes that

include 'E's as actual bases (i.e., $C_1A_2E_3T_4E_5G_6$) of $<D_{(k)j}>$ are deleted completely, except the planes of the 0-th coordinate $E_0$ and infinite tail '**EEE**. . .' resulting in $<D_{(k)j}> = <[E_0|C_1|A_2|E_3|T_4|E_5|G_6||\mathbf{E_7E_8E_9}. . .]> = [E_0|C_1|A_2|T_3|G_4||\mathbf{E_5E_6E_7}. . .]$.

Hence, the above bi-operator parts of Eq (34) are rewritten as

$$B_j(5 \rightarrow 3)/B_j^{\dagger}(3 \rightarrow 5) = [\omega_0/\omega_{0\ 0}|\omega_1/\omega_{4\ 1}|\omega_2/\omega_{3\ 2}|\omega_0/\omega_{0\ 3}|\omega_3/\omega_{2\ 4}|\omega_0/\omega_{0\ 5}|\omega_4/\omega_{1\ 6}$$
$$||\mathbf{\omega_0/\omega_{0\ 7}|\omega_0/\omega_{0\ 8}|\omega_0/\omega_{0\ 9}}| \ldots]. \tag{35}$$

At this point, to enhance usability, we suggest the following optional representation in which a bijection for the vertices is considered. The vertices for bases C, A, T/U, and G on the pentagon are mapped onto the vertices of a square; the identity base E, however, is mapped onto the midpoint between the vertices of C and G [**Fig 6(A)**]. Although the squares in **Fig 6(A)** match up with the pentagons [**Fig 7(A)**], the positions of C, A, T/U, G on the planes are equi-spaced with an angular separation of 'π/2'. To be specific, they have been placed at E (1,0), C (1,1), A (−1,1), T/U (−1,−1), and G (1,−1) on the plane tangent to the flow of the sequence $D_{(k)j}$. This tangent plane is not the Gaussian plane [**Fig 4(B),** right panel] but the simple two-dimensional Euclidean plane. With this representation, the bijection between the vertices of the two planes is written E (1,0)↔E (= Exp(0)), C (1,1)↔C (= Exp(ω·i)), A (−1,1)↔A (= Exp(2·ω·i)), T/U (−1,−1)↔T/U (= Exp(3·ω·i)), and G(1,−1)↔G (= Exp(4·ω·i)) [**Figs 6(A) and 7(A)**]. Here, mathematically and operationally, we assume the rotational group W over the Gaussian plane is always in the background even if we display a figure using E (1,0), C (1,1), A (−1,1), T/U (−1,−1), and G(1,−1). Furthermore, we stipulate that when a polygonal line of bases is drawn, the position of '$E_0$' is always fixed, as in **Fig 4(B)**. This transformation enables a simple illustration of subsequent operations.

## §3–4. Notation for the identity elements that were explicitly/implicitly deleted using $< >$ or {}

At this point, to gain more usability, we consider a further manipulation concerning the notation $< >$. Concerning the meaning of $<D_{(k)j}>$, we stipulate that planes that include 'E's other than the initial 0-th E ($E_0$) and the infinite tail of 'E's (e.g., $||\mathbf{E_5E_6E_7}$. . .) of $D_{(k)j}$ are deleted entirely, and the remaining indexing coordinates closed up. As a result, the square pole corresponding to $<D_{(k)j}>$ that does not include 'E's as bases other than the initial 0-th E and the infinite tail of 'E's is

$<D_{(k)j}> = <[E_0|C_1|A_2|E_3|T_4|E_5|G_6||\mathbf{E_7E_8E_9}. . .]> = [E_0|C_1|A_2|T_3|G_4||\mathbf{E_5E_6E_7}. . .]$

[Fig 6(B)]

The meaning of this rule becomes clear in a later description; however, to avoid confusion over the use of the symbol, we provide a brief description of $< >$ and another symbol {} that was introduced in [6]. Let '$D_{(k)j} = [E_0|C_1|A_2|E_3|T_4|E_5|G_6||\mathbf{E_7E_8E_9}. . .]$'; then, there are two optional displays:

1) in $<D_{(k)j}>$, imaginary bases 'E's as bases other than the initial 0-th E and the infinite tail of 'E's are deleted from $D_{(k)j}$; that is, $<D_{(k)j}> = [E_0|C_1|A_2|T_3|G_4||\mathbf{E_5E_6E_7}. . .]$. and

2) in {$D_{(k)j}$}, bases 'E's remain in all sequences despite them being included explicitly in $D_{(k)j}$; {$D_{(k)j}$} = $[C_1|A_2|T_4|G_5||\mathbf{E_7E_8E_9}. . .]$ (the 0-th E, any E, and the infinite tail sequence of 'E's exist; however, the display of E in the '1−N'-th coordinate (in this case, N = 6) is only omitted in the display). In fact, the coordinates of $D_{(k)j}$ and {$D_{(k)j}$} are equivalent. However, for $<D_{(k)j}>$, we assume that, for a sequence of single-strand DNA, there exists a bijection between the sequence of $<D_{(k)j}>$ and the shape of its polygonal line along the square pole. This bijection

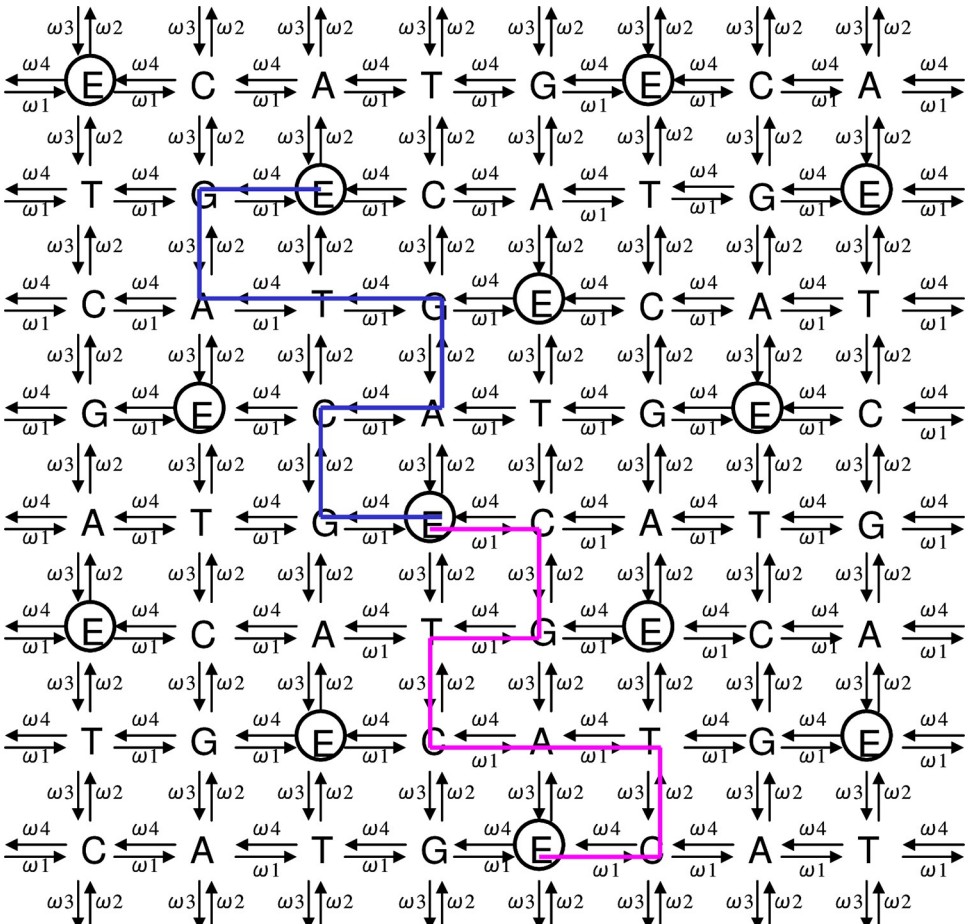

**Fig 8. Illustration of a pair of complementary DNA sequences on the DNA wallpaper.** For the wallpaper pattern, we suppose $D_{(k)j}$ (in place $p_k$) = [$E_0|G_1|C_2|E_3|A_4|E_5|G_6|T_7|E_8|A_9|G_{10}||\mathbf{E_{11}E_{12}E_{13}}$...]', with $<>$ meaning the deletion of imaginary bases 'E' other than the initial 0-th E and the infinite tail of 'E's. Hence, $<D_{(k)j}>$ is written $<D_{(k)j}>$ = [$E_0|G_1|C_2|A_3|G_4|T_5|A_6|G_7||\mathbf{E_8E_9E_{10}}$...]. A rule applies in which a polygonal line begins from an arbitrary $E_0$, so, in this Figure, $<D_{(k)j}>$ is described as the polygonal line marked in blue. Because $D_{(k)j}{}^{\dagger}$ (the complementary sequence of $D_{(k)j}$) is given as $D_{(k)j}{}^{\dagger}$ = [$E_0|C_1|G_2|E_3|T_4|E_5|C_6|A_7|E_8|T_9|C_{10}||\mathbf{E_{11}E_{12}E_{13}}$...], $<D_{(k)j}{}^{\dagger}>$ is displayed as $<D_{(k)j}{}^{\dagger}>$ = [$E_0|C_1|G_2|T_3|C_4|A_5|T_6|C_7||\mathbf{E_8E_9E_{10}}$...] and is drawn as a polygonal line marked in pink. Obviously, the shape of $<D_{(k)j}>$ and $<D_{(k)j}{}^{\dagger}>$ are symmetrically disposed with respect to '$E_0$' over the wallpaper pattern. Therefore, the shape of the polygonal lines of $<D_{(k)j}>$ and $<D_{(k)j}{}^{\dagger}>$ are point-symmetry-related around a common $E_0$. Interestingly, we may infer there is a bijection between the shape of polygonal line $<D_{(k)j}>$ and the contents of a sequence of bases '$G_1|C_2|A_3|G_4|T_5|A_6|G_7||\mathbf{E_8E_9E_{10}}$...'. In other words, if a polygonal line is provided only with $E_0$', the content of bases can be determined uniquely and automatically (The blue polygonal line instantly provides the above base sequence). Adapted from **Fig 4** of [Ref. 6] with permission (S1 File).

also holds between $<D_j>$ and the shape of its polygonal line over the DNA wallpaper (see **Fig 6** of [Ref. 6]. The revised figure is drawn in [**Fig 8**]. Similarly, we assume that there is bijection between $<D_{(k)j}>$ and the shape of its polygonal line [**Fig 6(B)**]. We conjecture that this bijection enables an unknown base sequence of $<D_{(k)j}>$ to be identified from the shape of its polygonal line if the latter is already known.

Naturally, to treat this issue with only a regular pentagon pole is possible [see **Fig 7(A) and 7(B)**]. In this article, we use the regular $p$-polygon pole with '$p = 5$' (pentagon pole) where multiple phasor diagrams (planes) in the Gaussian plane are placed in parallel. By connecting the individual vertices, the DNA and other sequences can be drawn as an orbit on a regular pentagon pole [**Fig 7(A) and 7(B)**]. Precisely, there is a bijection between $D_{(k)j}$ and the shape of its

polygonal line [**Fig 7(A)**] and another bijection between $<D_{(k)j}>$ and the shape of its polygonal line [**Fig 7(B)**]. In this regard, for ease of use and the advantage of this mapping, the simplification that we prefer is to focus on the reduction in the number of vertices from five to four in shape [**Fig 6(A) and 6(B)**] and to gain visual simplicity when presented in three dimensions.

## §3–5. On the mapping from DNAs to RNAs and the conversion of their dimensions using a unit of Descartes' geometry

Next, apart from the above stipulation, and to view the mapping between $D_{(k)j}$ (DNA) into $R_{(k'\backslash j}{}^{\dagger}$ (RNA), we consider the following cyclic group $\Gamma = \{\sigma, n\}$, where element n is the identity operation that does not change any base; e.g., A;n = A, T;n = T and E;n = E. Conversely, element $\sigma$ produces the change $\{E, C, A, T, G\} \leftrightarrow \{E, C, A, U, G\}$, and '$\sigma^2$ = n (no operation)' holds. In practice, $\sigma$ yields the following changes,

$$E; \sigma = E(\text{or } E \times \sigma = E), \ C; \sigma = C(\text{or } C \times \sigma), \ T; \sigma = U(\text{or } T \times \sigma = U),$$
$$U; \sigma = T(\text{or } U \times \sigma = T), \ G; \sigma = G(\text{or } G \times \sigma = G). \tag{36}$$

Hence, the simple conversion between DNA and RNA is written

$$D_{(k)j}; \sigma = R_{(k)j}{}^{\dagger}, \ R_{(k)j}{}^{\dagger}; \sigma = D_{(k)j}. \tag{37}$$

Because the group $\Gamma = \{\sigma, n\}$ is $\{Z_2,; \text{or} \times\}$, $D_{(k)j}$ and $R_{(k'\backslash j}{}^{\dagger}$ may be expressible as a Cartesian product $\{Z_5{}^{\times\infty} \times Z_2,; \text{or} \times\}$ where '; ' (= $\times$) denotes the operation between '$D_{(k)j}$'s or '$R_{(k')j}{}^{\dagger}$'s.

If the primary DNA sequence is written '$D_{(k)j} = [E_0|G_1|E_2|T_3|A_4|C_5||\mathbf{E_6}|\mathbf{E_7}|...]$', the corresponding RNA sequence $R_{(k')j}{}^{\dagger}$ is written $[E_0|G_1|E_2|T_3|A_4|C_5||\mathbf{E_6}|\mathbf{E_7}|...];\sigma$ (or $[E_0|G_1|E_2|T_3|A_4|C_5||\mathbf{E_6}|\mathbf{E_7}|...]\times\sigma$) ($\sigma$ acting on any coordinate independently) = $[E_0;\sigma|G_1;\sigma|E_2;\sigma|T_3;\sigma|A_4;\sigma|C_5;\sigma||\mathbf{E_6};\sigma|\mathbf{E_7};\sigma|...] = [E_0|G_1|E_2|U_3|A_4|C_5||\mathbf{E_6}|\mathbf{E_7}|...]$. In other words, any DNA (RNA) sequence is an element of the Cartesian product $Z_5{}^{\times\infty} \times Z_2$ (or $D_{(k)j} \times \Gamma$), such as '$R_{(k')j}{}^{\dagger} = [E_0|G_1|E_2|U_3|A_4|C_5||\mathbf{E_6}|\mathbf{E_7}|...];n$ (or $[E_0|G_1|E_2|T_3|A_4|C_5||\mathbf{E_6}|\mathbf{E_7}|...]\times\sigma$)'. In this regard, the expression '$D_{(k)j}$' is also regarded as an abbreviation of '$D_{(k)j};n$ ($\in Z_5{}^{\times\infty} \times Z_2$)'. In practice, this is always possible if we are given '$D_{(k)j} = [E_0|G_1|E_2|T_3|A_4|C_5||\mathbf{E_6}|\mathbf{E_7}|...]$' as '$D_{(k)j};n;n;n;...;n$ ($\in Z_5{}^{\times\infty} \times Z_2 \times Z_2 \times Z_2 \times ... \times Z_2$)'. If we regard '$D_{(k)j};n$ ($\in Z_5{}^{\times\infty} \times Z_2$)' as the product after '$m-1$' shrinkages '$Z_2 \times ... \times Z_2$ (m-tuple) = $Z_2$ (one-tuple)', then that is considered possible. In other words, the essential form of the token $D_{(k)j}{}^{\bullet}$ is expressed as $Z_5{}^{\times\infty} \times Z_2$; the optional expression

$$Z_5{}^{\times\infty} \times Z_2 \times Z_2 \times Z_2 \times ... \times Z_2 \text{ or } Z_5{}^{\times\infty}, \tag{38}$$

is permissible. If we shrink the parts of '$...\times Z_2 \times Z_2 \times Z_2 \times ... \times Z_2$', only the appropriate power of the binary operation n;n (= n) is necessary and sufficient to meet the conditions stipulated. Conversely, if a specific form of '$Z_5{}^{\times\infty} \times Z_2 \times ... \times Z_2$ (r-tuple)' is required from '$Z_5{}^{\times\infty} \times Z_2 \times ... \times Z_2$ (s-tuple)' (for non-negative integers $r,s$; $r \geq s$), '$r - s$' applications of the operation 'n;n = n' completes this transformation between the two tuples. In this way, there is freedom to choose the number of applications '$\times$n' in accordance with the operations of the group $\Gamma = \{\sigma, n\}$. For $D_{(k)j}$ '($\in Z_5{}^{\times\infty} \times Z_2 \times ... \times Z_2$ (r-tuple))' and $D_{(k)l}$ '($\in Z_5{}^{\times\infty} \times Z_2 \times ... \times Z_2$ (s-tuple))', with $r \neq s$, the corresponding tokens $D_{(k)j}$ and $D_{(k)l}$ do not seem to belong to the same place $p_k$. A difference clearly exists, and usually we only assume the essential case,

$$D_{(k)j}(\in Z_5{}^{\times\infty}) \text{ and } R_{(k)j}{}^{\dagger}(\in Z_5{}^{\times\infty}). \tag{39}$$

We introduce the rule that the 3-dimensional polygonal line of '$D_{(k)j} = D_{(k)j};n$' or '$R_{(k')j}{}^{\dagger} = D_{(k)j};\sigma$' ($\in Z_5{}^{\times\infty} \times Z_2$)' is expressed as '$[D_{(k)j}] = D_{(k)j};n$' and '$[R_{(k')j}{}^{\dagger}] = D_{(k)j};\sigma = R_{(k')j}{}^{\dagger};n$'. In this way, for

these extreme circumstances, we find an arbitrary/appropriate number ';n $(\in Z_2)$' at the end of any $D_{(k)j}$ or $R_{(k)j}{}^{\dagger}$ if needed. This rule is an application of the use of the 'unit 1' in Descartes' geometry [49]. Therefore, we can regard $D_{(k)j}$ $(\in Z_5{}^{\times\infty})$ as

$$D_{(k)j}; n; n; n; \ldots n \ (\in Z_5{}^{\times\infty} \times Z_2 \times Z_2 \times Z_2 \times \ldots \times Z_2). \tag{40}$$

If $D_{(k)j}$ also expresses the state $D_{(k)j}$ where all 'n;n;n;...n' are deleted from '$D_{(k)j}$;n;n;n;...;n', then, the following relations hold; $D_{(k)j} = D_{(k)j}$;n;n;n;...;n or $D_{(k)j}$;n $= D_{(k)j}$, $D_{(k)j}$;$\sigma = R_{(k)j}{}^{\dagger}$. Similarly, for $R_{(k)j}{}^{\dagger}$,

$$R_{(k)j}{}^{\dagger} = R_{(k)j}{}^{\dagger}; n; n; n; \ldots; n \text{ or } R_{(k)j}{}^{\dagger}; n = R_{(k)j}{}^{\dagger}, \ R_{(k)j}{}^{\dagger}; \sigma = D_{(k)j}. \tag{41}$$

Moreover, the binary compositions between two DNAs such as '$D_j;D_k$', '$D_j;D_j{}^{\dagger}$' or '$D_j;D_j{}^{-1}$' are regarded as being nonsensical as an operation because $D_j$ and $D_j{}^{-1}$ are not operators. However, note that the primary single-strand DNA sequence '$D_{(k)j}$' and the complementary sequence $D_{(k)j}{}^{\dagger}$ have a symmetry with respect to the horizontal plane that includes all Es [see **Fig 9(A) and 9(B)**].

## §3–6. Method for the detection of the complementary region of a pair of DNA(RNA) sequences using two polygonal lines

For more generality, we introduce another rule: if we interpret any single-strand DNA fragment such as '$A_3T_4$' in '$D_{(k)r} = [E_0|G_1|E_2|\mathbf{A_3}|\mathbf{T_4}|C_5||E_6|E_7|E_8|\ldots]$', this part is also recognized as '$D_{(k)s} = [E_0|\mathbf{A_1}|\mathbf{T_2}||E_3|E_4|E_5|\ldots]$'. With this understanding, any DNA sequence '$D_{(k)r}$' or its partial sequence '$D_{(k)s}$' may be related by bijection with the shape of its polygonal line for either the DNA wallpaper, the square pole or the pentagon pole.

For the DNA wallpaper, the bijection is repeated, and the shape of the polygonal line of $<D_{(k)j}>$ may be determined uniquely over the DNA wallpaper [**Fig 8**]; adapted from **Fig 4** of [Ref. 6]), where naturally, any polygonal line of the $<D_{(k)j}>$s never crosses over the 'E's. If we distinguish $<D_{(k)j}>$ from $<D_{(k)j}>_\theta$ that has been rotated through $\theta$ (rad) around $E_0$, the polygonal line of $<D_{(k)j}{}^{\dagger}>$ (the complementary sequence of $<D_{(k)j}>$) may be obtained automatically by the counterclockwise rotation of $\pi$ (rad) around $E_0$ of the DNA wallpaper [**Fig 8**]. In the three cases—DNA wallpaper, square pole, and pentagon pole—the polygonal line of $<D_{(k)j}>$ and the actual single-strand DNA sequence such as 'CATG...' are in correspondence, and there is a bijection between the two polygonal lines [**Figs 8 and 9(A), 9(B)**].

Therefore, the specific sequence of bases in any arbitrary $D_{(k)j}$ or a pair of DNA sequences have complementary sequences that may be identified graphically. They can be converted into complementary sequences of DNA that can be performed easily from the shape of the polygonal line. For the complementary region of $D_{(k)j}$ and $D_{(k)l}$ (single-strand DNA sequences in place $p_k$), the composition of the operator parts of '$D_{(k)j} = [\ldots|C_m|A_{m+1}|T_{m+2}|G_{m+3}|\ldots] = [\ldots|E°\omega_{1\ m}|E°\omega_{2\ m+1}|E°\omega_{3\ m+2}|E°\omega_{4\ m+3}|\ldots]$' and '$D_{(k)l} = [\ldots|G_m|T_{m+1}|A_{m+2}|C_{m+3}|\ldots] = [\ldots|E°\omega_{4\ m}|E°\omega_{3\ m+1}|E°\omega_{2\ m+2}|E°\omega_{1\ m+3}|\ldots]$' (i.e., $B_{(k)j} \bullet B_{(k)l} = [\ldots|\omega_{1\ m}|\omega_{2\ m+1}|\omega_{3\ m+2}|\omega_{4\ m+3}|\ldots] \bullet [\ldots|\omega_{4\ m}|\omega_{3\ m+1}|\omega_{2\ m+2}|\omega_{1\ m+3}|\ldots]$) is considered to satisfy the following relation,

$$B_{(k)j} \bullet B_{(k)l} (= B_{(k)l} \bullet B_{(k)j}) = [\ldots|\omega_1 \bullet \omega_{4\ m}|\omega_2 \bullet \omega_{3\ m+1}|\omega_3 \bullet \omega_{2\ m+2}|\omega_4 \bullet \omega_{1\ m+3}|\ldots]$$
$$= [\ldots|\omega_{0\ m}|\omega_{0\ m+1}|\omega_{0\ m+2}|\omega_{0\ m+3}|\ldots] = B_{(k)0} \text{ or } 'D_{(k)0} \circ (B_{(k)j} \bullet B_{(k)l}) = D_{(k)0}$$
$$= [\ldots|E_m|E_{m+1}|E_{m+2}|E_{m+3}|\ldots]', \tag{42}$$

meaning that $B_{(k)j}$ and $B_{(k)l}$ are 'inverses' in accordance with their shapes [**Fig 9(A) and 9(B)**]. This relationship may aid in assessing whether arbitrary regions of two single-strand DNA sequences are complementary, although careful treatment is necessary regarding the direction of $<5'$(five prime)$\to 3'$(three prime)$>$ because there may be instances for which $D_{(k)j}$ and an

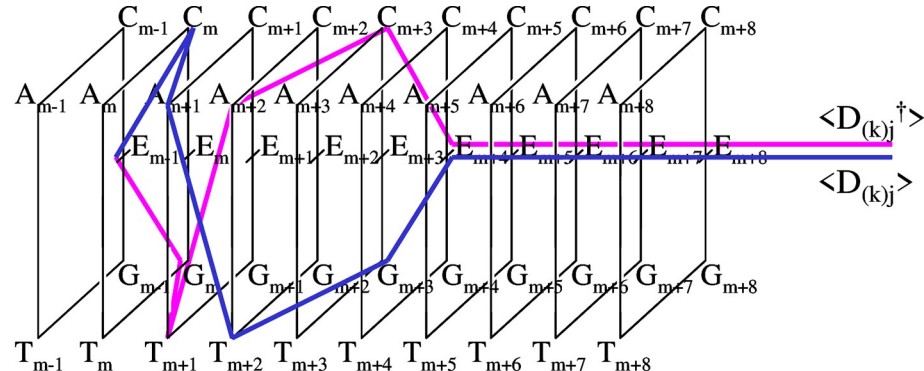

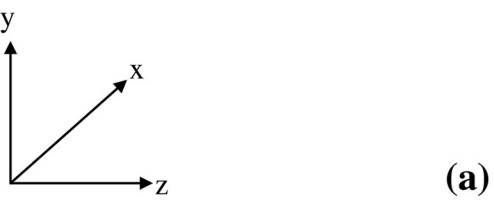

**(a)**

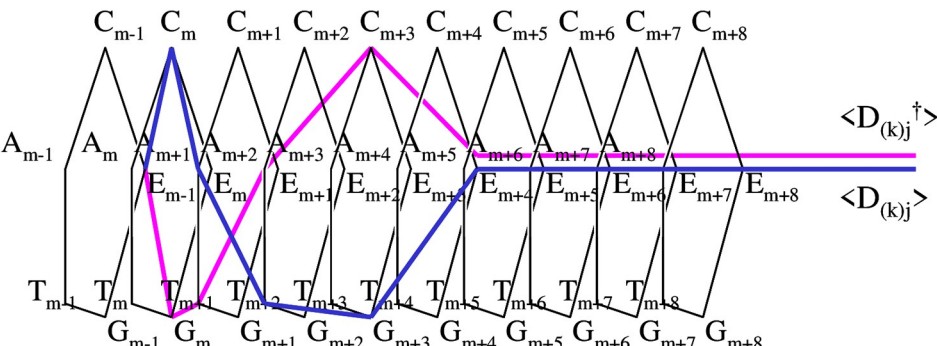

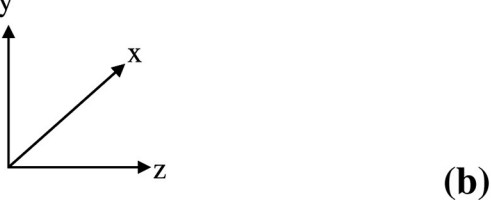

**(b)**

**Fig 9. Illustrations of the complementary region of a pair of DNA sequences on two types of poles.** (a). Given the two polygonal lines $<D_{(k)j}> = [\ldots|C_m|A_{m+1}|T_{m+2}|G_{m+3}|\ldots]$ (marked in blue) and $<D_{(k)l}> = <D_{(k)j}^\dagger> = [\ldots|G_m|T_{m+1}|A_{m+2}|C_{m+3}|\ldots]$ (marked in pink), they are symmetry-related in the 'x-z' plane that includes the line '$\ldots E_m E_{m+1} E_{m+2} E_{m+3}\ldots$'. This symmetry helps to identify a pair of two complementary segments of single-strand DNA (RNA) sequences. Note that the shape of their polygonal lines underscores a bijection between $<D_{(k)j}>$ and $<D_{(k)j}^\dagger>$ (= $<D_{(k)l}>$). However, in general, even if the shapes of $<D_{(k)a}>$ and $<D_{(k)b}>$ are complementary-related, they may not always be bijective or complementary. (b). Given the two polygonal lines $<D_{(k)j}> = [\ldots|C_m|A_{m+1}|T_{m+2}|G_{m+3}|\ldots]$ (marked in blue) and $<D_{(k)l}> = <D_{(k)j}^\dagger> = [\ldots|G_m|T_{m+1}|A_{m+2}|C_{m+3}|\ldots]$ (marked in pink), they are clearly symmetry-related in the 'x-z' plane that includes the line '$\ldots E_m E_{m+1} E_{m+2} E_{m+3}\ldots$'.

inverse sequence of $D_{(k)l}$ are related, as in Eq (42), but have been omitted. Clearly, this similarity relationship and usage may be used on the pentagon pole representation [**Fig 9(B)**].

## §3–7. Supplemental explanations concerning the explicit/implicit identity sequences in category theory

In the present article, we introduce a rule by which an explicit identity sequence '$D_0 = D_{(k)0}$ (in place $p_k$) = $[E_0|E_1|E_2|E_3|\ldots|E_{N-1}|E_N||E_{N+1}E_{N+2}\ldots]$' may be converted into an implicit identity sequence '$D_\phi$ (in place $p_\phi$) = $[( )\,( )\,( )\ldots( )\,( )\,||( )\,( )\ldots] = ( )$' [**Fig 10(A) and 10(B)**], and vice versa using the annihilation/creation morphism (as $D_0 \leftrightarrow D_\phi$), and '$<D_0> = <D_{(k)0}> = <[E_0|$

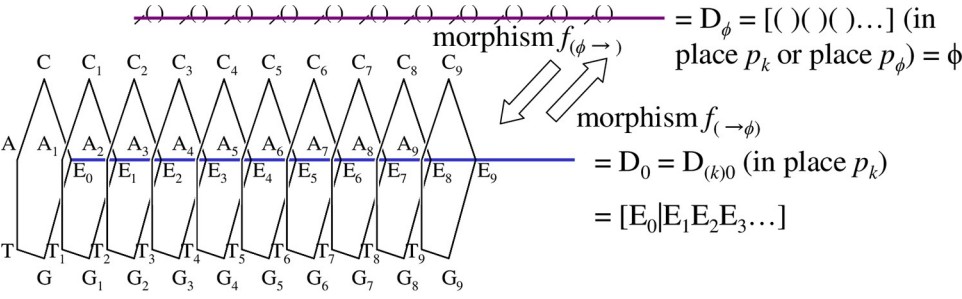

**(a)**

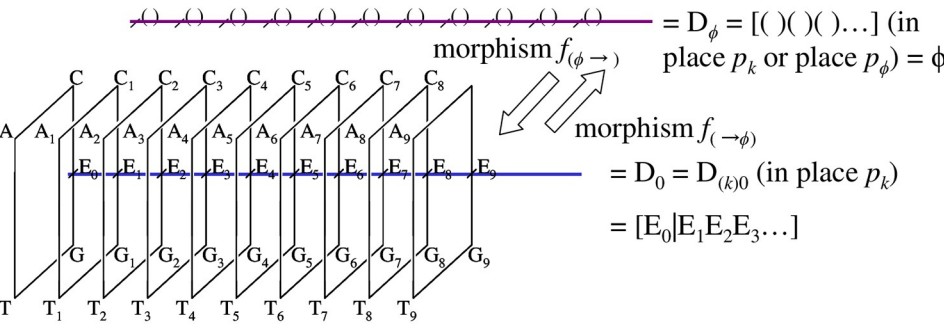

**(b)**

**Fig 10. Illustrations of the relationships between explicit and implicit identity DNA sequences on two types of poles.** (a). By making use of the annihilation morphism $f_{(\to\phi)}$, an explicit identity sequence $D_0$ (a line that passes through the vertex of '$E_0$s on the pentagon pole) is converted into $D_\phi$ (the implicit identity sequence or nothing) in accordance with '$D_0;f_{(\to\phi)} = D_\phi$'. This procedure is also written as '$<D_0> = <[E_0|E_1|E_2|E_3|\ldots|E_{N-1}|E_N||E_{N+1}E_{N+2}\ldots]>$ $= D_\phi$'. In contrast, $f_{(\phi\to)}$ yields an explicit identity sequence $D_0$ from $D_\phi$ in accordance with '$D_\phi;f_{(\phi\to)} = D_0$'. (b). With no regard to the shape of the polygonal line, the same conversion is considered performable as '$D_0;f_{(\to\phi)} = D_\phi$' and '$D_\phi;f_{(\phi\to)} = D_0$'.

$E_1|E_2|E_3|\ldots|E_{N-1}|E_N||E_{N+1}E_{N+2}\ldots]> = D_{(k)0}$' (in place $p_k$). Because $<D_0>$ may be obtained by the map $f_{(\to\phi)}$ on $D_0$ as '$D_0;f_{(\to\phi)} = D_\phi$', then from a certain viewpoint, for any '$D_0$', the map $<D_0>$ is a result of the map of an $n$-tuple of annihilation morphisms $f_{(\to\phi)};f_{(\to\phi)};\ldots f_{(\to\phi)} = f_{(\to\phi)}^{\times n}$ ($n$; natural number or $\infty$). However, we must stipulate that $f_{(\to\phi)}$ changes all '$E_0E_0\ldots E_0$' in $D_0$ to () () $\ldots$(); henceforth, we write $f_{(\to\phi)}^{\times n}$ only as $f_{(\to\phi)}$. Significantly, similar to '$S_! \neq S_0$' given in Eq (16) and (17) in **§2-2-3**, because $D_{(k)0}$ (in place $p_k$) may be regarded as a terminal object that is defined on category $A_k$ (at place $p_k$ obtained via viewing group (or monoid) $A_k$ as category), and $D_\phi$ (at place $p_\phi$) may be seen as the initial object of category $A_k$, there is no zero object '$D_0 = D_\phi$' in category $A_k$ (at place $p_k$).

Similarly, in **§2-2-2**, a unique place $p_\phi$ for $D_\phi$ (= () () $\ldots$(); implicit identity sequence of single-strand DNA) need not always be expressed explicitly in the above conversion procedure because it suffices to consider it somewhere in the entire EPN for an arbitrary annihilation/creation morphism like $f_{(k\to\phi)}$ or $f_{(\phi\to k)}$. If it is imposed, $R_\phi$ (= () () $\ldots$(); implicit identity sequence of a single-strand RNA) is equivalent to '$D_\phi$ = () () $\ldots$()'.

In place $p_k$, a morphism $f_{(k)a}$ (an operator, for example, such as '$f_{(k)a} = B_{(k)a} = [\omega_{2\ 1}|\omega_{3\ 2}|\omega_{0\ 3}|\omega_{4\ 4}||\mathbf{\omega_{0\ 5}}|\mathbf{\omega_{0\ 6}}|\ldots]$, $B_{(k)b}$, $B_{(k)c}\ldots$; $\in$group W') may act on any token $D_{(k)j}\bullet$ and produce an image. If the image of $f_{(k)a}$ to $D_{(k)j}$ is denoted by '$D_{(k)j};f_{(k)a}$' or '$f_{(k)a}°D_{(k)j}$', the operation itself is denoted '$D_{(k)j};f_{(k)a};$' or '$f_{(k)a}°D_{(k)j};$'. In fact, $D_{(k)j}$ is a similar product of $B_{(k)a}$ ($a$; non-negative integer) because $D_{(k)j}$ is determined as $D_{(k)j} = D_{(k)0};B_{(k)(0\to j)}$; thereby, a simple operation may be performable among any arbitrary $D_{(k)j}$ and morphism $f_{(k)a}$ similar to the example in **§3**.

## §3–8. An explicit/implicit identity protein and its treatment

Furthermore, an interpretation of the codon (triplet of bases) from the polygonal line comprising bases given in **Fig 6(A) and 6(B)**, and **Fig 9(A) and 9(B)** is possible in accordance with the conventional codon table [see **Fig 11(A)**]. The combination of three vertices (to each, a base 'C, A, T, G or E' is allocated) may be translated into an amino-acid sequence (protein: a sequence of amino acids) in parallel via morphism $f_{(4\to6)}$ or $f_{(5\to7)}$. If limited to the procedure of a morphism $f_{(4\to6)}$, $f_{(6\to4)}$, $f_{(5\to7)}$ or $f_{(7\to5)}$, the mapping from a sequence of bases to that of an amino-acid sequence (protein) is performable. There, we believe that the inverse procedure is automatically performable using electronic tools such as a scanner for sequences of molecules. We stipulate that by recognizing the shape of the polygonal line of the amino-acid sequence (protein), any sequence of triplets of lines of codons may be interpreted from the corresponding amino-acid sequences (proteins) such as 'AUG→(M)', 'CUG→(L)', and 'AGU→(S)'; that is, $Rs_{(4)j}^{\dagger}$ (in place $p_4$, $j$ a non-negative session number) = $[E_0|A_1U_2G_3C_4U_{5}$$G_6A_7G_8U_9||E_{10}E_{11}E_{12}\ldots] \to$ '$Pr_{(6)j}$ = (M)(L)(S)$\Delta\Delta\Delta\ldots$'(in place $p_6$) → '$<Pr_{(7)j}>$ = (M)(L)(S)' (in place $p_7$), where $\Delta$ means an explicit identity (empty) amino acid (protein), and a conversion '$\Delta\leftrightarrow$()' (implicit $\Delta$, nothing) may be considered. In more detail, we may instigate a rule by which a triplet 'EEE' is mapped to an explicit identity amino acid (protein) '$\Delta$'; for example, with '$Rs_{(4)1}^{\dagger}$ = $[E_0|A_1U_2G_3\mathbf{E_4}C_5U_6G_7A_8G_9U_{10}||E_{11}E_{12}E_{13}E_{14}\ldots]$' (in place $p_4$) and the appropriate use (firing) of an operator $f_{(4)a}$ (that is, an operator like $B_{(j\to k)}$ in place $p_4$), we produce the following changes

$$Rs_{(4)1}^{\dagger} \text{ (in place } p_4) \to Rs_{(4)1a}^{\dagger} = [E_0|A_1U_2G_3\mathbf{E_4E_5E_6}C_7U_8G_9A_{10}G_{11}U_{12}||E_{13}E_{14}E_{15}E_{16}\ldots]$$
$$= [E_0|(A_1U_2G_3)(\mathbf{E_4E_5E_6})(C_7U_8G_9)(A_{10}G_{11}U_{12})||(E_{13}E_{14}E_{15})E_{16}\ldots] \text{ (in place } p_4). \tag{43}$$

Additionally, the morphism $f_{total\text{-}splicing}$ (= $f_{t\text{-}sp} = f_{(4\to6)}$) leads to further changes;

{$Rs_{(4)1a}^{\dagger}$}; $f_{t\text{-}sp}$ = $Pr_{(6)1} = \Delta_0(M_1)\Delta_2(L_3)(S_4)\Delta_5\Delta_6\Delta_7\ldots$' (in place $p_6$) (we stipulate that an initial $\Delta_0$ and an infinite trail such as '$\Delta_5\Delta_6\Delta_{7\ldots}$' are glued automatically) [**Fig 11(B)**].

| Second base position | | | | | | | | | |
|---|---|---|---|---|---|---|---|---|---|
| | | U | | C | | A | | G | |
| U | UUU | F | UCU | S | UAU | Y | UGU | C | U |
| | UUC | | UCC | | UAC | | UGC | | C |
| | UUA | L | UCA | | UAA | Stop | UGA | Stop | A |
| | UUG | | UCG | | UAG | | UGG | W | G |
| C | CUU | L | CCU | P | CAU | H | CGU | R | U |
| | CUC | | CCC | | CAC | | CGC | | C |
| | CUA | | CCA | | CAA | Q | CGA | | A |
| | CUG | | CCG | | CAG | | CGG | | G |
| A | AUU | I | ACU | T | AAU | N | AGU | S | U |
| | AUC | | ACC | | AAC | | AGC | | C |
| | AUA | | ACA | | AAA | K | AGA | R | A |
| | AUG | M | ACG | | AAG | | AGG | | G |
| G | GUU | V | GCU | A | GAU | D | GGU | G | U |
| | GUC | | GCC | | GAC | | GGC | | C |
| | GUA | | GCA | | GAA | E | GGA | | A |
| | GUG | | GCG | | GAG | | GGG | | G |

*First base position* (left), *Third base position* (right)

**(a)**

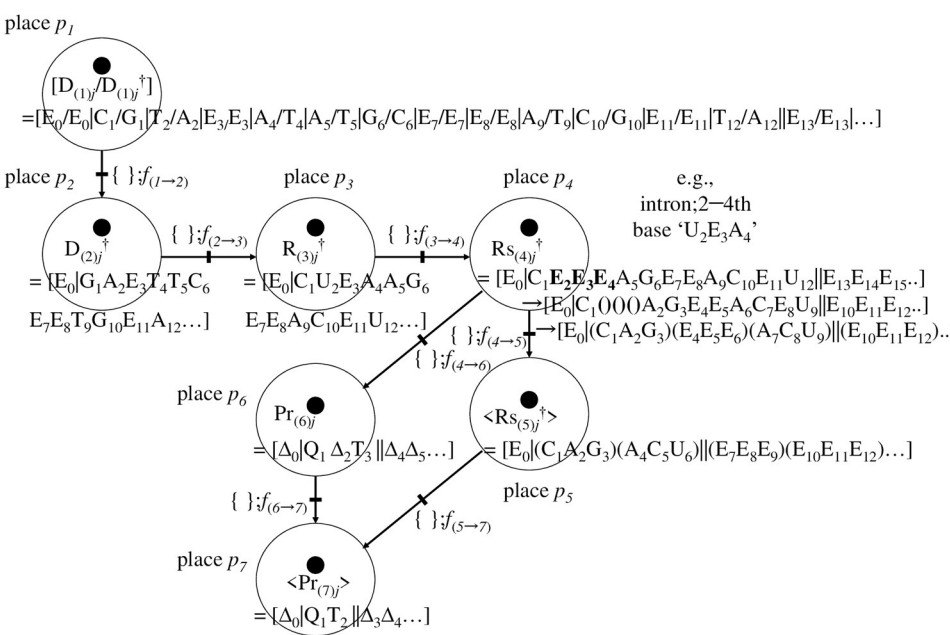

**(b)**

**Fig 11. Codon table and illustration of the canonical central dogma using an EPN.** (a). The conventional codon table is displayed. A codon (triplet of bases) is translated into an amino acid that is incorporated in the growing polypeptide chain. The number of triplet combinations is 64. The stop codons are UAA, UAG, and UGA. (b). The canonical central dogma from the double-strand DNA to the protein Q (Glutamine). . . is displayed in simplified form using the EPN. First, a token ● is included in place $p_1$. Subsequently, morphisms $f_{(1\to2)}, f_{(2\to3)}, f_{(3\to4)}, \dots, f_{(6\to7)}$ are fired in order. There are two pathways via 'place $p_4 \to p_6 \to p_7$' or 'place $p_4 \to p_5 \to p_7$'. Although multiple tokens are drawn, only one token is included in one of the places in $p_1, p_2, \dots, p_7$. Multiplying a marking x by an incidence matrix

produces one move of a token. However, the incidence matrices of the EPN are slightly different depending on which pathway is selected from the above three.

Then, $Pr_{(6)1}$ (in place $p_6$) is translated into $<Pr_{(7)1}> = \Delta_0(M_1)(L_2)(S_3)\Delta_4\Delta_5\Delta_6\ldots$ (in place $p_7$) via morphism $f_{post\text{-}prot}$ ($= f_{po\text{-}pr} = f_{(6\to7)}$). In another example, with '$Rs_{(4)2}^\dagger = [E_0|$ $\mathbf{E_1}A_2U_3G_4\mathbf{E_5}C_6U_7G_8A_9G_{10}U_{11}||E_{12}E_{13}E_{14}E_{15}\ldots]$', a specific morphism $f_{(4\to5)}$ (that is, an operator like $B_{(j\to k)}$ defined from place $p_4$ in place $p_5$) produces the result, $Rs_{(4)2}^\dagger$ (in place $p_4$)$\to$'$<Rs_{(5)2}^\dagger>$ (in place $p_5$) $= [E_0|()\ A_1U_2G_3()\ C_4U_5G_6A_7G_8U_9||E_{10}E_{11}E_{12}E_{13}\ldots]$

$= [E_0|(A_1U_2G_3)(C_4U_5G_6)(A_7G_8U_9)||E_{10}E_{11}E_{12}E_{13}\ldots]$'; once firing of the specific morphism $f_{total\text{-}prot}$ ($= f_{t\text{-}pr} = f_{(5\to7)}$) produces the next change $<Rs_{(5)2}^\dagger>$ (in place $p_5$)$\to$'$\{<Rs_{(5)2}^\dagger>\}$; $f_{(5\to7)} = <Pr_{(7)2}> = [\Delta_0|(M_1)(L_2)(S_3)||\Delta_4\Delta_5\Delta_6\ldots]$ in place $p_7$ (via morphism $f_{(5\to7)}$)'. The above manipulations of 'E's are to adjust their number so that all bases form complete triplets. Note that, for the firing rules, we stipulate that when a morphism $f_{(4\to6)}$ ($= f_{t\text{-}sp}$) is assumed, we disregard 'pathway place $p_4 \to p_5 \to p_6$' because $f_{(5\to6)}$ is an inconvenience and nonsensical if $f_{(4\to6)}$ and $f_{(5\to6)}$ are always fireable/non-fireable contemporaneously. Because sequences in place $p_5$ such as $<Rs_{(5)2}^\dagger> = [E_0|(A_1U_2G_3)(C_4U_5G_6)(A_7G_8U_9)||E_{10}E_{11}E_{12}E_{13}\ldots]$ do not include 'E's as coordinates, other than an initial $E_0$ and an infinite tail of 'E$_0$'s at the end of the sequence, only products that can go through morphism $f_{(5\to7)}$ (when $f_{(5\to7)}$ is fireable) such as '$<Rs_{(5)2}^\dagger>$; $f_{(5\to7)} = <Pr_{(7)2}> = [\Delta_0|(M_1)(L_2)(S_3)||\Delta_4\Delta_5\Delta_6\ldots]$' in place $p_7$ can also be outputs in place $p_6$ such as a sequence like $Pr_{(6)2} = [\Delta_0|(M_1)(L_2)(S_3)||\Delta_4\Delta_5\Delta_6\ldots]$ via morphism $f_{(5\to6)}$ (when $f_{(5\to6)}$ is fireable). Therefore, there are cases where morphism $f_{(5\to6)}$ is fireable. However, morphism $f_{(5\to7)}$ is sufficient, and morphism $f_{(5\to6)}$ is not always necessary. For simplicity of the model, we omit morphism $f_{(5\to6)}$. In this manner, any amino-acid sequence (protein) in place $p_6$ may be expressed as a sequence of polygonal lines appended by an infinite explicit trailing $\Delta\Delta\Delta\ldots$. Similarly, any amino-acid sequence (protein) in place $p_7$ may be expressed as $\phi$, a sequence of polygonal lines for which there is no infinite explicit trailing $\Delta\Delta\Delta\ldots$. At this stage, an identity sequence of amino acids (proteins) is also definable where the infinite number of $\Delta$ is ordered as '$Pr_{(k)0} = \Delta\Delta\Delta\Delta\Delta\Delta\ldots$' (in this case, $k = 6$, because place $p_6$ is designed for an amino-acid sequence (protein) that may include $\Delta$'s). We call $Pr_{(k)0}$ an explicit identity protein sequence. Moreover, $<Pr_{(k)0}>$ is defined as the state where the '$\Delta$'s of the explicit identity protein are deleted via the annihilation morphism for $Pr_{(k)0}$; '$\{Pr_{(k)0}\}; f_{(k\to\phi)} = <Pr_{(k)0}> = [\Delta_0|\Delta_1\Delta_2\Delta_3\ldots] \neq [()\ ()\ ()\ \ldots] = \phi\phi\phi\ldots\phi = \phi = Pr_\phi$' (in this instance, $k = 6$). Note the relation '$<Pr_{(k)0}> = Pr_{(k)0}$ (in place $p_k$) $= [\Delta_0|\Delta_1\Delta_2\Delta_3\ldots]$'. We refer to it as an implicit identity protein sequence. Naturally, an inverse procedure is considered via the creation morphism, '$\{\phi\}; f_{(\phi\to k)} = \{Pr_\phi\}; f_{(\phi\to k)} = \{<Pr_{(k)0}>\}; f_{(\phi\to k)} = Pr_{(k)0} = \Delta\Delta\Delta\ldots\Delta$' (with $k = 6$). By this rule, any sequence of arbitrary amino-acid sequences (protein) $Pr_{(k)j}$ ($j$; the session number, non-negative number $j = 0$, 1,2,...) can be defined so that $Pr_{(k)j}$ belongs to set P (the set of amino-acid-sequences (protein)), although no operational meaning is assumed between the '$Pr_{(k)j}$'s at this moment. The inverse translation (firing) for the token from place $p_6$ to place $p_4$ is considered to be always possible; however, that from place $p_7$ to place $p_6$ or from place $p_5$ to place $p_4$ seems to be impossible because in the latter cases, the bijection is lost. Furthermore, using a notation similar to that in (**#remark 1**), for morphisms $f_{t\text{-}sp}$ ($= f_{(4\to6)}$) and $f_{po\text{-}pr}$ ($= f_{(6\to7)}$), the associative law holds if an adequate adjustment is made to the index numbers; for morphism $f_{t\text{-}sp}$ ($= f_{(4\to6)}$), the index numbers of the amino-acid sequences (proteins) $Pr_{(k)j}$ that are transformed from $Rs_{(k)j}^\dagger$ do not change until all manipulations to the sequence of operations is completed; in detail, the associative law within Eq (43) is obvious because this procedure in place $p_4$ obeys the group operation.

## §3–9. Regarding the symbols [[]], (()) and $<<>>$ that treat the explicit/implicit deletion of identity elements

Then, for the 'part of the operation $f_{t\text{-}sp}$ (= $f_{(4\to6)}$)' and, with, say, token 'Rs$_{(4)2}^{\dagger}$ (expressed in Eqs (44) and (45))', the intron parts are changed into appropriate 'E's in accordance with the former part of the splicing procedure in place $p_4$.

If the deletion of explicit 'E's from the $r$-th to the $s$-th coordinate ($s,r$: non-negative integers with $s\leq r$; $s,r = 0,1,2,\ldots$) of token Rs$_{(4)2}^{\dagger}$ is denoted 'Rs$_{(4)2}^{\dagger};<>_{s,r}$', the following manipulation is confirmed;

$$\text{for } \text{Rs}_{(4)2}^{\dagger} = [E_0|A_1U_2G_3\mathbf{E_4E_5E_6}C_7U_8G_9\mathbf{E_{10}E_{11}E_{12}}C_{13}G_{14}U_{15}\mathbf{E_{16}E_{17}E_{18}}A_{19}C_{20}G_{21}$$
$$||E_{22}E_{23}E_{24}E_{25}\ldots](\text{in place } p_4), \tag{44}$$

$$\text{Rs}_{(4)2}^{\dagger};<>_{10,12},$$
$$= [E_0|A_1U_2G_3(\mathbf{E_4E_5E_6})C_7U_8G_9()_{\mathbf{10-12}}C_{13}G_{14}U_{15}(\mathbf{E_{16}E_{17}E_{18}})A_{19}C_{20}G_{21}||E_{22}E_{23}E_{24}E_{25}\ldots] \tag{45}$$

(the place is not determined because this is an intermediate state of places $p_4$ and $p_5$), where the index numbers 10–12 remain until the sequence of operations is completed; this is displayed using double round brackets (()) as below. In this way, Rs$_{(4)2}^{\dagger};<>_{4,6};<>_{10,12};<>_{16,18}$ (essentially, the operations are performed in diagrammatic order, from left to right on Rs$_{(4)2}^{\dagger}$),

$= [E_0|A_1U_2G_3()_{\mathbf{4\text{-}6}}C_7U_8G_9()_{\mathbf{10\text{-}12}}C_{13}G_{14}U_{15}()_{\mathbf{16\text{-}18}}A_{19}C_{20}G_{21}||E_{22}E_{23}E_{24}E_{25}\ldots]$

(the place is not determined for the reason previously stated),

$= [E_0|A_1U_2G_3C_4U_5G_6C_7G_8U_9A_{10}C_{11}G_{12}||E_{13}E_{14}E_{15}E_{16}\cdot\ldots]$

(the place is not determined), using double round brackets (()),

$= ((\text{Rs}_{(4)2}^{\dagger};<>_{4,6};<>_{10,12};<>_{16,18}))$ (the place is not determined).

In the above example, by performing the additional operation $<>_{13,\infty}$ to Rs$_{(4)2}^{\dagger}$, $((\text{Rs}_{(4)2}^{\dagger};<>_{4,6};<>_{10,12};<>_{16,18}));<>_{13,\infty}$

$= [E_0|A_1U_2G_3C_4U_5G_6C_7G_8U_9A_{10}C_{11}G_{12}] = <\text{Rs}_{(4)2}^{\dagger}>$ (in place $p_5$).

Hence, the associative law for Rs$_{(4)2}^{\dagger}$, $(<>_{4,6};<>_{10,12});<>_{16,18} = <>_{4,6};(<>_{10,12};<>_{16,18})$ is evidently obeyed. In an ordinal $<\text{Rs}_{(4)2}^{\dagger}>$, $s = 1$, $r = \infty$ is imposed, and $<\text{Rs}_{(4)2}^{\dagger}> = <\text{Rs}_{(4)2}^{\dagger}>_{1,\infty} = [E_0|A_1U_2G_3C_4U_5G_6C_7G_8U_9A_{10}C_{11}G_{12}]$ (in place $p_5$).

For the 'part of the operation $f_{t\text{-}sp}$ (= $f_{(4\to6)}$) and $f_{post\text{-}prot.}$ (= $f_{po\text{-}pr} = f_{(6\to7)}$).'; we see $f_{(4\to6)}$); $f_{(6\to7)}$ and $f_{(4\to5)}$);$f_{(5\to7)}$ commute because all tokens of place $p_5$ are products of Rs$_{(4)2}^{\dagger};<>_{r,s};\ldots$) to a sequence of amino acids (protein) (in place $p_6$) from the $r$-th to the $s$-th coordinate ($s,r$: non-negative integers with $s\leq r$; $s,r = 0,1,2,\ldots$) of, say, token Rs$_{(4)2}^{\dagger}$, denoted '$<\text{Rs}_{(4)2}^{\dagger}>;[]_{s,r}$', the following manipulation is verified for 'Rs$_{(4)2}^{\dagger} = [E_0|A_1U_2G_3()_{\mathbf{4\text{-}6}}C_7U_8G_9()_{\mathbf{10\text{-}12}}C_{13}G_{14}U_{15}()_{\mathbf{16\text{-}18}}A_{19}C_{20}G_{21}]$ (in place $p_5$)',

$$<\text{Rs}_{(4)2}^{\dagger}>;[]_{7,9}, = [E_0|A_1U_2G_3()_{\mathbf{4-6}}\underline{L}_{\mathbf{7-9}}()_{\mathbf{10-12}}C_{13}G_{14}U_{15}()_{\mathbf{16-18}}A_{19}C_{20}G_{21}]. \tag{46}$$

The combination of amino acids (protein) and bases for which the part underlined $\underline{L}_{7\text{-}9}$ means the amino-acid sequence (protein) of molecule 'L (leucine)' and the place to which $<\text{Rs}_{(4)2}^{\dagger};[]_{7,9}>$ belongs is not determined because this is an intermediate state for places $p_5$ and $p_6$ for which the index numbers 7–9 remain until the sequence of operations is completed and displayed using double round brackets (()).

Additionally, for the 'entire operation $f_{(4\to6)};f_{(6\to7)})$ or $f_{(4\to5)};f_{(5\to7)}$', we shall see whether the associative law holds. If the conversion procedure from a sequence of bases including explicit 'E's (place $p_4$) to a sequence of amino acids (protein) including explicit identity (empty) amino acids (proteins) '$\Delta$'s (in place $p_6$) from the $r$-th to the $s$-th coordinate (s,r: non-negative integers with $s\leq r$; $s,r = 0,1,2,\ldots$) of, say, token Rs$_{(4)2}^{\dagger}$ denoted 'Rs$_{(4)2}^{\dagger};[[]]_{s,r}$', the

following manipulation is verified for 'Rs$_{(4)2}$$^\dagger$, = [E$_0$|A$_1$U$_2$G$_3$**E$_4$E$_5$ E$_6$**C$_7$U$_8$G$_9$**E$_{10}$E$_{11}$E$_{12}$**C$_{13}$-G$_{14}$U$_{15}$**E$_{16}$E$_{17}$E$_{18}$**A$_{19}$C$_{20}$G$_{21}$||E$_{22}$E$_{23}$E$_{24}$E$_{25}$. . .] (in place $p_4$)',

$$\text{Rs}_{(4)2}{}^\dagger; [[]]_{7,9}, = [\text{E}_0|\text{A}_1\text{U}_2\text{G}_3\mathbf{E_4E_5E_6}\ \underline{\text{L}_{7-9}}\ \mathbf{E_{10}E_{11}E_{12}}\text{C}_{13}\text{G}_{14}\text{U}_{15}\mathbf{E_{16}E_{17}E_{18}}\text{A}_{19}\text{C}_{20}\text{G}_{21}$$

||E$_{22}$E$_{23}$E$_{24}$E$_{25}$. . .](the place is undetermined because this is an intermediate state of places $p_4$ and $p_7$).  (47)

The combination of amino acids and bases that is underlined $\underline{\text{L}_{7-9}}$ denotes the amino-acid molecule 'L (leucine)', and the place to which Rs$_{(4)2}$$^\dagger$;[[]] $_{7,9}$ belongs is not determined because this is an intermediate state of places $p_4$ and $p_6$ for which the index numbers 7–9 remain until the sequence of operations is completed and is displayed using double round brackets (()). Therefore, because ([[]] $_{7,9}$;[[]] $_{4,6}$);[[]] $_{19,21}$ = [[]] $_{7,9}$;([[]] $_{4,6}$;[[]] $_{19,21}$), the associative law for Rs$_{(4)2}$$^\dagger$ holds.

In this way, 'Rs$_{(4)2}$$^\dagger$;[[]] $_{7,9}$;[[]] $_{4,6}$;[[]] $_{19,21}$' is calculated to be

$$= [\text{E}_0|\text{A}_1\text{U}_2\text{G}_3(\underline{\mathbf{\Delta_{4-6}}})(\underline{\text{L}_{7-9}})(\mathbf{E_{10}E_{11}E_{12}})\text{C}_{13}\text{G}_{14}\text{U}_{15}(\mathbf{E_{16}E_{17}E_{18}})(\underline{\text{T}_{19-21}})||\text{E}_{22}\text{E}_{23}\text{E}_{24}\text{E}_{25}. . .],  \quad (48)$$

where the underlined $\underline{\mathbf{\Delta_{4-6}}}$, $\underline{\text{L}_{7-9}}$, and $\underline{\text{T}_{19-21}}$ refer to the amino-acid molecule (T: tyrosine), (the place is not determined for the reason previously stated).

In the present state, the morphism $f_{t\text{-}sp}$ (= $f_{(4\to6)}$) is not fired completely for similar reasons stated for Eq (47). Moreover, if all triplets (codons) are converted into any appropriate amino-acid sequence (protein), the product may be included in place $p_6$ as a token. That is, an additional operation may change any coordinate into a suitable amino-acid sequence (protein); for example,

'((Rs$_{(4)2}$$^\dagger$;[[]] $_{7,9}$;[[]] $_{4,6}$;[[]] $_{19,21}$)); [[]] $_{1,3}$;[[]] $_{7,9}$;[[]] $_{10,18}$;[[]] $_{22,\infty}$,

= [E$_0$|A$_1$U$_2$G$_3$(**$\underline{\Delta_{4-6}}$**)($\underline{\text{L}_{7-9}}$)(**E$_{10}$E$_{11}$E$_{12}$**)C$_{13}$G$_{14}$U$_{15}$(**E$_{16}$E$_{17}$E$_{18}$**)($\underline{\text{T}_{19-21}}$)

||E$_{22}$E$_{23}$E$_{24}$E$_{25}$. . .];[[]] $_{1,3}$;[[]] $_{7,9}$;[[]] $_{10,18}$;[[]] $_{22,\infty}$,

= [$\Delta_0$|M$_1$$\Delta_2$L$_3$$\Delta_4$R$_5$$\Delta_6$T$_7$||$\Delta_8$$\Delta_9$$\Delta_{10}$. . .] = Pr$_{(6)2}$ (in place $p_6$)',where the parts to the left of || belong to an amino-acid sequence (protein), and that on the right of || is an explicit identity base of 'E's (This sequence denotes a combination of amino acids (protein) and bases. Additionally, $\Delta_0$ is required to be placed automatically). In detail, there are conversions: 'A$_1$U$_2$G$_3$→M$_1$ (methionine)', 'E$_4$E$_5$E$_6$→$\Delta_2$ (an explicit identity/empty an amino acid (protein))', 'C$_7$U$_8$G$_9$→L$_3$ (leucine)', 'E$_{10}$E$_{11}$E$_{12}$→$\Delta_4$', 'C$_{13}$G$_{14}$U$_{15}$→R$_5$ (arginie)', 'E$_{16}$E$_{17}$E$_{18}$→$\Delta_6$', 'A$_{19}$C$_{20}$G$_{20}$→T$_7$ (tyrosine)', 'E$_{22}$E$_{23}$E$_{24}$E$_{25}$. . .→$\Delta_8$$\Delta_9$$\Delta_{10}$. . . (infinite identities)' (in place $p_6$).

Here, the associative law for Rs$_{(4)2}$$^\dagger$, ([[]] $_{7,9}$;[[]] $_{4,6}$);[[]] $_{19,21}$ = [[]] $_{7,9}$;([[]] $_{4,6}$;[[]] $_{19,21}$), holds. In this regard, the practical utility of intermediate states such as those in Eqs (47) and (48) is unclear and less valuable because the associative law does not hold between morphism $f_{(4\to6)}$, $f_{(4\to5)}$, and $f_{(5\to6)}$.

Similarly, for morphism $f_{po\text{-}pr}$ (= $f_{(6\to7)}$); if the deletion of explicit '$\Delta$'s from the $r$-th to the $s$-th coordinate ($s$,$r$: non-negative integers with $s \leq r$; $s$,$r$ = 0,1,2,. . .) of, say, token 'Pr$_{(6)2}$ is denoted 'Pr$_{(6)2}$;<< >>$_{s,r}$,', the following manipulation is verified for 'Pr$_{(6)5}$ = [$\Delta_0$| M$_1$$\Delta_2$$\Delta_3$L$_4$R$_5$$\Delta_6$$\Delta_7$$\Delta_8$T$_9$R$_{10}$$\Delta_{11}$$\Delta_{12}$L$_{13}$M$_{14}$||$\Delta_{15}$$\Delta_{16}$$\Delta_{16}$. . .] = (in place $p_6$)',

Pr$_{(6)2}$$^\dagger$;<< >>$_{2,3}$ = [$\Delta_0$|M$_1$()**$_{2\text{-}3}$** L$_4$R$_5$$\Delta_6$$\Delta_7$$\Delta_8$T$_9$R$_{10}$$\Delta_{11}$$\Delta_{12}$L$_{13}$M$_{14}$||$\Delta_{15}$$\Delta_{16}$$\Delta_{17}$. . .].

In this way, Pr$_{(6)2}$$^\dagger$;<< >>$_{2,3}$;<< >>$_{6,8}$;<< >>$_{11,12}$ = [$\Delta_0$|M$_1$()**$_{2\text{-}3}$**L$_4$R$_5$()**$_{6\text{-}8}$** T$_9$R$_{10}$$\Delta_{11}$$\Delta_{12}$L$_{13}$M$_{14}$||$\Delta_{15}$$\Delta_{16}$$\Delta_{17}$. . .]. Then,

((Pr$_{(6)2}$$^\dagger$;<< >>$_{2,3}$;<< >>$_{6,8}$<< >>$_{11,12}$));<< >>$_{15,\infty}$

= [$\Delta_0$|M$_1$()**$_{2\text{-}3}$**L$_4$R$_5$()**$_{6\text{-}8}$**T$_9$R$_{10}$()**$_{11\text{-}12}$** L$_{13}$M$_{14}$||()$_{15}$()$_{16}$()$_{17}$. . .]

$$= \Delta_0(\text{M}_1)(\text{L}_2)(\text{R}_3)(\text{T}_4)(\text{R}_5)(\text{L}_6)(\text{M}_7) \quad \text{(in place } p_7\text{)}. \quad (49)$$

The associative law '(<< >>$_{2,3}$;<< >>$_{6,8}$);<< >>$_{11,12}$ = << >>$_{2,3}$;(<< >>$_{6,8}$<< >>$_{11,12}$)' holds for morphism $f_{po\text{-}pr}$ (= $f_{(6\to7)}$), although the proof is omitted.

In general, the associative law is necessary for composition in category theory and seems to be satisfied for morphisms $f_{t-sp}$ $(= f_{(4 \to 6)})$, $f_{po-pr}$ $(= f_{(6 \to 7)})$, and $f_{(4 \to 7)}$. The identity morphisms $f_{(t-sp)0}$ $(= f_{(4 \to 6)0})$, $f_{(po-pr)0}$ $(= f_{(6 \to 7)0})$, and $f_{(4 \to 7)0}$ (no operation) are also considered.

In summary, if a double-strand DNA of arbitrary length is given and is expressed as $D_{(k)j}$ (primary DNA sequence) or $[D_{(k)j}/D_{(k)j}^{\dagger}]$ (double-stranded DNA sequences), then the following occurs: i) the firing of a morphism from $f_{(1 \to 2)}$ and $f_{(2 \to 3)}$ can be performed automatically; ii) then, in place $p_3$, the appropriate order that determines what parts are introns and exons is necessary so that an adequate region of exons can be translated into amino acids (protein); iii) if that is conducted, then, for a sequence of bases, an appropriate adjustment for the composition of the complete sequence of triplets is necessary in place $p_4$, and iv) the pathway either via $f_{(4 \to 5)}$—$f_{(5 \to 7)}$ or via $f_{(4 \to 6)}$—$f_{(6 \to 7)}$ can be performed automatically. As a result, an arbitrary $D_{(k)j}$ or $[D_{(k)j}/D_{(k)j}^{\dagger}]$ can be translated into an amino-acid sequence (protein). For step iv), a visual graphic approach is possible using electronic tools that read/translate them.

## §3–10. On a potential method to describe the protein sequences using the rotation group

In general, a sequence of amino acids contains biological information concerning the synthesis of various proteins. Conventionally, 20 amino acids such as 'I (isoleucine), L (leucine), F (phenylalanine), G (glycine), S (serine), and R (arginine)' exist in the human body and they are used as codings for protein synthesis. Here, we believe there are more than 20 different amino acids plus a 'start amino acid' $\perp$ (methionine (M) often plays this role), a 'stop amino acid' $\top$, and a 'do-nothing amino acid (an identity amino acid/protein) $\Delta$'. They match up with the 23-fold phasor diagram on the Gaussian plane and the 23 polygon poles like the pentagon poles of **Figs 4(B), 7(A), 7(B) and 9(B)**. Here, the 'start amino acid' $\perp$ acts as an order to start reading the amino-acid code; the 'stop amino acid' $\top$ acts as an order to stop reading, and the 'do-nothing amino acid (an identity amino acid/protein) $\Delta$ does nothing. The rotational group of order 23 $\{\alpha_0, \alpha_1, \alpha_2. . .\alpha_{21}, \alpha_{22}\}$ is defined as '$\alpha_i;\alpha_j = \alpha_{i+j}$ (mod 23)' with '$\alpha_1 = 2\pi/23$ (rad)'. Similar to instances of a bijection between $<D_{(k)j}>$ and the shape of the polygonal line of a single-strand DNA sequence, we infer that there is a bijection between $<Pr_{(7)j}>$ (from $Pr_{(7)j}$, all '$\Delta$'s other than the initial $\Delta_0$ (0-th $\Delta$) and infinite trail '$\Delta\Delta\Delta$. . .' are deleted) and the shape of the polygonal line of the amino-acid sequence. The detection of an amino-acid sequence from only the shape of its polygonal line is possible, and vice versa. However, we believe that the development of this idea from the viewpoint of interpretation needs more analysis concerning the higher usable forms, e.g., $\perp$ and $\top$, which may better be placed symmetrically to the line that is drawn from protein $\Delta$ to the center of the 23 polygons. To some degree, this proposition is unrealistic because we were not able to provide a meaningful order for the 23 proteins for which an inverse of $\alpha_i^{-1}$ that satisfies '$\alpha_i^{-1};\alpha_1 = \alpha_1;\alpha_i^{-1} = \alpha_0$' could have more meanings than appears in the definition of $D_j$ or $R_j$; the shape of the polygonal line for the inverse $D_{(k)j}^{\dagger}$ is immediately obtained from the polygonal line of the symmetrical position of '$D_{(k)j}$' around an arbitrary $E_0$. For this reason, although it seems possible, the composition of the cyclic group using amino acids has to be curtailed at this stage.

Fortunately, for the transformation/exchange of sequences of amino acids (proteins), there is a considerable number of models that treat the manner of exchange using for example the rotational group based on the regular icosahedron, the 20 vertices of which are associated with the 20 amino acids. Therefore, the transformation between the amino-acid sequences may be suitably treated in this existing formalism.

As a result, the canonical central dogma may be describable in an EPN [**Fig 11(B)**] where individual tokens express one of the products including DNA, RNA, pre-mRNA, mRNA, and related biological products.

## §3–11. A treatment of the central dogma using matrices (Naganuma's theory of operation matrix)

To overview the rules developed for the present EPN model, we refer to our previous report. The following morphisms are assumed to express the behaviors of DNA/RNA sequences with some adjustments; in this regard, for the present description, '∘' means the left translation rule, ';' operations are executed in diagrammatic order for all processes, and {}; $f (= f^∘\{\})$ morphism f acts on element in the argument of object {}.

(**Appendix A in S1 Appendix**)

Conventionally, for the standard PN, an incidence matrix is often used to express the flow of the markings. Although not an incidence matrix, the matrix represents the flow of processes associated with the canonical central dogma. The entries of this "incidence-matrix-like" matrix A are operators with tokens for the respective markings. Multiplications involving A are interpreted as a flow of process information. Using the list of morphisms (**Appendix A in S1 Appendix**) and a flowchart [**Fig 11(B)**], examples of the operators and the entries in A (and A') are shown in **Appendix B in S1 Appendix**. This application is referred to as the "theory of the operation matrix" proposed by Naganuma where the 'operation matrix' contains operators as entries in a square matrix and mathematically expresses general physical aspects of various models, with a special focus on the theoretical/mathematical limitations of simple systems [50]. In our model, as a specific example of that theory, the operators that act on '$D_{(k)j}$'s and '$R_{(k)j}$'s are included in the entries of matrix A, and as a result, time flows associated with the canonical central dogma are expressed discretely as multiplications by matrix A provided all morphisms (transitions) for the flow [**Fig 11(B)**] are fireable. Details of the calculation are described in **Appendix B in S1 Appendix**.

## §3–12. A treatment of the somatic recombination (VDJ recombination) for antibodies using the concept of colimit

We now turn our attention to a more advanced use of our concepts, specifically, the expression of the limit or colimit ('tensor product' by ignoring the arrows and terminal object) that may apply to somatic recombination (VDJ recombination) for antibodies [51]. That is, the V segment is the sequence $V = [V_1|V_2|V_3|\ldots|V_{w1}||\mathbf{E}|\mathbf{E}|\mathbf{E}|\ldots]$; in each part of '$V_1,V_2,V_3,\ldots,V_i,\ldots,V_{w1}$', except for the infinite tail of 'E's ($||EEE\ldots$), each $V_i$ ($i = 1,2,\ldots,w_1$) is also a sequence of bases that are parts of set V as follows. Suppose '$N_v$' is the number equal to or greater than the maximum number of recognized non-E bases among

$$V_1, \ V_2, \ldots, V_{w1} (\in \text{object V});  \tag{50}$$

then, for example,

$V_i = [C_1|A_2|\ldots|G_{(Nv-1)}|T_{Nv}||\mathbf{E}_{(Nv+1)}|\mathbf{E}_{(Nv+2)}|\mathbf{E}_{(Nv+3)}|\ldots] = [C_1|A_2|G_3|T_4||\mathbf{E}_5|\mathbf{E}_6|\mathbf{E}_7|\ldots]$, where the number of recognized bases is '$N_v = 4$'. That is, in a certain V, with '$N_v = 4$', the number of recognized non-E bases is smaller than $N_v$, the coordinates being filled with explicit 'E's as in $V_3 = [T_1|G_2|T_3||\mathbf{E}_4\mathbf{E}_5\ldots]$ so that $N_v$ (in this case '$N_v = 3$') can be chosen equivalent among $V_1, V_2,\ldots,V_{w1}$ ($\in$object V) with '$V_N \geq V_1, V_2,\ldots,V_{w1}$'.

For a more practical example, we put '$V_3 = [A_1|C_2||\mathbf{E}_3\mathbf{E}_4\ldots]$'; we find that the sequence of bases $V_1, V_2,$ and $V_3$ belong to a unique object,

$$V_1 = [C_1|A_2|G_3|T_4||\mathbf{E}_5|\mathbf{E}_6|\mathbf{E}_7|\ldots] = [C_1|A_2|G_3|T_4||\mathbf{E}_5|\mathbf{E}_6|\mathbf{E}_7\ldots](\in \text{object V}),$$
$$V_2 = [T_1|G_2|T_3||\mathbf{E}_4\mathbf{E}_5\ldots] = [T_1|G_2|T_3|E_{4\_}||\mathbf{E}_5\mathbf{E}_6\mathbf{E}_7\ldots](\in \text{object V}),  \tag{51}$$
$$V_3 = [A_1|C_2||\mathbf{E}_3\mathbf{E}_4\ldots] = [A_1|C_2|E_{3\_}|E_{4\_}||\mathbf{E}_5\mathbf{E}_6\mathbf{E}_7\ldots](\in \text{object V}),$$

where the minimum $N_v$ is '$N_v = 4$' implying there are no 'non-E's on the right side of the $N_v$-th component in Eq (51). Note that in this case, $N_v \geq 4$ suffices in satisfying condition (51).

Similarly, the D segment is the sequence D = $[D_1|D_2|D_3|\ldots|D_{w2}||\mathbf{E}|\mathbf{E}|\mathbf{E}|\ldots]$, a direct combination of '$D_1, D_2, D_3, \ldots, D_i, \ldots, D_{w2}$' except for the infinite tail of 'E's ($||EEE\ldots$); e.g., $D_1$ ($i = 1, 2, \ldots, w_2$) is also a sequence of bases that are parts of object D. Let '$N_D$' be the number equal to or greater than the maximum number of recognized non-E bases among $D_1, D_2, \ldots, D_{w2}$ ($\in$object D). For example,

$D_2 = [G_1|\ldots|C_{ND}||\mathbf{E_{(ND+1)}}|\mathbf{E_{(ND+2)}}|\mathbf{E_{(ND+3)}}|\ldots] = [G_1|A_2|C_3||\mathbf{E_4}|\mathbf{E_5}|\mathbf{E_6}|\ldots]$,

where the number of recognized bases is '$N_D = 3$'. Similarly, the J segment is the sequence J = $[J_1|J_2|J_3|\ldots|J_{w3}||\mathbf{E}|\mathbf{E}|\mathbf{E}|\ldots]$, a direct combination of '$J_1, J_2, J_3, \ldots, J_i, \ldots, J_{w3}$' except for the infinite tail of 'E's ($||EEE\ldots$); e.g., $J_1$ ($i = 1, 2, \ldots, w_3$) is also a sequence of bases of object J. Let '$N_J$' be the number that is equal to or greater than the maximum number of recognized non-E bases among $J_1, J_2, \ldots, J_{w3}$ ($\in$object J). Hence, as an example, we have

$J_2 = [A_1|C2|\ldots|GNJ||E(NJ+1)|E(NJ+2)|E(NJ+3)|\ldots] = [A1|C2|T3|G_4|A_5||\mathbf{E_6}|\mathbf{E_7}|\mathbf{E_8}|\ldots]$,

where the number of recognized bases is '$N_J = 5$'. $N_D$ and $N_J$ must be counted in a similar manner as $N_V$ in object V.

Therefore, because the following limit or colimit $\Pi$ is considerable as in the other place $p_{k''}$ (k''; an appropriate number), $V \otimes D \otimes J = [V|D|J||EE\ldots]$ in place $p_{k''}$ (the product $\otimes$ means that for suitable numbers $N_v$, $N_D$, and $N_J$ the sequences are glued together). Thus,

$$\Pi_{NV,ND,NJ} = [V_1|V_2|V_3|\ldots|V_{Nv}||\mathbf{E}|\mathbf{E}|\mathbf{E}|\ldots] \otimes [D_1|D_2|D_3|\ldots|D_{ND}||\mathbf{E}|\mathbf{E}|\mathbf{E}|\ldots]$$
$$\otimes [J_1|J_2|J_3|\ldots|J_{NJ}||\mathbf{E}|\mathbf{E}|\mathbf{E}|\ldots], \tag{52}$$

($\in$ object $\Pi$; $\Pi$ is the object of $V \otimes D \otimes J$ with suitable $N_v$, $N_D$, and $N_J$). In the above description, '$V_a \otimes D_b \otimes J_c$' (a, b, c are appropriate non-negative integers) is for example an element (a Cartesian product) of the tensor $V \otimes D \otimes J$. Here, we introduce the rule that a sequence of infinite tails of 'E's ($||EEE\ldots$) be written automatically for any product in this model and disregarded when we discuss attributes such as the colimit (or tensor product), in practical theorems in category theory. Note that in category theory, the tensor product is viewed as a sort of colimit of objects V, D, and J where the forgetful functor acts on $V \otimes D \otimes J$ so that the initial and terminal objects are ignored. Hence, we have the following notation; for appropriate numbers $N_v$, $N_D$, and $N_J$,

$$\Pi_{NV,ND,NJ} = [V_1|V_2|V_3|\ldots|V_{NV}||D_1|D_2|D_3|\ldots|D_{ND}||J_1|J_2|J_3|\ldots|J_{NJ}||\mathbf{E}|\mathbf{E}|\mathbf{E}|\ldots], \tag{53}$$

($\in$ object $\Pi$; (co)limit $V \otimes D \otimes J$ with an infinite tail of 'E's ($||EEE\ldots$). Adjustments to the index number are necessary after trimming; however, we disregard strict counting of the index number provided that the respective bases are recognized rigorously.

Suppose that one of the elements of colimit (or tensor) $\Pi$ is a Cartesian product as for {$V_3$, $D_1$, $J_2$} denoted by $[V_3 \otimes D_1 \otimes J_2||EE\ldots]$ in place $p_{k''}$, where an infinite tail '$||EEE\ldots$' accompanies automatically the element. Being an arbitrary product, it does not break the definition of a tensor product provided in that the number of recognized bases are accurate; in the above case, '$N_V = 4$', '$N_D = 3$', and '$N_J = 5$'. As an example, given '$V_3 = [G_1|C_2|T_3|A_4|C_5||\mathbf{E_6}|\mathbf{E_7}|\mathbf{E_8}|\ldots]$, $D_1 = [T_1|G_2|C_3||\mathbf{E_4}|\mathbf{E_5}|\mathbf{E_6}|\ldots]$, and $J_2 = [C_1|T_2|A_3|G_4||\mathbf{E_5}|\mathbf{E_6}|\mathbf{E_7}|\ldots]$', their Cartesian product is written {$V_3$, $D_1$, $J_2$} = $[V_3 \otimes D_1 \otimes J_2||EE\ldots]$ and

$\Pi_{4,3,5}$
$= [G_1|C_2|T_3|A_4|C_5||\mathbf{E_6}|\mathbf{E_7}|\mathbf{E_8}|\ldots] \otimes [T_1|G_2|C_3||\mathbf{E_4}|\mathbf{E_5}|\mathbf{E_6}|\ldots] \otimes [C_1|T_2|A_3|G_4||\mathbf{E_5}|\mathbf{E_6}|\mathbf{E_7}|\ldots] \tag{54}$
$= [(G_1|C_2|T_3|A_4|C_5)(T_6|G_7|C_8)(C_9|T_{10}|A_{11}|G_{12})||\mathbf{E_{13}}|\mathbf{E_{14}}|\mathbf{E_{15}}|\ldots].$

Here, the infinite tail of 'E's (||EEE. . .) is automatically placed outside of the scope of the (co)limit (or Cartesian product). The projection from object V×D×J to that of V×D, D×J, V, D, and J are describable in accordance with the conventional definition of a category.

[Fig 12(A)]

These manipulations are interpreted as the result of the following operation via the morphism for the combination morphism $f_{Int}\{, \} = \{\} \cup \{\} = \{|\}$. Specifically, for the state '$\{D_j\}(+)\{J_k\}$ where (+) means the independent existence (or insertion) in the same place (in place $p_{k1}$),

$$f_{Int}\{D_j, J_k\} = \{D_j\} \cup \{J_k\} = \{[D_j||EE..]\} \cup \{[J_k||EE...]\}$$
$$= [\{D_j|J_k\}||EE...] = [\{D_jJ_k\}||EE...] \text{ (in place } p_{k2}) \text{ holds.} \tag{55}$$

Similarly, for the state '$\{V_i\}(+)\{D_jJ_k\}$ where (+) means the independent existence (or inclusion) in the same place,

$f_{Int}\{V_i, D_jJ_k\} = \{V_i\}\cup\{D_jJ_k\}: \{[V_i||EE...]\}\cup\{[D_jJ_k||EE...]\}$

$= [\{V_i|D_jJ_k\}||EE...] = [\{V_iD_jJ_k\}||EE...]$ in place $p_{k3}$ holds.

By iterating the map of this morphisms $f_{Int}\{, \}$, the regions V, D, and J can be integrated into a unique sequence giving

$$[V_iD_jJ_k||EE...]. \tag{56}$$

[Fig 12(B)]

Note that because we regard object T as an object of tensor product V⊗D⊗J, simple sequences such as Eq (56) are not an element of object T but an element of object D (a simple base sequence with an infinite tail of 'E's). There may occur an instance for which an inverse of morphism $f_{spl}\{, \}$ for the splitting procedure is considered, The splitting morphism $f_{spl}\{, \} = \{\}$ ((+)) $\{\} = \{\} (+) \{\}$ can be defined in accordance with Eq (29) with recognized tokens all included in the same place $p_k$ (hence, in this example, we can simply replace '((+))' by '(+)'); e.g., for token $V_iD_jJ_k$ (in place $p_{k3}$), $f_{spl}\{V_iD_j, J_k\} = \{V_iD_j//J_k\} = [\{V_iD_j\}||EE...]$ (+) $[\{J_k\}||EE...]$ $= [V_iD_j||EE...]$ (+) $[J_k||EE...]$ (in place $p_k$). Because '$f_{spl}\{D_j,J_k\} = \{D_j//J_k\} = [\{D_j\}||EE..]$ (+) $[\{J_k\}||EE...] = [D_j||EE..]$ (+) $[J_k||EE...]$ (in place $p_{k1}$)', the following holds,

$$\{\{V_i//D_j\}//J_k\} = [V_i||EE...](+)[D_j||EE..](+)[J_k||EE...] \text{ in the same place } p_k. \tag{57}$$

Note that the associative law holds because the result is equivalent,

$$\{\{V_i//D_j\}//J_k\} = [\{V_i\}||EE...](+)[\{D_j\}||EE...](+)[\{J_k\}||EE...] = \{V_i//\{D_j//J_k\}\}. \tag{58}$$

In general, for morphisms $f_{Int}\{, \}$ and $f_{spl}\{, \}$, the associative law holds,

$$\{\{D_j\} \cup \{D_k\}\} \cup \{D_l\} = \{D_j|D_k|D_l\} = \{\{D_j\} \cup \{\{D_k\} \cup \{D_l\}\}, \tag{59}$$

$$\{\{D_j//D_k\}//D_l\} = \{D_j//\{D_k//D_l\}\}, \tag{60}$$

where j, k, and l are any appropriate non-negative integers. A category relation involving them is displayed in **Fig 12(B)**.

Here, relationships between the morphisms are drawn although the infinite tails of 'E's are omitted in the display.

## §3−13. Concerning an operation that perform indels of DNA sequences and its notation

For problems with mismatches in the index numbers (**#remark 2**) for the respective places $p_k$ ($k$ = 1,2,. . .), almost all coordinates of $D_{(k)j}$ or $R_{(k')j}$ and $Rs_{(k)j}$ seem to be simple; there may

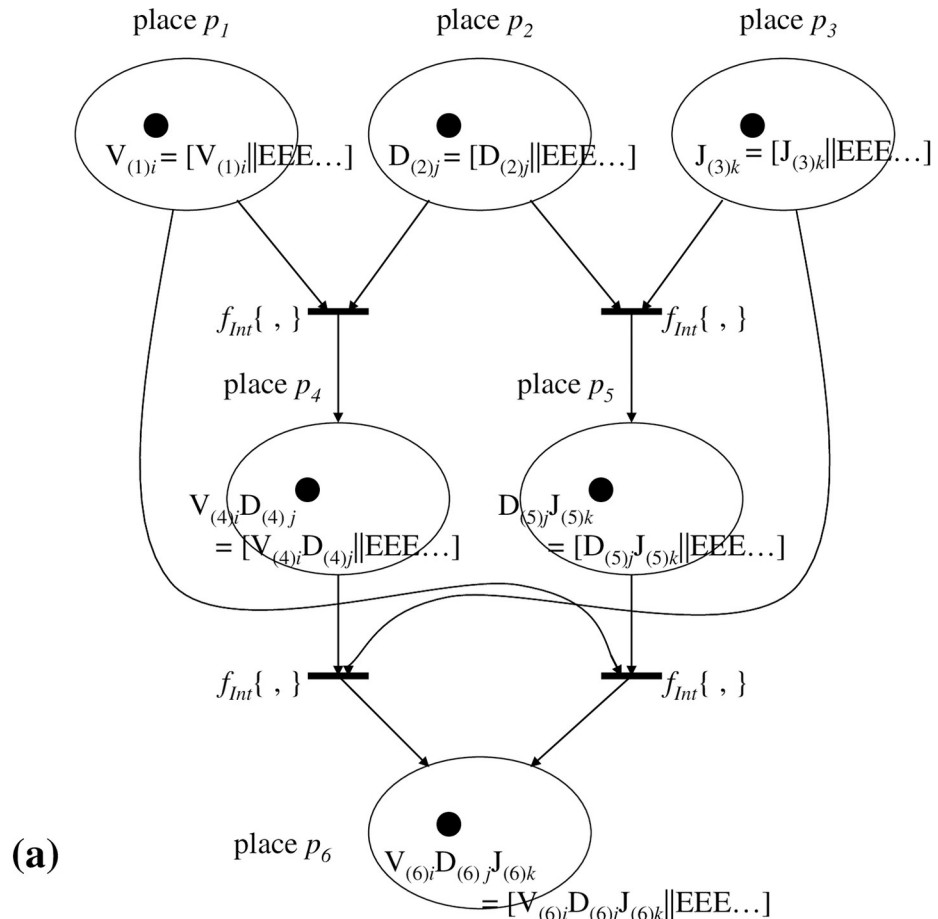

**(a)**

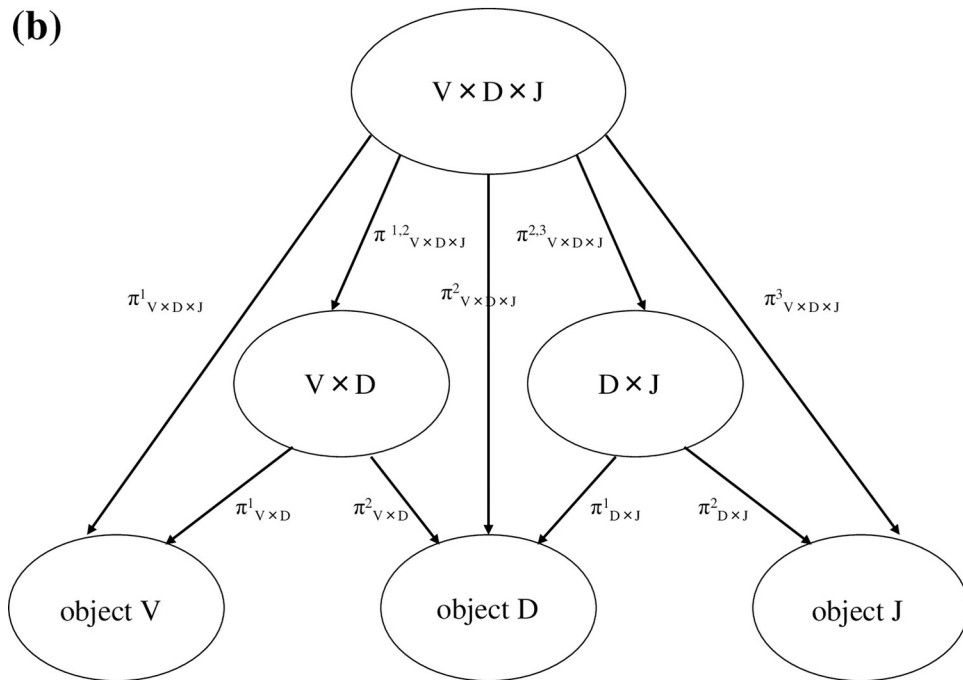

**(b)**

**Fig 12. Illustrations of a procedure for the recombination of an antibody using an EPN and conventional category theory.** (a). For reference, a procedure for the recombination of an antibody using compartments from segments V, D, and J is drawn as an EPN. In the present article, we mainly assume a 'state machine' where any transition has only a single input place and a single output place. However, this EPN does not comply with this rule. Using the map $f_{Int}\{,\}$ for compartments of the V, D, and J segments, the recombination of an antibody may be treated automatically if transition $f_{Int}\{,\}$ is free firing. Here, $V_i \times D_j \times J_k$ (or $V_i \otimes D_j \otimes J_k$) is assumed to be a simple combination of V, D, and J. For the unique product $V_i \times D_j \times J_k$, there exist multiple pathways that commute in the diagram although actually, there exists a majority or minority between them owing to some biological mechanism. (b). In the diagram, $V \times D \times J$ is composed as a limit of V, D, J from which the projection to $V \times D$, $D \times J$, V, D, and J are describable in accordance with conventional definitions of category theory. For example, $\pi^{1,2}{}_{V \times D \times J}$ (projection from object $V \times D \times J$ yields object $V \times D$, etc). In this regard, we disregard any infinite tail of 'E's ($||EEE...$) of each DNA sequences. In the dual diagram, $V \times D \times J$ is composed as a colimit of V, D, J where the direction of all projections is inverted. In the diagrams, the symbol for morphism (transition) '$-|\rightarrow$' is drawn as an ordinal arrow '$\rightarrow$' so as to be interpreted in the context of conventional category theory. For the composition of this EPN model, the morphism with transition | is also possible.

often occur instances for which the products of operations such as $D_{(k)l}$, $R_{(k')l}$, and $Rs_{(k'')m}$, $D_{(k)m}$, $R_{(k')m}$, and $Rs_{(k'')m}$ may be regarded as being included in the same place $p_k$ in extreme cases where all tokens are indexed with the same natural number 'k' as (k). Hence, the problem (**#remark 2**) does not matter so much. This circumstance seems to contribute to simplifications as exemplified in the following scenario.

With the developed notation for the recombination of DNA sequences, the following rule may be established; let $D_{(k)a}$ and $D_{(k)b}$ (in place $p_k$) be a certain finite DNA (or RNA; in this regard the place for DNA and that for RNA may be distinct with no specific use) sequence such as '$D_{(k)a} = [A_1|T_2|C_3|G_4]$' and '$D_{(k)b} = [G_1|C_2|T_3|A_4]$ (in place $p_k$)', and put '$D_{(k)j} = [G|C|T|A|C\{\mathbf{D_{(k)a}}\}\{\mathbf{D_{(k)b}}\}C|T|A|G||\mathbf{E}|\mathbf{E}|\mathbf{E}|...]$ in place $p_{k1}$'. In fact, the specific content is '$D_{(k)j} = [G_1|C_2|T_3|A_4|C_5\{\mathbf{A_6|T_7|C_8|G_9}\}\{\mathbf{G_{10}|C_{11}|T_{12}|A_{13}}\}C_{14}|T_{15}|A_{16}|G_{17}||\mathbf{E_{18}|E_{19}|E_{20}}|...]$ (in place $p_{k1}$)'. The morphism $f_{excl}\{//\}$ is applied to $D_{(k)j}$,

$f_{excl}\{\}$: $[G|C|T|A|C//\{\mathbf{D_{(k)a}}\}//\{\mathbf{D_{(k)b}}\}C|T|A|G||\mathbf{E}|\mathbf{E}|\mathbf{E}|...]$

$= [G_1|C_2|T_3|A_4|C_5//\{\mathbf{A_6|T_7|C_8|G_9}\}//\{\mathbf{D_{(k)b}}\}C_{14}|T_{15}|A_{16}|G_{17}||\mathbf{E_{18}|E_{19}|E_{20}}|...]$.

$= [(G_1|C_2|T_3|A_4|C_5)\{\mathbf{D_{(k)b}}\}(C_{10}|T_{11}|A_{12}|G_{13})||\mathbf{E_{14}|E_{15}|E_{16}}|...]$

$$(+)[\{\mathbf{A_1|T_2|C_3|G_4}\}||EE...] \text{ (in place } p_k), \tag{61}$$

$$= [(G_1|C_2|T_3|A_4|C_5)\{\mathbf{D_{(k)b}}\}(C_{10}|T_{11}|A_{12}|G_{13})||\mathbf{E_{14}|E_{15}|E_{16}}|...](+)[\{\mathbf{D_{(k)a}}\}||EE...]. \tag{62}$$

This type of manipulation may be of use to exclude parts of the DNA or RNA sequences. Conversely, for inclusion, the morphism $f_{inc}$ performs the following manipulation; $f_{inc}$: $[||||\{\}||||\mathbf{E}|\mathbf{E}||...]$. For example, $f_{inc}\{\mathbf{D_{(k)a}}\}$ yields

$$\{D_{(k)j}\} = [G|C|T|A|C\{\}C|T|A|G||\mathbf{E}|\mathbf{E}|\mathbf{E}|...] \text{ (in place } p_k),$$
$$= [G|C|T|A|C\{\mathbf{D_{(k)a}}\}C|T|A|G||E)E|E|...] \text{ (in place } p_k). \tag{63}$$

For both morphisms $f_{excl}$ and $f_{inc}$, the associative law holds.

## §3−14. Supplementary material concerning the concepts from the aspect of category theory

For all manipulations described above, the identity morphism performs nothing on any token. Thereby, the morphisms associated with combination, splitting, and insertion are considered to satisfy the postulate within category theory because associativity and the existence of an identity is met [8, 9]. For the above example, if the morphisms $f_{Int}$, $f_{spl}$, $f_{inc}$, and $f_{excl}$ are defined at the same place $p_k$ (this may be discriminated/separated as 'place $p_{k1}$, place $p_{k2}$, place $p_{k3}$...' when needed) and may be expressed in a single place $p_k$, then all manipulations are performed

in the same place $p_k$. For genetic coding, almost all tokens at their most complex are combinations of the five bases (C, A, T, G, E) or (C, A, U, G, E), and moreover are based on the operations of the group W. That is, the operation of a morphism, $f_{Int}$, $f_{spl}$, $f_{inc}$ or $f_{excl}$ defined at place $p_k$ on tokens such as $D_{(k)j}$, $R_{(k')j}$, and $Rs_{(k'')j}$ other than the double-stranded DNA $[D_{(k)j}/D_{(k)j}^\dagger]$ and the amino-acid sequences (proteins) $Pr_{(k)j}$ is considered possible only at a unique place (e.g., place $p_k$). We suppose that there is no contradiction stemming from differences in order and combinations of bases.

Although the treatments may have unknown difficulties that need, at least, further improvements, there is the potentiality that recombination of antibodies can be expressed as a form of colimit (tensor product) that may be linked to abstract algebraic products directly.

Generally speaking, in prokaryotic cells, because the bases are transcribed entirely, we believe that by using the appropriate notation the relationships based on **Fig 7** of [Ref. 6; S1 File] may be confirmed:

Naturally, although Category D is based on the above illustration, in eukaryotic cells, a natural transformation and right/left adjunction may not always exist when splicing occurs. Moreover, if indels of bases occur or a more complex mechanism occurs, a natural transformation may not be adequately definable although for that, an exact proof is necessary.

In following the conventional concept of category, we note that the 'natural transformation' is one of its important concepts. As an example, suppose 'morphism $B_{(k)(j\rightarrow l)}$: $D_{(k)j}\rightarrow D_{(k)l}$' (the operator that changes the sequence of DNA/RNA in place $p_k$) and 'morphism $B_{(k')(j\rightarrow l)}$: '$R_{(k')j}\rightarrow R_{(k')l}$' (exactly, $R_{(k')j}^\dagger\rightarrow R_{(k')l}^\dagger$) (the sequence of RNA is defined in place $p_{k'}$) are given along with two functors F, G: Category C→D. Then, functor F functions as 'morphism $B_{(k)(j\rightarrow l)}\rightarrow$ morphism $F(B_{(k)(j\rightarrow l)})$'. Similarly, functor G functions as 'morphism $B_{(k)(j\rightarrow l)}\rightarrow$ morphism $G(B_{(k)(j\rightarrow l)})$ ($\in$ Category D)'.

Next, let α be a morphism $\alpha_{D(k)j}$: '$F(D_{(k)j})\rightarrow G(D_{(k)j})$', and put '$\alpha_{D(k)l}$: $F(D_{(k)l})\rightarrow G(D_{(k)l})$'; then the following relationship holds: $\alpha_{D(k)l} \circ F(B_{(k)(j\rightarrow l)}) = G(B_{(k)(j\rightarrow l)}) \circ \alpha_{D(k)j}$ [**Fig 13(A)**]. We call α the 'natural transformation' mapping 'α: F→G'.

We infer that there exists a natural transformation based on a functor that produces changes in the sequence such as $D_{(k)j}$ in prokaryotes for which all sequences of DNA are copied. Conversely, a natural transformation may not be definable for some eukaryotes for which splicing occurs, and there may be instances for which inverse morphisms (functors) are non-fireable (definable).

Alternatively, suppose that functor A transforms DNAs to RNAs, and functor B transforms RNAs to DNAs in a sort of reverse transcription of a 'retrovirus' in the EPN exemplifying a flow from a primary single-strand DNA to a complementary RNA [**Fig 14(A)**]. Here, the morphisms in the EPN are based on the definition given in **Fig 7** [Ref. 6]. Together, if we assume two categories C and D and functor A that maps 'Category C→D', and suppose functor B maps 'Category D→C' representing the reverse transcription of a retrovirus from RNAs to primary single-strand DNAs, then an adjoint functor can be defined [**Fig 13(B)**]. With definitions η: $id_c\rightarrow BA$, and ε: $AB\rightarrow id_D$, a relation for the left/right adjoint obtains in accordance with conventional category theory. However, the above explanation is ambiguous and there is scope for a more rigorous methodology and deeper investigation.

We verify the expression for the present EPN model in light of Yoneda's lemma. First, in accordance with the canonical central dogma, we consider a certain 'DNA product', denoted a "Primer-RNA-$D_{(k)j}$ (= Primer-RNA-$[E_0|T_1A_2C_3A_4G_5C_6\ldots T_{N-3}G_{N-2}A_{N-1}C_N \|E_{N+1}E_{N+2}E_{N+3}\ldots]$)-TATA box". It includes a single-strand sequence $D_{(k)j}$, which is the part to be translated, where the 'primer-RNA' is connected at the head, and a TATA box at the tail, which is necessary to transcribe from the primary DNA sequence $D_{(k)j}$ to $D_{(k)j}^\dagger$ (complement of $D_{(k)j}$). Using a more general form, suppose 'morphism $f_1$: $D_{(k)j}\rightarrow D_{(k)l}$', 'morphism $f_2$: $D_{(k)l}\rightarrow D_{(k)m}$',

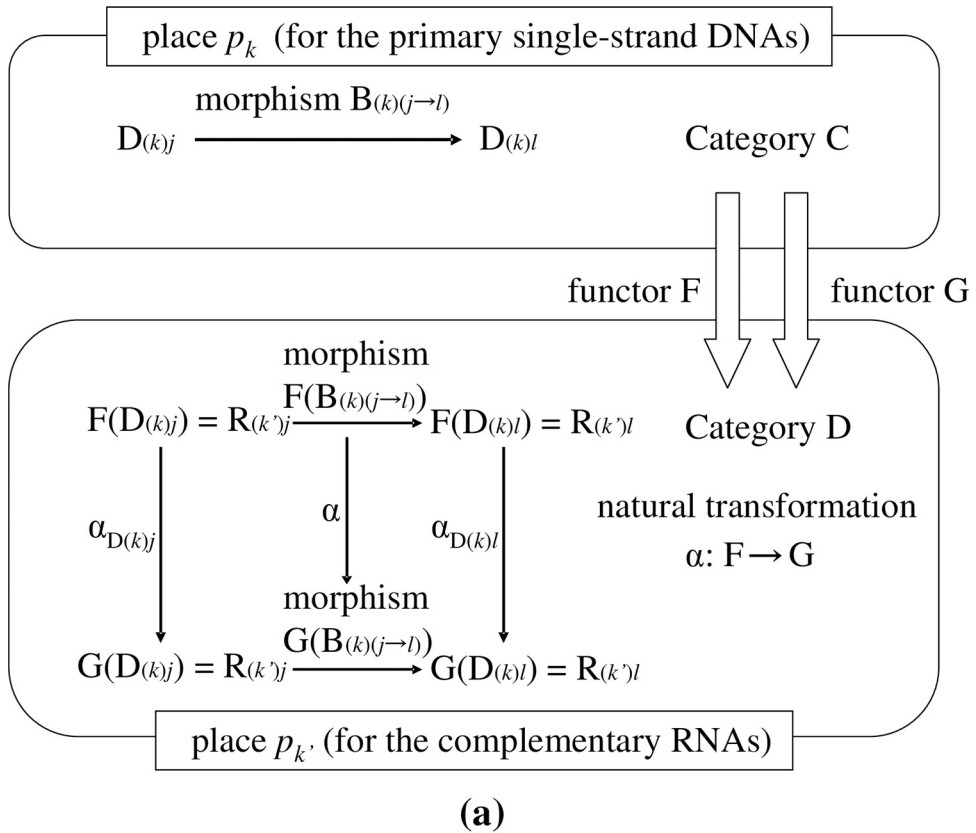

**(a)**

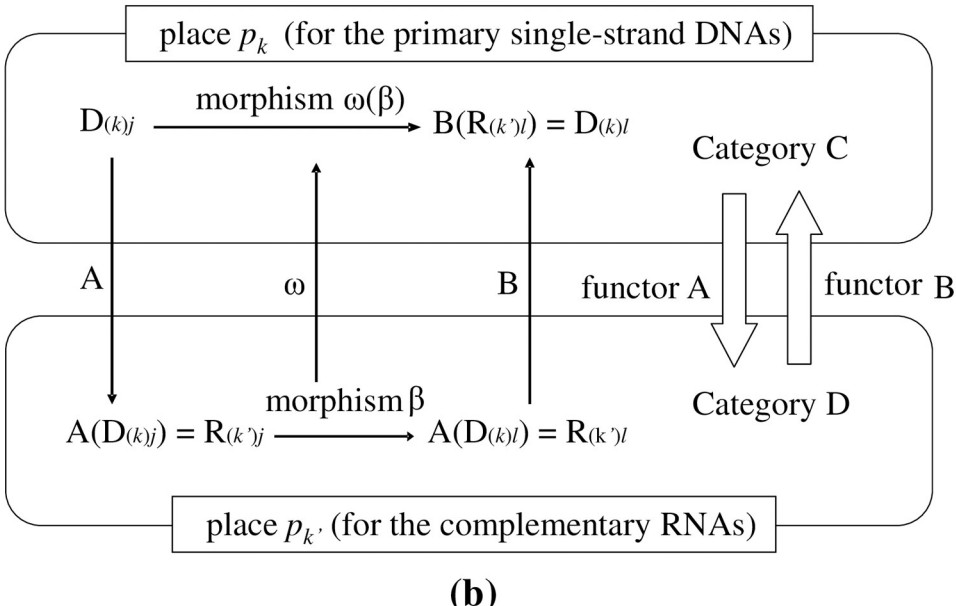

**(b)**

**Fig 13. Illustrations of a natural transformation and adjoint functors using DNAs and RNAs.** (a). DNA sequences $D_{(k)j}$, $D_{(k)l}$ and 'morphism $B_{(k)(j \to l)}$: $D_{(k)j} \to D_{(k)l}$' (the operator that changes the sequence of DNA/RNA in place $p_k$) along with 'morphism $B_{(k')(j \to l)}$: $R_{(k')j} \to R_{(k')l}$' (exactly, $R_{(k')j}^\dagger \to R_{(k')l}^\dagger$) (the sequence of RNA is defined in place $p_{k'}$), and two functors F, G: 'Category C→D' are drawn. They are transformed by α, $\alpha_{D(k)j}$: 'F($D_{(k)j}$)→G($D_{(k)j}$), and outputted '$\alpha_{D(k)l}$: F($D_{(k)l}$)→G($D_{(k)l}$)'. The procedure is commutative, $\alpha_{D(k)l} \circ F(B_{(k)(j \to l)}) = G(B_{(k)(j \to l)}) \circ \alpha_{D(k)j}$. The map α is called a

'natural transformation', defined as 'α: F→G'. Additionally, for the above formula, more detailed relationships hold; F $(D_{(k)j}) = R_{(k')j}$, $F(D_{(k)l}) = R_{(k')j}$, $G(D_{(k)j}) = R_{(k')j}$, and $G(D_{(k)j}) = R_{(k')l}$. (b). Suppose that functor A transforms DNAs to RNAs, and functor B transforms RNAs to DNAs as a reverse transcription of a 'retrovirus'. For two categories C (in place $p_k$) and D (in place $p_{k'}$), suppose functor A produces Category C→D and functor B produces Category D→C; then by functor A, $D_{(k)j}$ (∈Category C) is transformed into $A(D_{(k)j}) = R_{(k')j}$. Here, if there is a morphism β given as 'β: $R_{(k')}→R_{(k')l}$', by functor B, $R_{(k')l}$ is transformed as $B(R_{(k')l}) = D_{(k)l}$, and hence there exists morphism 'ω(β): $D_{(k)j}→D_{(k)l}$', where ω is called the adjoint functor.

and 'morphism $f_3$: $D_{(k)j}→D_{(k)m}$'. Then, using an indexed Category C that includes $D_{(k)j}$, $D_{(k)l}$, and $D_{(k)m}$, a triangular diagram in the little Category **C** can be constructed [see **Fig 15(A)**]. For Category **C**, if we stipulate that 'homset from a fixed object $D_{(k)j}$ to an arbitrary object $D_{(k)l}$' determines 'hom($D_{(k)j}$, $D_{(k)l}$)', then this forms a set for which 'morphism $f_2$: $D_{(k)l}→D_{(k)m}$' holds true [52]. Next, we consider the hom functor C($D_{(k)j}$, $D_{(k)l}$) that transforms Category **C** to **Set** (Category set). For brevity, if we write '$h^A \equiv$ C($D_{(k)j}$, $D_{(k)l}$)', there exists a natural transformation α with action 'α: $h^A$ ($\equiv$C($D_{(k)j}$, $D_{(k)l}$))→F'. The $h^A$ also performs the following transformation. $h^A$: 'hom($D_{(k)j}$, $D_{(k)l}$)→C($D_{(k)j}$, $D_{(k)l}$)', '$f_2$→C($D_{(k)j}$, $f_2$)', and 'hom($D_{(k)j}$, $D_{(k)m}$)→C($D_{(k)j}$, $D_{(k)m}$)'. For the above, additional relationships follow, specifically, the natural transformations '$α_X$: C($D_{(k)j}$, $D_{(k)l}$)→F($D_{(k)l}$)', '$α_Y$: C($D_{(k)j}$, $D_{(k)m}$)→F($D_{(k)m}$)', and 'α: C($D_{(k)j}$, $f_2$)→F($f_2$)' [**Fig 15 (A)**]. At this moment, we regard 'α: $h^A$ ($\equiv$C($D_{(k)j}$, $D_{(k)l}$))→F' as forming a functor Category [**C**, **Set**] where $h^A$ and F are objects with α a morphism (natural transformation), and we establish the functor category [**C**, **Set**] on which α acts, denoted by [**C**, **Set**]($h^A$, F). In this case, we refrain from using RNA sequences because whether Category C is composable is ambiguous because Category C is assumed to include DNA sequences $D_{(k)j}$ and $D_{(k)j}^†$, and RNA sequence $R_{(k')j}^†$ in accordance with the canonical central dogma. In other words, if a combination of bases T and U occurs when we compose a category, it may be considered non-essential for instances for which $D_{(k)j}$ = [EA**T**GE**U**TCG**U**EEE...] and that may cause contextual confusion. Hence, to avoid such instances, we confine our focus to only DNA.

For arbitrary objects $D_{(k)j}$, $D_{(k)l}$, and $D_{(k)m}$ in Category C, arbitrary functor 'f: **C**→**Set**', and functor category [**C**, **Set**], we have Yoneda's lemma using DNA sequences [8, 9, 52];

$$[\mathbf{C},\ \mathbf{Set}](h^A,\ F) \cong F(D_{(k)j}) \qquad (\cong\ \text{means isomorphism}). \tag{64}$$

Moreover, according to conventional wisdom, for arbitrary objects $D_{(k)j}$, $D_{(k)l}$, and $D_{(k)m}$ in Category C, and arbitrary functor 'f: **C**→**Set**', Yoneda's embedding holds; that is,

$$[\mathbf{C},\ \mathbf{Set}](C(D_{(k)j},\ D_{(k)m}),\ C(D_{(k)l},\ D_{(k)m})) \cong C(D_{(k)l},\ D_{(k)j}). \tag{65}$$

Here, natural transformations α': C($D_{(k)j}$, $D_{(k)m}$)→C($D_{(k)l}$, $D_{(k)m}$) and α": $D_{(k)l}→D_{(k)j}$ are assumed, and isomorphism α'≅α" holds. From Eq (65), transformations between one DNA sequence and other different structures may be embedded in a nested structure or de-embedded from a nested structure. With the repetitive use of this principle, nesting or de-nesting of structures from living creatures including human beings may be possible.

[Fig 15(B)]

By adding more explanation, according to "Yoneda embedding", a mere object can be regarded as a functor, and a mere morphism may be regarded as a natural transformation implying that a function of a specific compartment and that of a higher organ can be defined according to their levels. That is, the function of organs, tissues, cells, as well as intercellular activities and genuine genetic activities may be described in a unique embedded diagram [**Fig 15(B)**] as a hierarchical-cluster formulation (**#remark 3**) although this is only a primitive idea and a rigorous investigation is necessary.

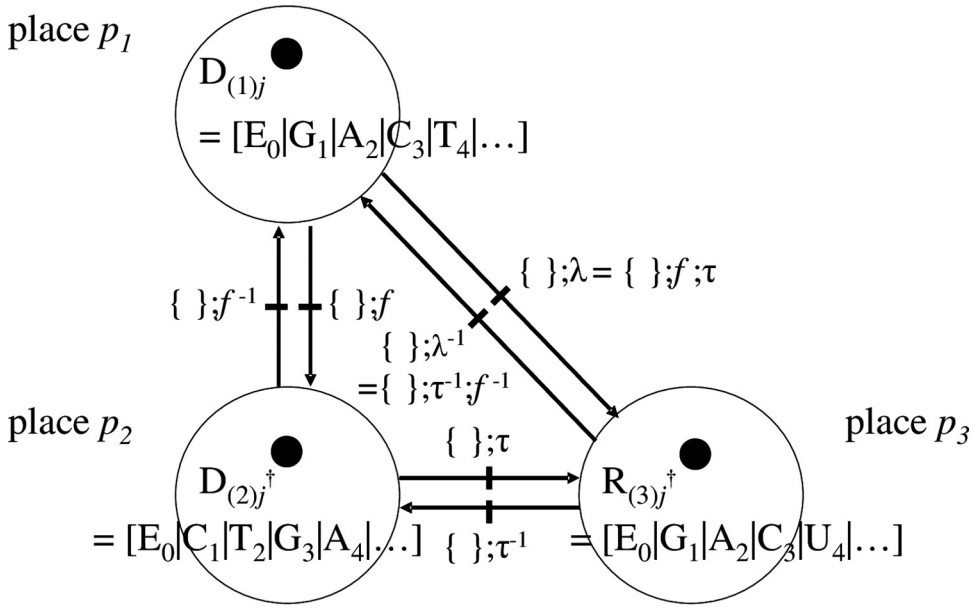

**(a)**

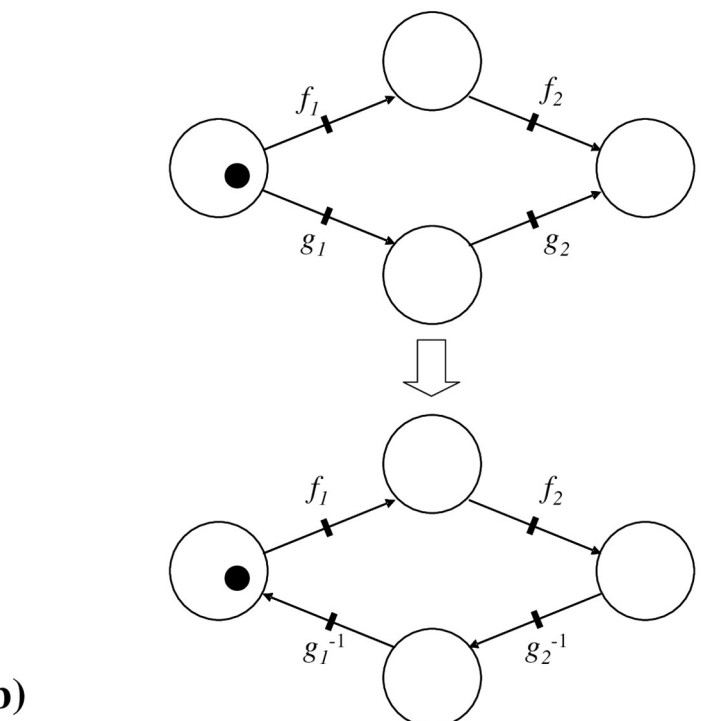

**(b)**

**Fig 14. Two supplementary illustrations for EPNs.** (a). EPN illustration for the flow from a primary single-strand DNA to a complementary RNA is described. Noting that 'morphism $\lambda$ = f;$\tau$' and 'morphism '$\lambda^{-1}$ = $\tau^{-1}$;f$^{-1}$' are true, we demonstrate the diagram is commutative. (b). For an upper diagram-like EPN, the closure of the relationship between $f_1;f_2$ and $g_1;g_2$ to commutativity ($f_1;f_2 = g_1;g_2$) may produce the same token in the output place (where one of each is fireable in the diagram). If $g_1^{-1}$ and $g_2^{-1}$ are the duals of $g_1$ and $g_2$, respectively, scope exist where this commutativity is expressible alternatively in a lower diagram-like EPN. Here, '$f_1;f_2;g_2^{-1};g_1^{-1}$ = Id' holds provided that the above commutativity is not broken. At this stage, its precise definition is tentative and to avoid inaccuracies, we add no further remarks. Note that examples '$f_1 = f_2 = g_1 = g_2 =$ Id', '$g_1^{-1} = f_1$', '$g_2^{-1} = f_2$', and '$g_2^{-1} = f_1$', '$g_1 = g_1^{-1} = f_2 =$ Id', do not satisfy ordinal 'commutativity' in a diagram-like EPN.

The characteristics of Yoneda's embedding may be translated into various words. For example, for a living body, it may be found peculiar that each organ may behave more selfishly than its entire body, individual tissue more selfishly than their organs, cells more selfishly than their tissue, DNA having more precedence than its cells, and individual compartments of DNA (cistrons) having more precedence over the full length of a DNA strand. This may be an alternative expression of C. R. Dawkins's concept of the 'selfish gene' [53] in a category using a hierarchical-cluster formulation.

We turn our attention next to the concept of 'Kan extensions'. An initial part of the procedure for the central dogma may be written in the form of the 'left Kan extension' [9]. In viewing [**Fig 16(A) and 16(B)**], which were obtained from a simplification of **Fig 11(B)** or **Fig 14 (A)**, the morphism that offers the complimentary sequence from the primary single-strand DNA '$D_{(k)j} \rightarrow D_{(k)j}{}^{\dagger}$' denoted by '$f_{(D(k)j \rightarrow D(k)j\dagger)}$' may be a functor F. Together with the functor, the morphism that produces this change from DNA to RNA '$D_{(k)j} \rightarrow R_{(k')j}{}^{\dagger}$' written as '$f_{(D(k)j \rightarrow R(k')j}{}^{\dagger})}$' may be a functor G. Thus, the procedure from the complementary DNA to the subsequent complimentary RNA '$D_{(k}{}^{\dagger}{}_{)j}{}^{\dagger} \rightarrow R_{(k')j}{}^{\dagger}$' can be expressed in two ways: one a left Kan extension, which is the pair $<F^{\dagger}G, \eta>$, with $\eta$: $G \rightarrow F^{\dagger}G°F$ (= F;G;F$^{\dagger}$) and the other a right Kan extension, which is the pair $<F^{\ddagger}G, \varepsilon>$, with $\varepsilon$: $F^{\ddagger}G°F$ (= F;G;F$^{\ddagger}$) $\rightarrow$ G. The left Kan extension of G along F is also denoted $Lan_F G$ (= $Lan_{f(D(k)j \rightarrow D(k}{}^{\dagger}{}_{)j}{}^{\dagger})} f_{(D(k)j \rightarrow R(k')j}{}^{\dagger})}$). Moreover, the right Kan extension, which is the dual of the left Kan extension of G along F where arrows of natural transformation are inverted, is also denoted by $Ran_F G$ (= $Ran_{f(D(k)j \rightarrow D(k}{}^{\dagger}{}_{)j}{}^{\dagger})} f_{(D(k)j \rightarrow R(k')j}{}^{\dagger})}$). Furthermore, the concept of the left Kan extension can be exchanged for the right/left adjunction: F$^{\dagger}$ of the left Kan extension is formally written 'F$^{\dagger}\dashv$ F', which reads F$^{\dagger}$ is the left adjunct to F. Similarly, F$^{\ddagger}$ of the right Kan extension is formally written 'F $\vdash$F$^{\ddagger}$', which reads F$^{\ddagger}$ is the right adjunct to F [9].

Similarly, the flow of the former part of '$R_{(k')j}{}^{\dagger} \rightarrow Rs_{(k'')j}{}^{\dagger}$' (the change from complimentary RNA in place $p_{k'}$ to that of pre-mRNA in place $p_{k''}$), and the latter part of that (the change from pre-mRNA to mRNA) may also be seen as the left and right Kan extensions with further attributes, the details of which we refrain from discussing at this stage.

This is only one of the potential applications for natural transformations, the left/right adjunct, Yoneda's lemma, and left/right Kan extensions. Although limited within parts and with details omitted, these concepts with future research are expected to play a valuable role in our model as well. As seen in the above, our hybrid model expresses aspects of the canonical central dogma. In particular, infinite $m$-tuple places are of practical use in composing the model, and the multiple placements of $\phi$ as indicated by $\{\phi\}p...$, based on Eqs (12) and (13) are assumed implicitly.

## §4. Results

The present hybrid model comprises: 1) elements of a group regarded as a token of PN; 2) objects of a category that are replaced by places of PN——as a result, broader usability is given to the conventional PN along with more logical consistency so that the canonical central

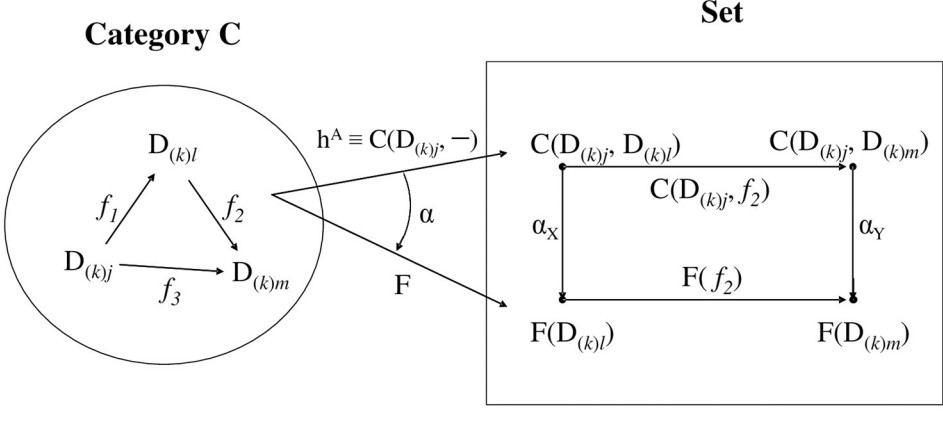

**(a)**

**(b)**

**Fig 15. Exemplified illustrations for Yoneda embedding using a biological system organized hierarchically.** (a). Given 'morphisms $f_1$: $D_{(k)j} \to D_{(k)l}$', '$f_2$: $D_{(k)l} \to D_{(k)m}$', and 'morphism $f_3$: $D_{(k)j} \to D_{(k)m}$', an indexed Category C that includes $D_{(k)j}$, $D_{(k)l}$ and $D_{(k)m}$, and the triangular diagram in the little Category C are drawn. There, 'homset from a fixed object $D_{(k)j}$ to an arbitrary object $D_{(k)l}$', denoted 'hom($D_{(k)j}$, $D_{(k)l}$)', forms a set where 'morphism $f_2$: $D_{(k)l} \to D_{(k)m}$ is assumed. Next, using 'hom functor C($D_{(k)j}$, −)', a natural transformation α that acts as 'α: $h^A$ ($\equiv$C($D_{(k)j}$, $D_{(k)l}$))→F' is definable. Then $h^A$ also performs the following transformation; $h^A$: 'hom($D_{(k)j}$, $D_{(k)l}$)→C($D_{(k)j}$, $D_{(k)l}$)', '$f_2$→C($D_{(k)j}$, $f_2$)', 'hom($D_{(k)j}$, $D_{(k)m}$)→C($D_{(k)j}$, $D_{(k)m}$)'. The following relationships also exist: natural transformation 'α$_X$: C($D_{(k)j}$, $D_{(k)l}$)→ F($D_{(k)l}$)', 'α$_Y$: C($D_{(k)j}$, $D_{(k)m}$)→F($D_{(k)m}$)', and 'α: C($D_{(k)j}$, $f_2$)→F($f_2$)'. (b). With reference to "Yoneda embedding", the

function of organs, tissues, cells, intercellular activity, and genuine genetic activities can be described within the hierarchical-cluster formulation. There, an embedding into a nested structure from one or a de-embedding from a nested structure from another is considered also composable with the appropriate adjustments. This is because an object can be regarded as functor, and a morphism can be regarded as a natural transformation, and vice versa. In the diagram, system A includes multiple individuals such as $^1A_1$ (object) and $^1A_2$ (object), and morphism $f_1$. However, in view of Yoneda embedding, $^2A_{11}$ (object) and $^2A_{12}$ (object) are functors and morphism $f_{11}$ a natural transformation. Together, $^2A_{11}$ and $^2A_{12}$ include object cells $^3A_{111}$ and $^3A_{112}$, and so on. One interaction between $^3A_{111}$ and $^3A_{112}$ may be written as a morphism (functor) $f_{111}$ and so on. The cells $^3A_{111}$ and $^3A_{112}$ have DNAs, here denoted $^4D_{1111}$ and $^4D_{1112}$, and so on. The morphism between $^4D_{1111}$ and $^4D_{1112}$ such as $B_{(k)(j\rightarrow l)}$ was illustrated before as an operator that converted one DNA sequence $D_{(k)j}$ into another $D_{(k)l}$ using $\omega_1, \omega_2, \omega_3,\ldots$, and these morphisms may also be viewed as functors from the Yoneda embedding. In addition, more detailed embeddings or de-embeddings from $^4D_{1111}$, $^4D_{1112},\ldots$ where $^4D_{1111}, ^4D_{1112},\ldots$ may be viewed as functors at the extremes. However, this diagram provides only a rough outline of a primitive idea for the application of Yoneda's lemma using the material of the present article.

dogma is described; 3) as a useful visual tool in future genomics, a square pole (rather than a pentagonal pole) is devised that enables the unique identification of single-strand DNA sequences solely from the shape of their respective polygonal lines, and vice versa—additionally, a complementary sequence (parts) may be easily identified; 4) a novel use of creation/annihilation morphisms that may generate/erase tokens freely via an identity token, that is usable to adjust the number of tokens; 5) the conversion between a single-strand DNA sequence and that of RNA is performable simply via the Cartesian product '$Z_5{\times}Z_2{\times}\ldots$' within the group-theoretical concept; 6) the somatic recombination (VDJ recombination) for antibodies concretely expressed in categorical concepts, specifically, the (co-)limit; 7) the novel 'identity protein $\Delta$', translated from a codon (a triplet) of identity bases 'EEE', that provides a conceptual advance on our previous presentation of the canonical central dogma to a more elevated level (practically, to protein synthesis); 8) the incidence matrix-like matrix A that includes operators (morphisms) as coordinates so that with reference to Naganuma's 'Theory of operation matrix' proposal only multiplication of A can yield a dynamic flow central to the canonical central dogma, and 9) basic topics regarding the canonical central dogma being displayed concretely using conventional category theory concepts such as 'adjoint', 'adjoint functor', 'natural transformation', 'Yoneda's lemma', and 'Kan extension' that may help readers to comprehend the model and to serve advocates in future proceedings of this field. These ideas also provide more advanced tools that may be added to our previous model concerning nucleic-acid-base sequences.

## §5. Discussion

A hybrid structure of an extended PN and a group (as one of the groups and their related products) was demonstrated using simple examples to broaden its potential application to dynamic phenomena. In summary, from a semantic viewpoint, the highlights of our model are the following:

i) an extended PN (EPN) structure that is closed with respect to the category postulates, in which the composition of morphisms and an identity morphism are defined, was displayed as exhibiting dynamic phenomena modeled as a stream of countable tokens that reflected physical attributes/states in the world;

ii) abstract algebraic characteristics were displayed visually using tokens based on a group or related products having inverse elements (monoidal structure with some inverse elements);

iii) annihilation/creation morphisms were introduced to adjust the number of tokens/to broaden the reachability of the EPN;

iv) as a developed example using category concepts, the DNA/RNA sequence was expressed graphically as a 3-dimensional infinite polygonal line, a prototype of which was suggested in our previous report [Ref. 6] but was insufficiently developed and hence re-designed in a more

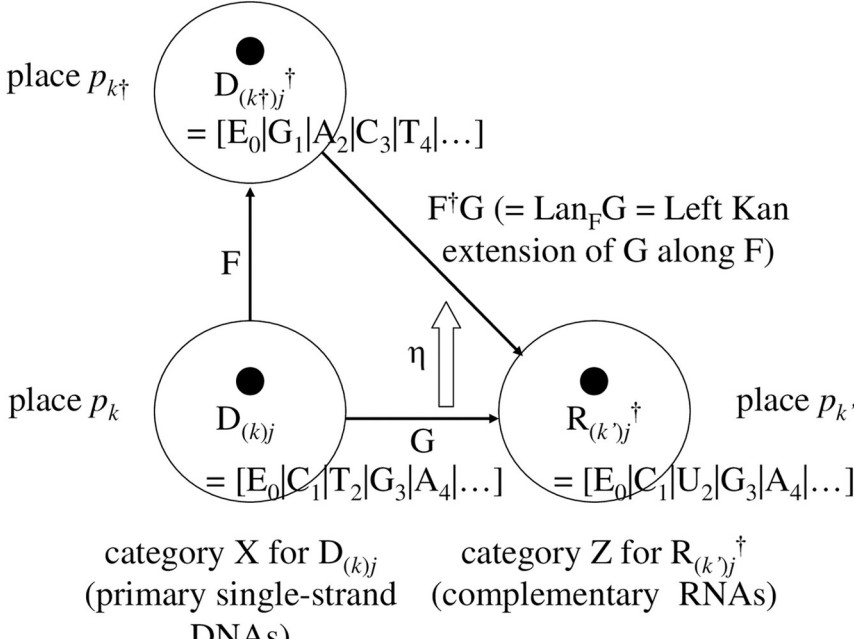

**(a)**

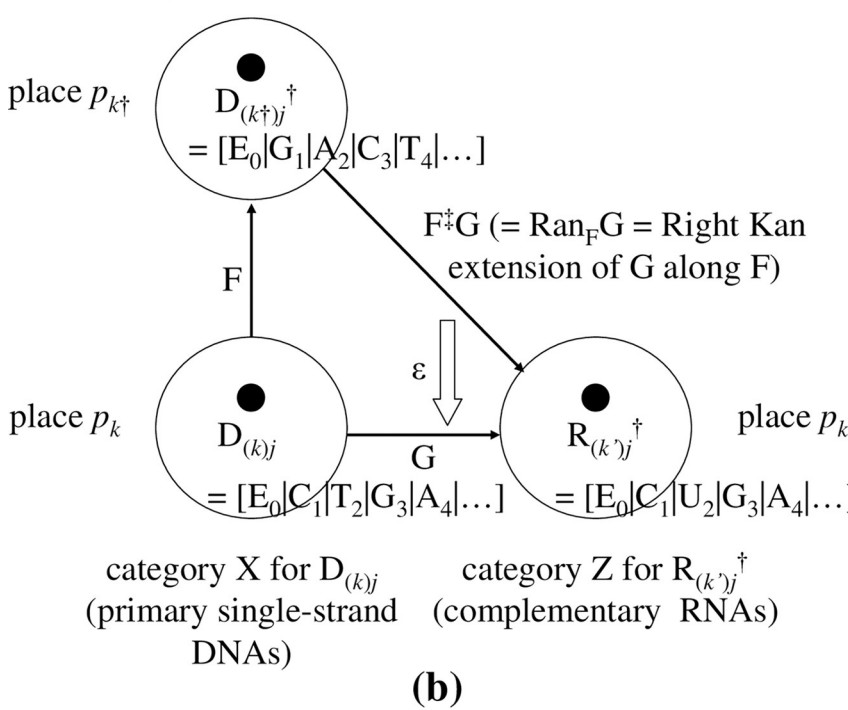

**(b)**

**Fig 16. Illustrations of the left and right Kan extensions using DNAs and RNAs.** (a). Let functor F be defined from Category X (for the primary single-strand DNA sequences; $D_{(k)j}$) to Category Y (for the complementary DNA sequences; $D_{(k}{}^{\dagger}{}_{)j}{}^{\dagger}$). Additionally, suppose functor G is defined from Category X (for '$D_{(k)j}$'s) to Category Z (for '$R_{(k')j}{}^{\dagger}$'s). Then, the procedure from a primary DNA to the subsequent complimentary RNA '$D_{(k)j} \rightarrow R_{(k')j}{}^{\dagger}$' is expressed in two ways; one is the left Kan extension, the pair $<F^{\dagger}G, \eta>$, with $\eta$: $G \rightarrow F^{\dagger}G{\circ}G$ (= F;G; $F^{\dagger}$). The left Kan extension of G along F is also denoted $Lan_F G$ (= $Lan_{f(D(k)j \rightarrow D(k}{}^{\dagger}{}_{)j}{}^{\dagger} f_{(D(k)j \rightarrow R(k')j}{}^{\dagger}_{)}$). (b). The right Kan extension of G along F is also denoted $Ran_F G$ (= $Ran_{f(D(k)j \rightarrow D(k}{}^{\dagger}{}_{)j}{}^{\dagger} f_{(D(k)j \rightarrow R(k')j}{}^{\dagger}_{)}$). Additionally, 'The pair $<F^{\ddagger}G, \varepsilon>$, with $\varepsilon$: $F^{\ddagger}G{\circ}F \rightarrow G$', is drawn.

comprehensive form. A method to visualize translations from a sequence of codons into a corresponding amino-acid sequence (protein) was also illustrated using the conventional codon table [**Fig 11(A)**]; and

v) the potential applicability of category theory is too general to be applied to concrete phenomenon but can be modified in a manner that treats in at least a more specified manner physical aspects slightly distinct from conventional ones.

In **§3**, if integration, insertion, division, deletion, and exclusion for tokens do not occur, as for example in **Fig 11(B)**, the EPN uses only $D_{(k)j}$, $R_{(k')j}$, $Rs_{(k'')j}$, and related bases. The process associated with the canonical central dogma from place $p_2$ to place $p_5$ in **Fig 11(B)** (from single-strand DNA to mRNA) may be expressible in a unique place $p$ where all morphisms such as $f_{(2 \rightarrow 3)}$, $f_{(3 \rightarrow 4)}$, and $f_{(4 \rightarrow 5)}$ are defined only within place $p_k$ (Note that $D_{(k)j}$, $R_{(k')j}$, and $Rs_{(k'')j}$ are products that are obtained by an operator such as $B_{(k)j}$ or $B_{(k)(j \rightarrow l)}$ accessorized with a constant $D_0$, as in Eq (33)). This was expressed in our previous work [Ref. 6]. Hence, the composition of the EPN model using multiple places and visualizing the flow of the canonical central dogma is one of the main objectives of the present article although they may be optional.

It may be peculiar, in the present article, that the binary compositions (operations) for two tokens such as $a_{(k)i}{}^{*}a_{(k)j}$ or $(a_{(k)i}{\bullet})^{*}(a_{(k)j}{\bullet})$ are not postulated although the composition of two tokens can be defined. Hence, another rule arises where another unique token may be derived from the composite of two tokens as an element of group or related products. Specifically, the composition of two tokens is non-operational because $D_{(k)j}$ and $D_{(k)l}$ (the sequence of DNA) are not operators in an exact meaning. Clearly, '$D_{(k)j} = D_0;B_{(k)(0 \rightarrow j)}$' and $B_{(k)j}$ (= $B_{(k)(0 \rightarrow j)}$) are similar products where the '$D_{(k)j}$'s are accessorized products of '$B_{(k)j}$'s that may be obtained by adding a constant $D_0$ on the head of '$B_{(k)j}$'s, i.e., '$D_0{\cdot}B_{(k)j}$'. These treatments among tokens would also be definable, and there would be room where more advanced definitions or compositions are performable; for example, Formula (55) may be formulated as a bi-functor or 2-category and more sophisticated systematization may be possible. Here, the changeable number of tokens may behave in a more flexible manner. On this possibility though, we refrained from saying more and thereby avoid uncharted waters. We believe a rigorous study is warranted in future research.

In the previous illustration, a simple formulation was intended to gain a more exhaustive treatment for the description of biological phenomena. For that purpose, despite a tentative trial, this article showed how the group and related concepts with the PN formalism can be hybridized in a simple manner, especially by regarding elements of the group as tokens of the PN in places that have corresponding group postulates. Here, two types of morphisms (actually a complex of transitions, arcs, and functions) are presented; one is for the inner morphism for an ordinal group operation (binary composition), and the other is for the trans-place morphism being indexed with a category-theoretical operation where an element of a group is transformed/transported from one place to another place. With this context, a group or category-theoretical operation may be viewed visually in the EPN formalism, in parallel, distributively, asynchronously or dynamically. While peculiar to some extent, the annihilation and creation morphisms were introduced to the mix. Here, the '$a_{(k)0} \leftrightarrow \phi$ conversion' seems to have similar

features of a bijection. This relationship is based on the uniqueness of the terminal object 1 and the initial object $\phi$ in category theory. However, conventionally, $\phi$ means the empty set [8, 9]. In our model, we gave $\phi$ the meaning of an implicit identity that may stem from an explicit identity $a_{(k)0}$, where it is permissible that $\phi$ is found freely in any place $p_k$. The $\phi$ is uncountable; therefore, $\phi+\phi+\phi+\ldots+\phi = \phi$ holds. That means that the infinite explicit identity $a_{(k)0}$ may be obtained via the creation morphism; e.g., '$\phi \rightarrow a_{(k)0}$' and 'two $\phi$' provide two $a_{(k)0}$s as in '$\phi\phi \rightarrow a_{(k)0}$ (+) $a_{(k)0}$' (in place $p_k$). Here, for convenience, the conversion between '$a_{(k)0} \leftrightarrow \phi$' is defined partially and locally in their respective places. In the genetics example (molecular biology) of [6], the relationship 'E$\leftrightarrow$()' is understood to have similar characteristics where the 'explicit E' corresponds to object '1', and the 'implicit ()' also corresponds to object '$\phi$'. However, in our model, () is taken to be countable in a duplicated manner so that () () $\ldots$() may be infinite, emphasizing the point that $\phi$ or () in the present report is different from ordinal empty '$\phi$' where expressions using multiple symbols like '$\phi\phi\phi\ldots$' are not used. We hope that, while these exact roles and characters are vague at this stage, this rule yields other as-yet-unknown conveniences.

However, it is expected that various sorts of group operations may at least be placed and performed in parallel, dynamically, and visually, preserving their respective postulates as groups or related products on a unique plane, and developments may be integrated into a unique form.

The example presented for DNA in **§3** is the simplest case that draws on the essential concepts based on the model we suggested in previous work [6]. Here, the previous static model for DNA was updated in a dynamic formulation with an exhaustive and slightly higher conceptual description, in which, with the further examples, natural transformation and right/left adjunction are introduced.

In **§3**, the vector-like Cartesian product was treated, especially within the hierarchical-cluster formulation preserving their asynchronous, in-parallel, dynamic characteristics. We believe these attempts highlight the potential for integration into (molecular) biology for this sophisticated model with its rigorously strict mathematics. Moreover, the EPN model brings the inclusion of incidence-like $m_k \times n_k$ matrices (e.g., A,B,...) as tokens in place $p_k$ for which a ring is defined and addition and multiplication are defined as index operations. There, the binary addition as an inner-place morphism, denoted '$f_k$:[{A}+{B}] = A+B', and similarly, multiplication, denoted '$f_k$:[{A}×{B}] = AB', may also be performable in the same place $p_k$. Provided that there exists a morphism $f_{(k \rightarrow l)}$: $m_k \times n_k$ matrix $A_k \rightarrow m_l \times n_l$ matrix $A_l$ from place $p_k$ to place $p_l$, this operation yields the transposition of matrix $A_k$ if '$m_l = n_k$, $n_l = m_k$' is also postulated. For another example, if '$m_k = n_k$' holds, and place $p_l$ includes only diagonalized $m_k \times m_k$ matrices, the morphism $f_{(k \rightarrow l)}$ means the diagonalization of the $m_k \times m_k$ matrix $A_k$ in place $p_k$ into a $m_l \times m_l$ matrix $A_l$ in place $p_l$. In summary, these may express a sequence of transformations among certain matrices in the EPN model although there remains a large scope for improvements. We believe that with some appropriate devices, the respective places in the EPN may hold group, ring, field, and other similar structures with inverse morphisms, and the formalism may lead us to more general, as-yet unknown perspectives in natural science with a complete exhaustive approach.

We envisage a potential utility of the expression for the left and right Kan extensions if a pair of functors for certain biological procedures is found [9]. If there are two arbitrary functors and objects (places) where the following relationships are satisfied on tokens $D_j$, $X_j$, and $Y_j$ in the flow of the canonical central dogma, such as $D_j \rightarrow X_j$ and $D_j \rightarrow Y_j$, there may exist a right Kan extension 'Ran$_{(Xj \rightarrow Yj)}$: $X_j \rightarrow Y_j$', and a left Kan extension 'Lan$_{(Xj \rightarrow Yj)}$'. There is the potential where unknown characteristics may be revealed as a product of left and right Kan extensions of two functors, although the details are unclear at this stage.

Finally, the use of unit '1' in Descartes's geometry may be one of the first applications of the binary operation of group where 'n;n;n;. . .;n = n;n = n' and 'n = n;n = n;n;n;. . .;n' are demonstrated multiple times so that the formula '$a^3$+2ab+3a/$b^2$' is adapted entirely in three dimensions as '$a^3$+2ab·1+(a/$b^2$)·$1^4$'. With Descartes' unit '1', 1 acts on element 1 to give 1 and also acts as the inverse element of 1.

Inevitably, at this initial stage of development, our proposal to incorporate category theory and EPN into the algebraic formulation of biology must be understood in the context of significant limitations. First, unavoidably, the complex systems or phenomena need, in general, very complex PN constructions, and sometimes they do not provide an appropriate systematization of the PN. With regard to (molecular) biology, it may be difficult or impossible to compose a well-defined and well-behaved model. The PN has a linear characteristic; however, non-linear phenomena are often observed. Hence, to what level this model is applicable is unclear. In the PN, matrices such as incidence matrices are often used. However, this tool needs substantially large weights that could produce other problems.

Second, the scope for employing the natural transformation and right/left adjunction seems to be ambiguous. In prokaryotic cells, the entire bases of DNA are transcribed into mRNA. This may reserve a certain order that leads to the reversibility of biological phenomena. If sustained, we envisage that the right/left adjunction may play a role between the conventional central dogma (DNA→RNA→mRNA) and the retrovirus that performs the reverse transcription (RNA→DNA). However, because indels and the splicing procedure in eukaryotic cells break the condition for the natural transformation and the right/left adjunction to exist, the irreversibility of cell differentiation is often observed as a consequence in biological creatures. If there is a limitation for functors, they cannot be systemized as objects (functor category) that can be used to compose hierarchical structures in some abstract algebraic manner; that may also mean an impossibility for cluster-based constructions in mathematical models of biological creatures. In other words, the possible non-existence of natural transformation and right/left adjunction may parallel the increase in entropy or other sorts of loss of order. Additionally, in the present model, some components emerge or are eliminated freely and that may also require more complex conditions for their mathematical formulation. This issue needs to be investigated in future studies.

Third, topics regarding 'T-invariance (transition-invariance)' and 'P-invariance (place-invariance)' are often discussed and often play an important role in the standard PNs but have not been introduced in the present article. One reason is that for T-invariance, a cyclic loop among objects is often incompatible with the diagrams of category theory. In these diagrams, 'commutativity' and 'uniqueness of a morphism except for isomorphism' have important meanings for the composition aspects of the theory; therefore, it is problematic if the composition of a cyclic loop using objects and morphisms is often difficult or non-essential. Our potentially compensatory idea is the following; in a diagram-like EPN, if $f_1;f_2$ and $g_1;g_2$ commute ($f_1; f_2 = g_1;g_2$) and therefore produce the same tokens in the output place, then commutativity may alternatively be expressed supposing $g_1^{-1}$ and $g_2^{-1}$ are duals of $g_1$ and $g_2$, respectively. Therefore, '$f_1;f_2;g_2^{-1};g_1^{-1}$ = Id' may hold under the condition that the above commutativity is not broken [see **Fig 14(B)**]. We envisage that this definition using cyclic loops is a 'T-invariant-like preserved' value in the present EPN model.

Similarly, P-invariance may be composable precisely because the number of tokens in the present EPN model can be systemized as an 'invariant value' that contains the changes in the number of tokens via the annihilation/creation morphism. Essentially, in the present EPN, assuming a 'state machine' and following changes in the number of tokens is possible, a law related to the invariant value may be established that governs the dynamic mechanics with strong consequences. Here, the incidence matrix-like products may play an important role with regard to the behavior of tokens (markings of the EPN). Further investigation is desired.

Fourth, the conditions for the respective places that stipulate the class to which the elements belong dictate overall processes. In category theory, for the composition of two or more morphisms, the target object and source objects must match. However, this postulate (compositionality between two objects) may be hard to fulfill if the morphisms are selected arbitrarily. Additionally, concepts such as limit, colimit, and tensor product are considered important. Therefore, these issues need to be explored.

Fifth, instances may occur where the self-referential paradox is unavoidable. For example, let $R_{(k\gamma)j}\bullet$ be a certain token of a clinical disease state; if some parts of the clinical phenomenon are explained from a genetic viewpoint, the global state $R_{(k\gamma)j}\bullet$ of a patient must include at least partially a token $D_{(k)j}\bullet$ (base sequences of DNA) or its coordinate. In this case, if some part of $D_{(k)j}$ has multiple biological roles, multiple biological-related outputs may be produced. There is no assurance whether any bases of the genome play a single role for a unique function. If some bases play multiple roles, the global clinical disease states must include multiple coordinates that stem from a unique part of the bases. Hence, such states should include at least $D_{(k)j}$ or its coordinates and in extremes, the token $R_{(k\gamma)j}\bullet$ must include a token $D_{(k)j}\bullet$ (base sequences of DNA) as a component. This line of reasoning derives from Cartesianism. Apart from the above issue, self-referential instances may occur in the coordinates where the same coordinates may be duplicated in some manner. In general, whether a certain token $R_{(k\gamma)j}\bullet$ (clinical disease states of a person) includes duplicated coordinates is extremely difficult to find because in most cases, the illness and results of a patient are often cloaked in another compartment or coordinates. In general, to our knowledge, in the composition of category theory, this contradiction issue is often avoided preliminarily and carefully at the stage of definition. In this regard, the word 'little' category is often used to avoid logical contradictions. In living bodies, feedback systems often cannot avoid self-referentials. Moreover, the areas where self-referential phenomena occur are considered to have more important meaning for systems in complex objects/creatures. Hence, it seems to us that the area to which non-self-referential system/tools apply is considerably limited. Complete linearity seems to be needed in objects such as cells, organs, and living bodies so that non-self-referential systems/tools can explain phenomena mathematically or completely. Nevertheless, the perspective of category theory may provide potential guidelines, viewpoints or clues to predict, analyze or comprehend complex phenomena/systems of such as living creatures. Ultimately, we may not be able to discriminate whether a certain token $R_{(k\gamma)j}\bullet$ includes only independent coordinates. For this reason, the display of integration/splitting of clinical disease states has a problem in that contradictions from paradoxes and self-referential issues are inherited. If parts of the coordinates are included in multiple tokens $R_{(k\gamma)j}\bullet$ and $R_{(k\gamma)l}\bullet$ in a duplicated manner, using these coordinates as factors that yield other results is inconvenient.

Sixth, for processes in living creatures, whether a natural transformation exists is ambiguous. In (molecular) biology, from a certain viewpoint, Eq (64) seems to express a potentially valuable message: the number of primary DNA sequences '$D_{(k)j}$' and that of their natural transformations α are equivalent, and the number of elements $F(D_{(k)j})$ in its set and its function (order) may be a bijection. To us, Yoneda's lemma (64) may potentially display the function of DNA '$D_{(k)j}$' and other biological functions and products including the canonical central dogma and the activities of cells, tissues, and organs. Although this may be only in appearance, a strict investigation is needed in the future. Furthermore, Eq (65) may be interpreted as biological orders/activities that may translate into layered structures. There may be a resonance or quasi-isomorphism between the activity of DNA and that of cells, organs, and living creatures. However, complete linearity in their phenomena is necessary for the enactment described by Eqs (64) and (65) in living bodies. However, biological mechanisms have a further complexity.

Even for DNA, misreading or errors are commonplace phenomena, and precise maps of orderings from sequences of DNAs and other processes seem rather rare phenomena in living bodies. In **Fig 15(B)**, described before (**#remark 3**), smaller objects that are composed of a DNA sequence ($D_{(k)j}$) may include also compartments of a larger DNA sequence ($D_{(k)l}$) as if program A invokes programs from program A itself within the same sequence of invocations (self-invoking) much like the programming language 'Pascal' [54, 55], which is composed of embedded structures [**Fig 15(B)**]. This self-invoking of the same DNA sequence such as a compartment of a single-strand DNA ($D_{(k)j}$ in place $p_k$) invoking an order to another compartment of the same DNA ($D_{(k)j}$) may occur casually in living creatures. Although such instances are considered expressible as an embedded structure [**Fig 15(B)**], they belong to non-linear phenomena and/or self-referencing phenomena in living bodies that also imply potential contradictions or that applying the EPN model is in principle impossible. Therefore, strict limitations and careful judgment are considered necessary for the treatment of natural transformations, Yoneda's lemma, Yoneda embedding, and right/left adjunction in the EPN model. We believe that one of the simplest candidates in which to observe phenomena to confirm the connection between living structure and mathematical laws is prokaryotic cells because their transcription from DNA/RNA to amino-acid sequences (proteins) is simple. In addition, it seems important to us that while **Fig 15(B)** may not hold precisely, the existence of functors between cells may be assured because, ordinarily, the entire sequences of DNA are equivalent in a single human being. This includes the occurrence of indels without complex splicing procedures. We believe that by considering identity units for the respective levels such as an identity base E for bases, an identity protein Δ for proteins, an identity cell □ for cells, an identity organ ◉ for organs, and an identity individual for individuals, the functor comprehension of a human body and the human community in a hierarchical-cluster form is possible [**Fig 15(B)**]. Therefore, the issue of possible duplication of coordinates and self-referencing in self-invoking expressions may be a potentially crucial problem; if possible, contradictions arising from duplicated coordinates should be investigated. There is, in principle, the possibility that the issue is insolvable.

Seventh, the visualization of the PN is envisaged to require more advanced abstract algebras such as Boolean algebra, Lie algebra, and calculus. For the moment, restraint is exercised because treating an unknown complex problem seems to quickly reveal many difficulties. Further exploration is desired.

Unfortunately, our approach is at present nascent, and the above issues are outside our area of expertise. However, we believe that our attempts, including the monoidal category expression of the PN, will contribute to exploring biological processes in a more sophisticated way through the application of the PN, groups, and related concepts. With a more well-conditioned approach and a more rigorous methodology, hopefully performed by mathematical researchers, further advances may be accomplished in the future.

## §6. Conclusions

Despite the lack of maturity and incompleteness of our methodology, the potential applicability of our hybrid model based on category theory and EPNs was illustrated using molecular/genetic biology input. Our simplistic dynamic/visual approach may yield, through additional studies, an abstract algebraic formulation.

## Supporting information

**S1 File. Citations for reference files are displayed.**
(PDF)

**S1 Appendix.**
(DOCX)

## Acknowledgments

The authors acknowledge Masahiko Tsuka, Daisuke Yasugi, Kaoru Sakamoto, Kazuo Yamada, Takashi Oshimo and Keiko Kojo for providing useful advice. We thank Richard Haase, Ph.D., from Edanz Group (https://www.edanz.com) for editing a draft of this manuscript.

## Author Contributions

**Conceptualization:** Jitsuki Sawamura.

**Methodology:** Jitsuki Sawamura, Shigeru Morishita, Jun Ishigooka.

**Supervision:** Shigeru Morishita, Jun Ishigooka.

**Validation:** Jitsuki Sawamura.

**Writing – original draft:** Jitsuki Sawamura.

**Writing – review & editing:** Jitsuki Sawamura.

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
