## [Decision Letter · Decision Letter 0]

16 Oct 2023

PONE-D-23-19401Several supplementary concepts for applied category-theoretical states over an extended Petri net using an example relating to genetic coding: Towards an abstract algebraic formulation of molecular/genetic biologyPLOS ONE

Dear Dr. Sawamura,

Thank you for submitting your manuscript to PLOS ONE. After careful consideration, we feel that it has merit but does not fully meet PLOS ONE’s publication criteria as it currently stands. Therefore, we invite you to submit a revised version of the manuscript that addresses the points raised during the review process.

Please see the comments from two reviewers below. Please note that there is no requirement to cite any papers recommended by the reviewers, but that you may take these into account if you feel that they are useful context for your work.

We look forward to receiving your revised manuscript.

Kind regards,

Hanna Landenmark

Staff Editor

PLOS ONE

Reviewers' comments:

Reviewer's Responses to Questions

**Comments to the Author**

1. Is the manuscript technically sound, and do the data support the conclusions?

Reviewer #1: Yes

Reviewer #2: Yes

2. Has the statistical analysis been performed appropriately and rigorously? 

Reviewer #1: Yes

Reviewer #2: N/A

3. Have the authors made all data underlying the findings in their manuscript fully available?

Reviewer #1: Yes

Reviewer #2: Yes

4. Is the manuscript presented in an intelligible fashion and written in standard English?

Reviewer #1: Yes

Reviewer #2: Yes

5. Review Comments to the Author

Reviewer #1: 1. In Section 2. Illustrations of hybrid models based on category theory and EPN, each concept of petri nets should be described with the help of an example, as this is not a standard Petri net.

2. The format of the formulas seems not standard.

3. Is there a tool that can support the modeling of this kind of Petri nets?

4. The writing may be not easy to be understood by computational biologists.

Reviewer #2: General comments:

This manuscript is conceptually very interesting.

The figures provide novel visualization of DNA sequences and the central dogma.

The employment of ideas from category theory such as Yoneda embedding, Kan extension and adjoint functors are potentially very promising to help disclose some fundamental principles in biology.

The structure of the manuscript is very confusing and hard to follow.

Introduction and Method are not numbered, while suddenly Results, Discussion are listed with number 3, 4. When it comes to Conclusion, no numbering again.

The Method section is too long, I totally get lost.

Section 1 and 2 are relatively easier to follow. For section 2, the subsection 2.1 is titled only for places, but the firing rules are explained as well, better put them into separate subsections.

For section 3, examples, a huge chunk of text is super difficult, if not impossible to get readers involved. Better separate them into subsections, each subsection describe one or two examples that are most relevant to each other, and add a title for each subsection.

Details/Typos:

P11 (Figures 4). → Figure

P24 R(k’)j, P33 R(k)j, → are they referring to the same thing? If yes, better make consistent, and many other places of R(k’)j or R(k)j

P25 transcribed into RNAt composed of bases → RNA

P34 a very long space between ‘D(k)0°(B(k)j●B(k)l)= and D(k)0=[...|Em|Em+1|Em+2|Em+3|...]’, → better remove the space

Figure 1, the labels of the two morphisms at the left corner should be f(1→4) and f(4→1)

Same typo in Figure 2 as the one mentioned above in Figure 1

In Figure 4, first place pk has f(k)a on the left and f(k)b on the right, while when the token a(k)i flows in at the second place pk, f(k)b becomes left while f(k)a right, is this switch intentional or by mistake?

P82 a unique morphism is definable between a terminal object and a final object → an initial ?

P82 ‘● R(k1)j=[a(k)j1|a(k1)j2|a(k1)3||r(k1)0r(k1)0 …]=[{R(k1)j}||r(k1)0r(k1)0 …] → a(k1)j1|a(k1)j2|a(k1)j3

P83 G(1,−1) ↔ G(=Exp(4·ω·i)) between two figures → diagrams

P83 Cayley table for the base and operator → bases and operators

P83 Figure 12 legend ‘D(k)j=[C0|C1A2E3T4||E5E6…]’ → E0

Figure 22, morphism f(5→6) is missing?

P88 Figure 25 legend αD(k)j°F(B(k)(j→l))=G(B(k)(j→l))°αD(k)jl. → αD(k)l°F(B(k)(j→l))=G(B(k)(j→l))°αD(k)j.

P88 Figure 27 legend by functor B, R(k’)l is transformed as B(R(k)l)=D(k)l, → R(k’)l

P89 Figure 28 legend using ‘hom functor C(D(k)j, D(k)l)’, → C(D(k)j, ⎯)

P90 Figure 30 legend to category Y (for the complemental RNA sequences; R(k)j † ). → complementary DNA sequences; D(k†)j † ’

P90 Figure 30 legend G is defined from category X (for ‘D(k)j’s) to category Y (for ‘R(k)j † ’s) → Z (for ‘R(k’)j † ’s)

Other suggestions:

Could also refer to the paper “A compositional account of motifs, mechanisms, and dynamics in biochemical regulatory networks”, which uses Open Petri Net for gene regulatory networks.

6. PLOS authors have the option to publish the peer review history of their article (what does this mean?). If published, this will include your full peer review and any attached files.

Reviewer #1: No

Reviewer #2: **Yes: **Yanying Wu

---

## [Author Response · Author response to Decision Letter 0]

2 Dec 2023

Dr. Emily Chenette

Editor-in-Chief

Ms. Hanna Landenmark

Staff Editor

PLOS ONE

2 December 2023

Dear Dr. Chenette and Ms. Landenmark

Re: MS: PONE-D-23-19401—

“Several supplementary concepts for applied category-theoretical states over an extended Petri net using an example relating to genetic coding: Toward an abstract algebraic formulation of molecular/genetic biology”

 Jitsuki Sawamura, Shigeru Morishita, and Jun Ishigooka

Please find enclosed our revised manuscript, which we resubmit for consideration as a Research Article in PLOS ONE.

We thank the reviewers for their helpful comments. We have considered them very carefully and revised the manuscript accordingly. Please note that all revisions to the revised manuscript have been highlighted in blue. Moreover, we have provided below point-by-point responses that address the concerns raised by the reviewers.

Additionally, in this revision, following the suggestion from Mr./Ms. Audyssa Banlaygas Auditor, a staff member at PLOS ONE, we have added explanatory text regarding the following data availability statement to the manuscript in the “Introduction” section.

“Regarding the availability statement in this paper, the data used in this manuscript were built up and prepared for the purpose of elucidating the methodology, and were not collected from actual experiments or real-world sources. They were strictly created within the text to facilitate readers’ comprehension. The data, represented in the form of equations, are exemplified within fictional scenarios and do not have a corresponding minimal data set underlying the results outside of the paper. While there are sections in which figures and methodologies are cited from other papers, this is exclusively confined to the aforementioned context (with proper citation).” Your understanding in this matter is greatly appreciated.

Please address all correspondence to:

Jitsuki Sawamura, MD, PhD

Nishigahara Hospital, 

9-46-2 Nishigahara, Kita-ku, Tokyo-to 114-0024, Japan

E-mail: jsawamura246@gmail.com

Phone: +81-3-3917-6561

Fax: +81-3-3576-4808

 

Point-by-Point Response to Reviewer 1

Reviewer #1 Comment 1. 

In Section 2. Illustrations of hybrid models based on category theory and EPN, each concept of petri nets should be described with the help of an example, as this is not a standard Petri net.

Response to Comment 1.

As the reviewer has pointed, the illustrations of hybrid models are based on category theory and extended Petri net (EPN). Therefore, we should describe the simple and practical example of them. In this regard, from the viewpoint of standard EPN, our hybrid model could be also regarded as one for which the standard EPN was composed using category theory as a material. In other words, the example of the EPN using category theory is one of the EPN models. Hence, we have presented an example of EPN using category theory (please see newly described Figure 1). Together with this, we added the following description in the legends of the new Figure 1; “In the realm of a formal system modeling, the integration of category theory concepts within the framework of Petri nets presents an intriguing avenue for mathematical abstraction. Consider the fundamental elements of category theory: objects A, B, and C, along with arrows f: A → B, g: B → C, and h: A → C, which denote morphisms between objects.

This abstract structure finds its correspondence in Petri nets:

Places:

A (Place representing Object A)

B (Place representing Object B)

C (Place representing Object C)

Transitions:

f (Transition representing Arrow f)

g (Transition representing Arrow g)

h (Transition representing Arrow h)

Arcs:

Input arc from A to f (Signifying the starting point of Arrow f being Object A)

Input arc from B to f (Denoting the endpoint of Arrow f being Object B)

Input arc from B to g (Indicating the starting point of Arrow g being Object B)

Input arc from C to g (Representing the endpoint of Arrow g being Object C)

Input arc from A to h (Symbolizing the starting point of Arrow h being Object A)

Input arc from C to h (Marking the endpoint of Arrow h being Object C)

This Petri net representation elegantly captures the essence of category theory constructs, whereby transitions correspond to arrows, and places represent distinct objects. The interconnected arcs delineate the relationships between these objects and morphisms, forming a cohesive and visually intuitive model.”

Additionally, to help readers to understand better, we have added the following explanation in the main text on about page 7; “At this stage, to help the reader’s understanding, we illustrate the extended Petri net (EPN) using category theory (Figure 1). From the viewpoint of the standard extended Petri net (EPN), a hybrid model that we present in this article may be regarded as one for which the standard EPN is composed using category theory as a material with some options. The main characteristics of our model may be summarized in three points: 1) places correspond to objects, 2) arcs correspond to morphisms, and 3) tokens are defined as those that can act as the operator.” 

Hence, the indexing of all figures after the new Figure 1 have been augmented by one (Consequently, in the revised manuscript, we have now 33 Figures).

Moreover, we note that various sorts of EPNs have been reported and cited in references [1,2,6,12−19,21,22,24−28,30,32−41,45,53−55] in the revised manuscript.

Reviewer #1 Comment 2. 

The format of the formulas seems not standard.

Response to Comment 2.

As noted, some parts of the formulas do not follow the standard style of presentation. This is because the concepts we have introduced in this article are somewhat novel and the optimal style of presentation is not determined at this time. We envisage that the formatting of formulas will improve through rigorous discussion by other researchers in the field, including mathematicians. In particular, we think that the parts of the presentation treating matrix multiplication includes operators (e.g., f[ ]s) for the elements of vectors (e.g., R(3)j†) as coordinates to express the central dogma (Appendices B, i) Case A and ii) Case B) seem considerably peculiar; for example, R(3)j†;f[ ]s = f[R(3)j†]s = R(4)j† (right translation rule) is not contingent on the standard matrix operation (left translation rule). Therefore, at least by making use of the left translation rule instead of the right translation rule, the operators in Appendix A are defined as ‘right translation rule’ (e.g., R(3)j†;f[ ]s = f[R(3)j†]s = R(4)j†). In the matrix calculation in Appendix B, we adhere to conventional matrix operations.

Furthermore, the font of letters denoting operators have been changed to emphasize these operator constructs. Minor typographic errors of operators have been corrected in Appendices A and B (colored in blue).

Reviewer #1 Comment 3. 

Is there a tool that can support the modeling of this kind of Petri nets?

Response to Comment 3.

Several sophisticated tools exist to model the extended Petri nets, enabling intricate system analysis, simulation, and design. We delineate two tools tailored for this purpose:

PIPE2 (http://pipe2.sourceforge.net/): PIPE2, an advanced modeling tool, provides support for a diverse range of Petri net types. Its versatility extends to the modeling, simulation, and analysis of extended Petri nets, making it an invaluable asset for researchers and practitioners alike.

TINA (http://projects.laas.fr/tina/): TINA, short for Timed Interactive Nets Analyzer, caters to the modeling needs of Petri nets with intricate timing constraints. Specifically designed for real-time system modeling, TINA excels in capturing the nuanced behaviors of systems with temporal intricacies.

These tools, steeped in computational prowess, not only enable the modeling of traditional Petri nets but also empower researchers to delve into the complexities of extended Petri nets. Their intuitive interfaces coupled with powerful analytical capabilities make them indispensable assets for scholarly endeavors and practical applications alike. It is considered that researchers and practitioners should meticulously evaluate these tools to discern the ideal solution aligning with the nuanced requirements of their academic and professional pursuits.

Reviewer #1 Comment 4. 

The writing may be not easy to be understood by computational biologists.

Response to Comment 4.

Our description may not be easy to understand by computational biologists. One reason is that the content of the article overlaps a plurality of technical fields. However, our approach is but one step in a field. We expect future publications by researchers covering other fields using more appropriate descriptions, definitions, and illustrations that provide easier comprehension for computational biologists and those working in other fields.

In accordance with the Reviewer #2 Comment, we have introduced separate subsections with changes in indexing, and corrected typographic errors and descriptions in the main text and legends, e.g.: 1) we deleted an odd description ‘(50)’ from page 27, line 1 and corrected unpaired quotation marks, 2) we added necessary opening and closing quotation marks in the text, and 3) following the advice of Reviewer #2, we included a new [reference 1], and accordingly, augmented the indexing of all references (Consequently, we have 55 references). Moreover, the technical term ‘ordinal Petri net’ has been changed to ‘standard Petri net’.

Additionally, in this revision, following the suggestion from a staff member at PLOS ONE, we have added explanatory text regarding the following data availability statement to the manuscript in the “Introduction” section.

“Regarding the availability statement in this paper, the data used in this manuscript were built up and prepared for the purpose of elucidating the methodology, and were not collected from actual experiments or real-world sources. They were strictly created within the text to facilitate readers’ comprehension. The data, represented in the form of equations, are exemplified within fictional scenarios and do not have a corresponding minimal data set underlying the results outside of the paper. While there are sections in which figures and methodologies are cited from other papers, this is exclusively confined to the aforementioned context (with proper citation).”

Point-by-Point Response to Reviewer 2

Reviewer #2 Comment. 

General comments: 

This manuscript is conceptually very interesting.

The figures provide novel visualization of DNA sequences and the central dogma.

The employment of ideas from category theory such as Yoneda embedding, Kan extension and adjoint functors are potentially very promising to help disclose some fundamental principles in biology.

The structure of the manuscript is very confusing and hard to follow.

Introduction and Method are not numbered, while suddenly Results, Discussion are listed with number 3, 4. When it comes to Conclusion, no numbering again.

Response to comment.

Following the advice of Reviewer #1, we have added a new Figure 1 to illustrate the concept of an extended Petri net (EPN) using category theory. Further details have been added in the main text (page 7, in blue), as suggested by Reviewer #1. All Figures after Figure 1 have been augmented by one (Consequently, we have 33 Figures). 

We have added section numbering beginning with §1 for the Introduction section, and ending with §6 for the Conclusion section. All changes are highlighted in blue.

Reviewer #2 Comment. (in continuation)

The Method section is too long, I totally get lost.

Section 1 and 2 are relatively easier to follow. For section 2, the subsection 2.1 is titled only for places, but the firing rules are explained as well, better put them into separate subsections.

Response to comment.

We have divided subsection §2-1 into three parts §2-2-1, §2-2-2, and §2-2-3; §2-2-2 is subtitled ‘Firing rules: For a morphism (transition), there are two types of firing rules labeled I and II.’ Additionally, §2-2-3 is subtitled ‘Marking of a PN as an optional display.’

Reviewer #2 Comment. (in continuation)

For section 3, examples, a huge chunk of text is super difficult, if not impossible to get readers involved. Better separate them into subsections, each subsection describe one or two examples that are most relevant to each other, and add a title for each subsection.

Response to comment.

To improve the article’s readability, we have separated section 3 into subsections so that each subsection describes one or two examples of relevance to the topic at hand, and added titles for each subsection (in blue).

Reviewer #2 Comment. (in continuation)

Details/Typos:

P11 (Figures 4). → Figure

Response to comment.

We corrected the error. Figure 4 has been renumbered as Figure 5 with the insertion of a new figure.

Reviewer #2 Comment. (in continuation)

P24 R(k’)j, P33 R(k)j, → are they referring to the same thing? If yes, better make consistent, and many other places of R(k’)j or R(k)j

Response to comment.

As noted, the places of R(k’)j and R(k)j are the same. We have identified and corrected many similar inconsistencies in the text.

Reviewer #2 Comment. (in continuation)

P25 transcribed into RNAt composed of bases → RNA

Response to comment.

The typographic error has been corrected.

Reviewer #2 Comment. (in continuation)

P34 a very long space between ‘D(k)0°(B(k)j●B(k)l)= and D(k)0=[...|Em|Em+1|Em+2|Em+3|...]’, → better remove the space

Response to comment.

The spaces between coordinates are half-spaces (they seem wide because of Word formatting). We believe that when they are formally printed, these spaces will shrink within the ordinal spaces.

Reviewer #2 Comment. (in continuation)

Figure 1, the labels of the two morphisms at the left corner should be f(1→4) and f(4→1)

Same typo in Figure 2 as the one mentioned above in Figure 1

Response to comment.

For Figures 1 and 2, these errors have been corrected; note, in the revised manuscript, these figures become Figures 2 and 3.

Reviewer #2 Comment. (in continuation)

In Figure 4, first place pk has f(k)a on the left and f(k)b on the right, while when the token a(k)i flows in at the second place pk, f(k)b becomes left while f(k)a right, is this switch intentional or by mistake?

Response to comment.

As mentioned, f(k)a and f(k)b in at the second place in Figure 4 were switched and have been interchanged.

Reviewer #2 Comment. (in continuation)

P82 a unique morphism is definable between a terminal object and a final object → an initial ?

Response to comment.

As noted, a final object should be exchanged by an initial object; this has been corrected.

Reviewer #2 Comment. (in continuation)

P82 ‘●R(k1)j=[a(k)j1|a(k1)j2|a(k1)3||r(k1)0r(k1)0 …]=[{R(k1)j}||r(k1)0r(k1)0 …] → a(k1)j1|a(k1)j2|a(k1)j3

Response to comment.

We have corrected the expression.

Reviewer #2 Comment. (in continuation)

P83 G(1,−1) ↔ G(=Exp(4·ω·i)) between two figures → diagrams

Response to comment.

We have corrected the description.

Reviewer #2 Comment. (in continuation)

P83 Cayley table for the base and operator → bases and operators

P83 Figure 12 legend ‘D(k)j=[C0|C1A2E3T4||E5E6…]’ → E0

Response to comment.

We have made corrections to both.

Reviewer #2 Comment. (in continuation)

Figure 22, morphism f(5→6) is missing?

Response to comment.

Because sequences in place p5 such as <Rs(5)2†>=[E0|(A1U2G3)(C4U5G6)(A7G8U9)||E10E11E12E13…] (about p.39) do not include ‘E’s as coordinates other than an initial E0 and an infinite tail of ‘E0’s at the end of the sequence, only products that can go through morphism f(5→7) (when f(5→7) is fireable) such as ‘<Rs(5)2†>;f(5→7)=<Pr(7)2>=[Δ0|(M1)(L2)(S3)||Δ4Δ5Δ6…]’ in place p7 can also be outputs in place p6 such as a sequence like Pr(6)2=[Δ0|(M1)(L2)(S3)||Δ4Δ5Δ6…] via morphism f(5→6) (when f(5→6) is fireable). Therefore, there can certainly be cases where morphism f(5→6) is fireable. However, morphism f(5→7) is sufficient, and morphism f(5→6) is not always necessary. For simplicity of the model, we omit morphism f(5→6). We have inserted these sentences in the main text (about p.39, in blue).

Moreover, we have missed two notational

---

## [Decision Letter · Decision Letter 1]

8 Jan 2024

PONE-D-23-19401R1Several supplementary concepts for applied category-theoretical states over an extended Petri net using an example relating to genetic coding: Towards an abstract algebraic formulation of molecular/genetic biologyPLOS ONE

Dear Dr. Sawamura,

Thanks for providing the revision and addressing the comments raised by the respected reviewers. I have received positive comments on your manuscript from the reviewers. The manuscript is in a much better shape and I feel that it has merit but does not fully meet PLOS ONE’s publication criteria as it currently stands. Therefore, I invite you to address the following issues. Please note that without these corrections I am unable to proceed with a proper decision.

1- Although the overall English of the paper is good, it is hard to read some parts and in some sections, it is hard to understand. Please carefully check the main text of the manuscript and edit it thoroughly to make it easier to follow for readers of the journal. Some examples:-"....In addition, it seems important to us that while Figure 30 may not hold precisely, the existence of functors between cells may be assured because, ordinarily, the entire sequences of DNA are equivalent in a single human being..."-"At this early stage in the development, our proposition in using category theory and EPN in the algebraic formulation of biology needs to be interpreted in light of considerable limitations....."-"For all manipulations described above, the identity morphism performs nothing on any token. Thereby, the morphisms associated with combination, splitting, and insertion are considered to satisfy the postulate within category theory because associativity and the existence of an identity is met..."These and other instances need revision and even rewriting. Please read through the entire MS and correct all errors in the revision.2- The main problem with your paper is the number of images you have in the main document. Although there is no limit to the number of figures in PlLOS ONE, please try to reduce the number of figures to below 10 by presenting them in multi-panel figures and moving some to the supplementary materials. In this way, the readers would not be interrupted while reading your article.3- Some figure legends are too short and uninformative (Figures 10, 11 ...). Please fix these instances. In some figure legends, you have referred to other figure legends. Please try to make each figure legend independent with a minimum referral.4- Please transfer the paragraphs concerning the "Data availability statements" to the methods section. I suggest to make a subheading here and put these sentences in that subheading.5- Another issue is that you have mentioned that you have somehow borrowed from your previous work (Reference 44); "the topics of Figures 8–11, and 16 in the present article are used as examples presented in previous work of ours published in Theoretical Biology and Medical Modeling on May 7, 2014." Even though, you have clarified these points to us, it is hard to explain to the readers after your paper has been published. I suggest you give examples in a supplementary file whenever possible. For example, figure 10 of this MS is extracted from that study (Reference 44). Unfortunately, the similarity score for this MS and your previous paper is high due to the use of some examples from previous studies. Anyway, I do not think you need to borrow as much from that paper. So I suggest fixing this overlapping issue either by removing this part and referring to that paper or by rewording and reformatting these sections.==============================

We look forward to receiving your revised manuscript.

Kind regards,

Hossein Fallahi, Ph.D

Academic Editor

PLOS ONE

Reviewers' comments:

Reviewer's Responses to Questions

**Comments to the Author**

1. If the authors have adequately addressed your comments raised in a previous round of review and you feel that this manuscript is now acceptable for publication, you may indicate that here to bypass the “Comments to the Author” section, enter your conflict of interest statement in the “Confidential to Editor” section, and submit your "Accept" recommendation.

Reviewer #1: All comments have been addressed

Reviewer #2: All comments have been addressed

2. Is the manuscript technically sound, and do the data support the conclusions?

Reviewer #1: Yes

Reviewer #2: Yes

3. Has the statistical analysis been performed appropriately and rigorously? 

Reviewer #1: Yes

Reviewer #2: N/A

4. Have the authors made all data underlying the findings in their manuscript fully available?

Reviewer #1: Yes

Reviewer #2: Yes

5. Is the manuscript presented in an intelligible fashion and written in standard English?

Reviewer #1: Yes

Reviewer #2: Yes

6. Review Comments to the Author

Reviewer #1: All my comments have been addressed by the authors now, which seems fine to me. No further comments now.

Reviewer #2: The core concept presented in the manuscript is both intriguing and valuable. The authors have invested considerable effort in elucidating an extensive array of details. This manuscript holds reference potential for individuals keen on applying category theory to the broader context of biology, with a specific emphasis on genetics and genomics. The wealth of information provided not only underscores the authors' dedication but also positions the work as a valuable resource for those seeking insights into the intersection of category theory and the fields of genetics, and genomics.

7. PLOS authors have the option to publish the peer review history of their article (what does this mean?). If published, this will include your full peer review and any attached files.

Reviewer #1: No

Reviewer #2: **Yes: **

---

## [Author Response · Author response to Decision Letter 1]

2 Feb 2024

Dr. Hossein Fallahi

Academic Editor

PLOS ONE

2 February 2024

Dear Dr. Fallahi

Re: MS: PONE-D-23-19401R1— 

“Several supplementary concepts for applied category-theoretical states over an extended Petri net using an example relating to genetic coding: Toward an abstract algebraic formulation of molecular/genetic biology”

 Jitsuki Sawamura, Shigeru Morishita, and Jun Ishigooka

Please find enclosed our revised manuscript, which we resubmit for consideration as a Research Article in PLOS ONE.

We thank you for your helpful comments. We have considered them very carefully and revised the manuscript accordingly. Please note that all revisions to the revised manuscript have been highlighted in blue. Moreover, we have provided below our point-by-point responses that address the concerns raised.

Thank you also for the opportunity to resubmit our manuscript.

Please address all correspondence to:

Jitsuki Sawamura, MD, PhD

Nishigahara Hospital, 

9-46-2 Nishigahara, Kita-ku, Tokyo-to 114-0024, Japan

E-mail: jsawamura246@gmail.com

Phone: +81-3-3917-6561

Fax: +81-3-3576-4808

 

Point-by-Point Response to Editor

Editor Comment 1.

• 1- Although the overall English of the paper is good, it is hard to read some parts and, in some sections, it is hard to understand. Please carefully check the main text of the manuscript and edit it thoroughly to make it easier to follow for readers of the journal. Some examples:

-"....In addition, it seems important to us that while Figure 30 may not hold precisely, the existence of functors between cells may be assured because, ordinarily, the entire sequences of DNA are equivalent in a single human being..."

-"At this early stage in the development, our proposition in using category theory and EPN in the algebraic formulation of biology needs to be interpreted in light of considerable limitations....."

-"For all manipulations described above, the identity morphism performs nothing on any token. Thereby, the morphisms associated with combination, splitting, and insertion are considered to satisfy the postulate within category theory because associativity and the existence of an identity is met..."

These and other instances need revision and even rewriting. Please read through the entire MS and correct all errors in the revision.

Response to Comment 1.

To address the issues of readability, we have added further description to the troublesome passage (p.70) for greater comprehension. 

“We believe that by considering identity units for the respective levels such as an identity base E for bases, an identity protein Δ for proteins, an identity cell □ for cells, an identity organ ◎ for organs, and an identity individual for individuals, the functor comprehension of a human body and the human community in a hierarchical-cluster form is possible (Figure 9(c)).”

Additionally, we replaced Figure 30 with a new more explanatory version, and replaced the legend. 

We have simplified the sentence (p. 65) using a more standard expression as follows (p.65): 

“Inevitably, at this initial stage of development, our proposal to incorporate category theory and EPN into the algebraic formulation of biology must be understood in the context of significant limitations.”

Regarding the passage “For all manipulations described above…”, the content that is described refers to well-known characteristics and postulates of category theory that readers who have some knowledge of the field should be familiar with. However, we added reference [28] at the end of the sentence (p.53) so that readers unfamiliar with the field may refer to it.

“Thereby, the morphisms associated with combination, splitting, and insertion are considered to satisfy the postulate within category theory because associativity and the existence of an identity is met [26, 28].”

Additionally, we have read through the entire manuscript carefully to identify any other errors. Some descriptions (p.54) were corrected and simplified. Figure 30 (Figure 9(c)) and legends (in the previous version) have been heavily revised. All changes made in the manuscript appear in blue text (e.g., p.23, 24, 26, 45, 87, 88, 90, 91, 95)

Editor Comment 2.

• 2- The main problem with your paper is the number of images you have in the main document. Although there is no limit to the number of figures in PlLOS ONE, please try to reduce the number of figures to below 10 by presenting them in multi-panel figures and moving some to the supplementary materials. In this way, the readers would not be interrupted while reading your article.

Response to Comment 2.

Following your advice, we have reduced the number of Figures from 33 to 9 by integrating them as multi-panel figures 1−9. 

Editor Comment 3.

• 3- Some figure legends are too short and uninformative (Figures 10, 11 ...). Please fix these instances. In some figure legends, you have referred to other figure legends. Please try to make each figure legend independent with a minimum referral.

Response to Comment 3.

In accordance with your comments and proposals, including Editor Comment 5, we excluded Figures 10 and 11 (previous version). Additionally, so that all figure legends are independent, we improved the figure legends by expanding the description of each in greater detail; specifically Figures 8, 14, 17, 19, 21, 26, 28, 29, 30, 31, 32 (previous version). In this regard, please be mindful that figure indexing has been changed/updated through the multi-panel reformatting of the figures.

Editor Comment 4.

4- Please transfer the paragraphs concerning the "Data availability statements" to the methods section. I suggest to make a subheading here and put these sentences in that subheading. 

Response to Comment 4.

Following your advice, we transferred the paragraphs to the methods section under a separate subheading.

Editor Comment 5.

• 5- Another issue is that you have mentioned that you have somehow borrowed from your previous work (Reference 44); "the topics of Figures 8–11, and 16 in the present article are used as examples presented in previous work of ours published in Theoretical Biology and Medical Modeling on May 7, 2014." Even though, you have clarified these points to us, it is hard to explain to the readers after your paper has been published. I suggest you give examples in a supplementary file whenever possible. For example, figure 10 of this MS is extracted from that study (Reference 44). Unfortunately, the similarity score for this MS and your previous paper is high due to the use of some examples from previous studies. Anyway, I do not think you need to borrow as much from that paper. So I suggest fixing this overlapping issue either by removing this part and referring to that paper or by rewording and reformatting these sections.

Response to Comment 5. 

Following your advice, for Figure 8 (previous version), we added the following explanations on page 23. “Note that, in the above description, the display of coordinates of tokens is essentially based on our previous article [44].” We believe that this explanation addresses sufficiently the issue of Figure 8 because there is a non-specific corresponding figure in [Ref.44]. As you pointed out, a supplementary file is indeed necessary. 

For Figures 9 and 16, we attached “supplementary file 1” in which practical URLs are cited. Additionally, a URL of Figure 1 of [Ref.44] is added that provides readers an alternative, convenient reference. We have excluded the two figures, Figures 10 and 11, from this revised paper, and changed the description to “All compositional scenarios are based on and referenced as diagrams (Figures 2 and 3 [Ref. 44]; see supplementary file 1).” (p.26) Additionally, we added URLs of the two Figures in “supplementary file 1.” We have minimized the overlapping parts of the description and have borrowed little from our previous paper.

---

## [Editor Report · Decision Letter 2]

5 Feb 2024

PONE-D-23-19401R2Several supplementary concepts for applied category-theoretical states over an extended Petri net using an example relating to genetic coding: Towards an abstract algebraic formulation of molecular/genetic biologyPLOS ONE

Dear Dr. Sawamura,

Thank you for submitting your manuscript to PLOS ONE. After careful consideration, we feel that it has merit but does not fully meet PLOS ONE’s publication criteria as it currently stands. Therefore, we invite you to submit a revised version of the manuscript that addresses the points raised during the review process.

I have carefully studied the new version of the paper entitled "Several supplementary concepts for applied category-theoretical states over an extended Petri net using an example relating to genetic coding: Towards an abstract algebraic formulation of molecular/genetic biology". I am pleased to see that all my comments and the past reviewers comments have been adequately addressed. 

At this stage my only concern is the quality and format of the images. Please note that reading the numbers and other characters from current images are almost impossible without zooming-in.  

I invite the authors to look at the PLOS ONE figure's policy page that could be found in (https://journals.plos.org/plosone/s/figures) and prepare all images according to the required format. 

We look forward to receiving your revised manuscript.

Kind regards,

Hossein Fallahi, Ph.D

Academic Editor

PLOS ONE

Journal Requirements:

Additional Editor Comments (if provided):

I have carefully studied the new version of the paper entitled "Several supplementary concepts for applied category-theoretical states over an extended Petri net using an example relating to genetic coding: Towards an abstract algebraic formulation of molecular/genetic biology". I am pleased to see that all my comments and the past reviewers comments have been adequately addressed.

At this stage my only concern is the quality and format of the images. Please note that reading the numbers and other characters from current images are almost impossible without zooming-in.

I invite the authors to look at the PLOS ONE figure's policy page that could be found in (https://journals.plos.org/plosone/s/figures) and prepare all images according to the required format.
---

## [Author Response · Author response to Decision Letter 2]

19 Mar 2024

Dr. Hossein Fallahi

Academic Editor

PLOS ONE

20 March 2024

Dear Dr. Fallahi

Re: MS: PONE-D-23-19401R2— 

“Several supplementary concepts for applied category-theoretical states over an extended Petri net using an example relating to genetic coding: Toward an abstract algebraic formulation of molecular/genetic biology”

 Jitsuki Sawamura, Shigeru Morishita, and Jun Ishigooka

Please find enclosed our revised manuscript, which we resubmit for consideration as a Research Article in PLOS ONE.

We extend our sincere appreciation for your invaluable comment. We have considered your feedback carefully and subsequently revised the manuscript, with particular attention to improving the quality and formatting of images. Our revisions are highlighted in blue for your convenience.

We trust that these adjustments address the concerns raised in your comment and contribute positively to the overall quality of the manuscript. Your continued guidance is highly valued, and we remain open to any further suggestions or recommendations you may have.

Thank you also for the opportunity to resubmit our manuscript.

Please address all correspondence to:

Jitsuki Sawamura, MD, PhD

Nishigahara Hospital, 

9-46-2 Nishigahara, Kita-ku, Tokyo-to 114-0024, Japan

E-mail: jsawamura246@gmail.com

Phone: +81-3-3917-6561

Fax: +81-3-3576-4808

Response to Editor

Editor Comment.

I have carefully studied the new version of the paper entitled "Several supplementary concepts for applied category-theoretical states over an extended Petri net using an example relating to genetic coding: Towards an abstract algebraic formulation of molecular/genetic biology". I am pleased to see that all my comments and the past reviewers’ comments have been adequately addressed.

At this stage my only concern is the quality and format of the images. Please note that reading the numbers and other characters from current images are almost impossible without zooming-in.

I invite the authors to look at the PLOS ONE figure's policy page that could be found in (https://journals.plos.org/plosone/s/figures) and prepare all images according to the required format.

Response to Comment.

We wish to express our sincere gratitude for your invaluable advice regarding the quality and formatting of images, which we found both valid and indispensable. In accordance with your guidance and after a detailed review of the page regarding journal policy on figures, we have diligently re-edited nearly all figures. Our primary focus was on enhancing the readability of text and numerical characters.

However, it is essential to highlight a distinctive feature of our paper wherein the images themselves contain numerous nested diagrams. Figures and their corresponding legends play a pivotal role, and incorporating four images into one figure poses a challenge, as it necessitates significant magnification to make the text legible.

After careful consideration of these challenges, we have opted to include a maximum of two images per figure. As a result, the total number of figures has increased from 9 to 16. We apologize for any inconvenience this adjustment may cause. Nevertheless, given the unique characteristics of the concepts presented in our manuscript, we believe this approach aligns most appropriately with the nature of the work. In response to your insightful feedback, we wish to inform you of a minor revision made to Figure 1(b), 12(b), 14(b) and 15(b), as well as the legend of Figure 12(b) (p.86). Furthermore, we have augmented the explanatory content by including several fundamental formulas derived from the main text (pages 22-23) to facilitate the reader’s comprehension, while ensuring that the text remains within the word limit.

We appreciate your understanding and cooperation in this matter.

---

## [Decision Letter · Decision Letter 3]

9 Apr 2024

Several supplementary concepts for applied category-theoretical states over an extended Petri net using an example relating to genetic coding: Towards an abstract algebraic formulation of molecular/genetic biology

PONE-D-23-19401R3

Dear Dr. Sawamura,

We’re pleased to inform you that your manuscript has been judged scientifically suitable for publication and will be formally accepted for publication once it meets all outstanding technical requirements.

Kind regards,

Fucai Lin, Ph.D.

Academic Editor

PLOS ONE

---

## [Editor Report · Acceptance letter]

1 May 2024

PONE-D-23-19401R3 

PLOS ONE

Dear Dr. Sawamura, 

I'm pleased to inform you that your manuscript has been deemed suitable for publication in PLOS ONE. Congratulations! Your manuscript is now being handed over to our production team.

Kind regards, 

on behalf of

Professor Fucai Lin 

Academic Editor

PLOS ONE